# Fructose-1,6-bisphosphate couples glycolytic activity to cell adhesion

Lennart Hoffmann [1,2], Marlen Duchmann[1], Katina Lazarow[3],
Yun-Hsuan Huang[4], Fabian Lukas[1], Wen-Ting Lo [5], Regina Feil[6],
Christopher Schmied [7], Martin Lehmann [7], John E. Lunn [6],
Ilaria Piazza [4,8,9], Jens P. von Kries[3], Volker Haucke [5,10,11] &
Tanja Maritzen [1,2] ✉

Cellular adhesion to the extracellular matrix is essential for morphogenesis, tissue integrity and survival signalling. The best understood adhesion structures are focal adhesions (FAs). In spite of their importance, our knowledge of upstream factors that integrate FA dynamics with other cellular processes, such as metabolism, remains fragmentary. Using a genome-wide screen, we identify aldolase A, a key glycolytic enzyme that converts fructose-1,6-bisphosphate (FBP), as a regulatory switch that links metabolic flux to FA assembly and cell morphogenesis. We show that cellular FBP serves as a signalling metabolite, which transmits information about the metabolic cell state to the actin-based machinery for cell adhesion and protrusion. This mechanism involves FBP binding to the Rac1 inhibitor RCC2 and a concomitant elevation of Rac1 activity resulting in actin reorganization, increased FA assembly and elevated protrusive activity. Here we predict this mechanism to be crucial for processes ranging from development to cancer.

Cell adhesion to the extracellular matrix is essential for numerous cellular processes including cell shape regulation, proliferation and differentiation[1]. The best characterized cell-matrix adhesion structures are focal adhesions (FAs) comprising heterodimeric integrin transmembrane receptors, which connect the extracellular matrix to the actin cytoskeleton. In addition to integrins, FAs contain scaffold proteins, adaptor proteins and enzymes regulating processes such as mechanotransduction and cell signalling[2-4]. The functional importance of FAs is reflected by the fact that their components are linked to diseases including osteoporosis[5], cardiovascular disease, muscular dystrophy[1] and cancer. In fact, a considerable integrin adhesome subset is connected to tumour progression[6].

FAs are dynamic structures, and their assembly and disassembly are tightly regulated. FA assembly requires ligand engagement and integrin clustering, which is promoted by Arp2/3-mediated polymerization of branched actin[7]. The initial connection of integrin clusters to the lamellipodial actin network by adaptors such as talin and vinculin in nascent adhesions is essential for membrane protrusion[8]. In addition, the subsequent adaptor-mediated linking of integrins to actomyosin bundles[9-11] in combination with myosin II-mediated tension is necessary for the maturation into FAs[12]. Conversely, FA disassembly is promoted by the loss of tension and controlled by a variety of pathways including microtubule-dependent[13,14]

[1]Department for Nanophysiology, RPTU University Kaiserslautern-Landau, Kaiserslautern, Germany. [2]Membrane Traffic and Cell Motility Group, Leibniz-Forschungsinstitut für Molekulare Pharmakologie, Berlin, Germany. [3]Screening Unit, Leibniz-Forschungsinstitut für Molekulare Pharmakologie, Berlin, Germany. [4]Max-Delbrück-Center for Molecular Medicine in the Helmholtz Association, Berlin, Germany. [5]Department for Molecular Pharmacology and Cell Biology, Leibniz-Forschungsinstitut für Molekulare Pharmakologie, Berlin, Germany. [6]Metabolic Networks, Max Planck Institute of Molecular Plant Physiology, Potsdam-Golm, Germany. [7]Cellular Imaging Facility, Leibniz-Forschungsinstitut für Molekulare Pharmakologie, Berlin, Germany. [8]Science for Life Laboratory, Department of Molecular Biosciences, The Wenner-Gren Institute, Stockholm University, Stockholm, Sweden. [9]Department of Microbiology, Tumor and Cell Biology, Karolinska Institutet, Solna, Sweden. [10]Faculty of Biology, Chemistry, Pharmacy, Freie Universität Berlin, Berlin, Germany. [11]NeuroCure Cluster of Excellence, Charité Universitätsmedizin Berlin, Berlin, Germany. ✉e-mail: maritzen@rptu.de

delivery of proteases for secretion[15], endocytosis[13,16–18] and autophagy of FA components[19,20].

However, which upstream factors regulate FA assembly and/or turnover has remained unclear. Nevertheless, it is evident that cellular metabolism is a critical determinant of FA assembly and maturation by supplying energy for the necessary cytoskeletal remodelling. A range of studies has revealed links between metabolic enzymes and the cytoskeleton[21–23], which adapt metabolic pathways to cellular needs. For example, the glycolytic enzyme phosphofructokinase (PFK) is stabilized upon force-mediated stress fibre formation to ensure a sufficient energy supply for cytoskeletal reorganization[24]. However, there is no evidence yet for metabolic pathways shaping in turn cytoskeletal and adhesion dynamics. Although it seems likely that the connection between metabolism, cytoskeletal dynamics and cellular adhesion goes beyond supplying the necessary energy for actin polymerization, it has remained speculative that metabolites may fine-tune the function of proteins involved in actin reorganization and cell-matrix adhesion[23,25].

Earlier screens for FA regulators had limited potential to uncover metabolic links since they were based on small preselected siRNA libraries comprising for example adhesion- and cytoskeleton-related genes or kinases and phosphatases[26,27]. To identify regulators of FA dynamics in a comprehensive and unbiased manner, we conducted a genome-wide loss-of-function screen using RNA interference (RNAi) coupled to automated fluorescence microscopy. Our results reveal a previously unknown layer of metabolic control of FA dynamics and cell protrusion, which depends on the glycolytic-flux signalling metabolite fructose-1,6-bisphosphate (FBP).

## Results

### Genome-wide screen identifies aldolase as a regulator of FAs

To identify regulators of FA dynamics in an unbiased manner, we conducted a genome-wide loss-of-function screen. We quantified the number of FAs per cell and the area of FAs in U-2 OS cells following small interfering RNA (siRNA)-mediated depletion of individual proteins—two parameters commonly altered downstream of changes in FA dynamics[18,20,28]. We monitored these FA parameters by immunolabelling U-2 OS cells for the FA marker paxillin[3] followed by automated confocal microscopy and image analysis (Fig. 1a). We screened 18,091 pools of four siRNAs for their impact on FA number and area using positive (combined depletion of the actin regulators cofilin and destrin), negative (paxillin depletion) and transfection controls (siTOX) for internal quality evaluation (Extended Data Fig. 1a–d). Our primary screen identified 280 candidate genes resulting in increased FA area or number per cell (Extended Data Fig. 1e,f and Supplementary Table Tab 1). Among these were genes encoding known regulators of FA dynamics including talin-1, α-actinin-4, DAB2, ATG4A and regulators of the actin cytoskeleton substantiating the validity of our approach (Supplementary Table Tab 1). We examined the identified candidates for their reproducibility (Fig. 1b) narrowing the candidate list down to 50 genes, many of which had not been associated with FA dynamics previously (Supplementary Table Tab 2).

Given that cellular metabolism shapes diverse cell processes and the impact of metabolic pathways on FA dynamics remains poorly understood, we focused our analysis on the *ALDOA* gene. This gene encodes the glycolytic enzyme fructose-1,6-bisphosphate aldolase A, which is associated with cancer progression[29–31]. While three aldolases with differential expression patterns exist in mammals (A, B and C), aldolase A is expressed ~140-fold higher than aldolase C and ~1,900-fold higher than aldolase B in U-2 OS cells (www.proteinatlas.org) explaining why only the knockdown of aldolase A (hereafter referred to as aldolase) elicited effects in our screen. Loss of aldolase in U-2 OS cells reproducibly increased FA abundance per cell (Fig. 1c,d). This phenotype was rescued by re-expression of an siRNA-resistant version of the wild-type (WT) enzyme (Fig. 1e,f) confirming that the phenotypes were not caused by siRNA off-target effects. Furthermore, aldolase depletion

led to an increase in cell area (Fig. 1g), a phenotype commonly associated with increased FA numbers[32,33].

### FBP levels control FA number and cell size

Aldolase is well known for its role in glycolysis, where it catalyses the cleavage of FBP into dihydroxyacetone phosphate and glyceraldehyde-3-phosphate, thereby controlling the availability of glycolytic intermediates for ATP production. In addition to its catalytic activity, aldolase has been reported to have several 'moonlighting' functions such as WASP and actin binding[34–36], regulation of endocytosis via SNX9[37] and modulation of AMPK signalling[38,39]. To investigate which of aldolase's functions affect FA dynamics, we created stable cell lines either expressing WT aldolase, the D33S mutant, which remains predominantly in an FBP-bound form[36], or the K229A mutant, which is unable to bind FBP[36], and depleted them of endogenous aldolase. WT aldolase but neither of the two catalytically inactive mutants was able to rescue elevated FA number and increased cell size upon loss of the endogenous enzyme suggesting that aldolase's enzymatic function is required for its rescue ability (Fig. 2a–c).

If the role of aldolase in ATP production is important for the regulation of FA number and cell size, depletion of other glycolytic enzymes should phenocopy loss of aldolase. To explore this, we depleted U-2 OS cells of either aldolase, PFK (via triple knockdown of the isoforms PFKL, PFKM and PFKP; or via silencing of PFKP via an independent siRNA (Extended Data Fig. 2a–d)), the enzyme upstream of aldolase, or glyceraldehyde-3-phosphate dehydrogenase (GAPDH), the enzyme downstream of aldolase (Fig. 2d,e). Surprisingly, we observed striking differences in the effects on FA numbers and cell size upon loss of these glycolytic enzymes (Fig. 2f–h). Loss of PFK resulted in a decrease of FA abundance and a concomitant reduction in cell size, that is, in a phenotype opposite to that of aldolase depletion, whereas GAPDH knockdown did not result in any adhesion phenotype (Fig. 2f–h). To ensure that these results were not impacted by a differential influence of the different knockdowns on cell viability, we monitored the silenced cells for compromised health using the nucleic acid dye SYTOX, which cannot cross the plasma membrane of healthy cells. This showed that none of the glycolytic enzyme knockdowns caused increased cell death (Extended Data Fig. 2e,f). As FA abundance also depends on cell cycle stage, with cells in the S phase having strongly increased adhesions[40], we analysed the cell cycle profile of the different knockdown conditions (Extended Data Fig. 2g). However, aldolase depletion reduced the fraction of cells in the S phase in line with published data[41], suggesting that the increased FA number of aldolase-knockdown cells is not due to altered cell cycle progression.

To rule out that the observed phenotypes are a peculiarity of U-2 OS cells, we verified our results in another cancerous as well as two non-malignant cell lines. Indeed, MDA-MB-231 breast cancer cells and non-malignant retinal pigment epithelium cells (hTERT RPE-1) as well as human foreskin fibroblasts (HFF) also showed a significant increase in cell area and FA number upon aldolase depletion but not upon PFK or GAPDH loss, demonstrating that the differential impact of glycolytic enzymes on adhesions is conserved across cell types (Extended Data Fig. 3).

In contrast to the divergent effects on FA number, knockdown of any of these glycolytic enzymes caused a similar reduction of cellular ATP levels (Extended Data Fig. 4a). These results led us to two unexpected conclusions: first, differences in FA number and cell size upon aldolase depletion are not a direct consequence of reduced ATP levels caused by a general defect in glycolysis but are due to changes in glycolytic intermediates. This idea was supported by the fact that cells depleted of ATP via inhibition of mitochondrial respiration by antimycin A were indistinguishable from controls with respect to FA numbers and cell size (Extended Data Fig. 4b–e). Consistently, we did not observe a correlation between ATP levels and FA number or cell area across various treatments (Fig. 2i,j). Second, the opposing FA phenotypes of PFK and aldolase knockdown indicate that metabolic control of the glycolytic intermediate FBP, whose levels are

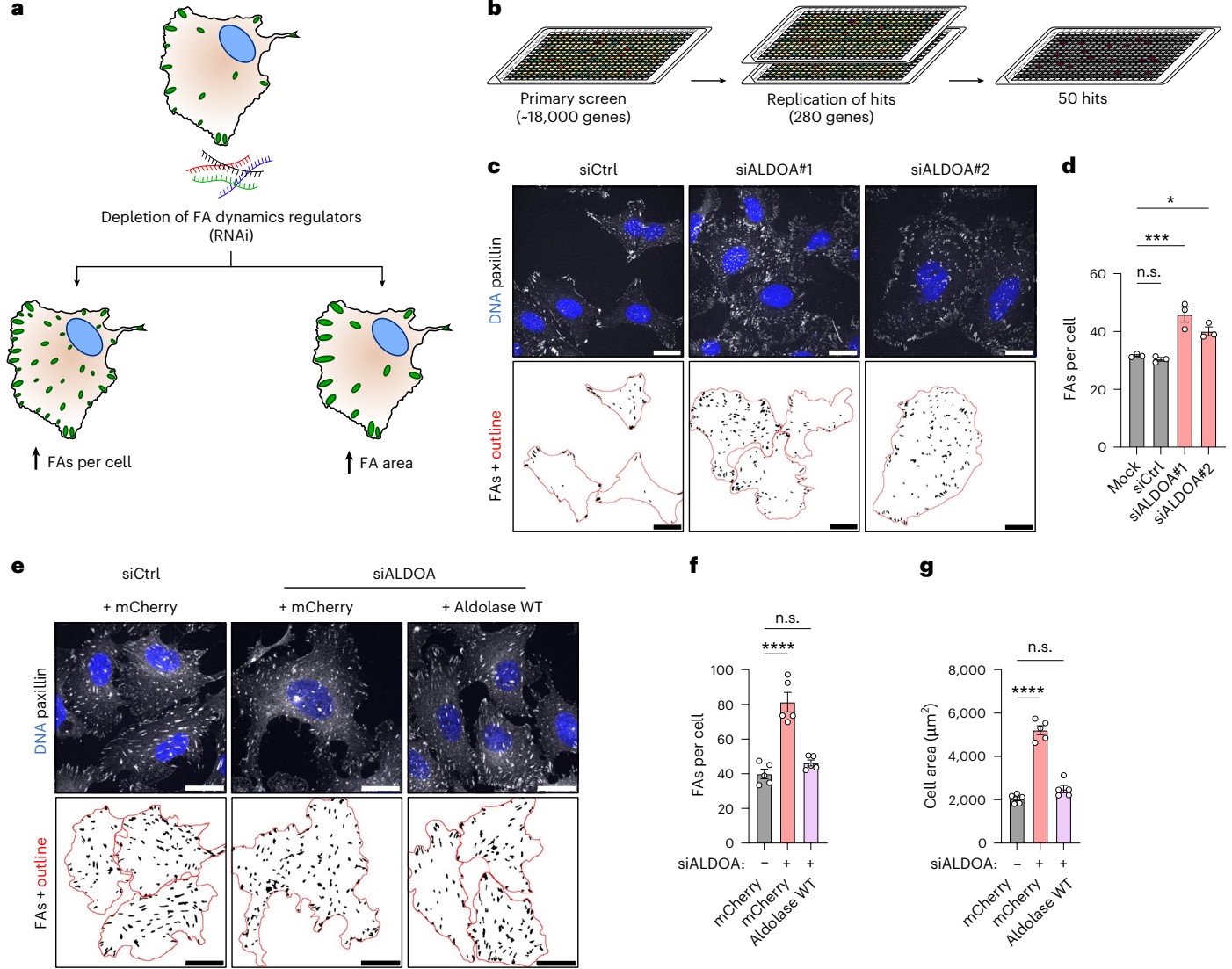

**Fig. 1 | Genome-wide siRNA screen identifies aldolase as a regulator of FA dynamics. a**, Rationale of the genome-wide screen: loss of FA dynamics regulators by siRNA-mediated knockdown results either in an increase in FAs per cell or in an increased FA area. The green dots represent FAs, and the blue circles represent nuclei. **b**, Overview of the screening procedure: initial candidates were identified by screening a genome-wide siRNA pool library in 384-well format. The resulting 280 candidates were rescreened two times to identify reproducible hits. **c,d**, Loss of aldolase causes an increase in FAs per cell. **c**, Representative confocal images of paxillin-labelled FAs in U-2 OS cells treated with control (siCtrl) or aldolase-specific (siALDOA) siRNAs. FA segmentation is shown below. For better visualization of the phenotype, manually drawn cell outlines were added for cells completely contained in images (red). **d**, A quantification of FAs per cell shown in **c**. The data represent mean ± s.e.m.; $n = 3$ independent experiments; one-way ANOVA with Dunnett's post-test ($P$ values: (mock versus siCtrl) = 0.8904; $P$ (mock versus siALDOA#1) = 0.0005; $P$ (mock versus siALDOA#2) = 0.0131). **e–g**, Aldolase re-expression rescues FA and cell size increase in silenced cells. **e**, Representative confocal images of paxillin-labelled FAs in U-2 OS cells stably expressing either HA-tagged mCherry as control or siRNA-resistant aldolase and treated with control (siCtrl) or aldolase-specific (siALDOA) siRNAs. FA segmentation and cell outlines (red) are shown below. **f,g**, A quantification of FAs per cell (**f**) and cell area (**g**) shown in **e**. The data represent mean ± s.e.m.; $n = 5$ independent experiments; one-way ANOVA with Dunnett's post-test ($P$ values (for **f**): (siCtrl + mCherry versus siALDOA + mCherry) <0.0001; (siCtrl + mCherry versus siALDOA + aldolase wt) = 0.4083; $P$ values (for **g**): (siCtrl + mCherry versus siALDOA + mCherry) <0.0001; (siCtrl + mCherry versus siALDOA + aldolase wt) = 0.133). n.s., not significant; *$P$ < 0.05, ***$P$ < 0.001, ****$P$ < 0.0001. Scale bars, 25 μm.

regulated by these enzymes, crucially affects FAs. Consistently, inhibiting glycolytic flux with 2-deoxy-D-glucose (2-DG), which cannot be metabolized and thus reduces FBP levels, phenocopied PFK loss (Extended Data Fig. 4f–i). Furthermore, metabolic measurements revealed that the FBP concentration is very high in aldolase-depleted cells (Fig. 3a), which have an increased FA number and cell size (Fig. 2f–h) while being very low in PFK-depleted cells (Fig. 3a), which exhibit a low FA number and cell area (Fig. 2f–h). In GAPDH-depleted cells (Fig. 3a), the FBP concentration was unaltered in line with the absence of changes in FA number and cell area (Fig. 2f–h). Hence, the FBP concentration strongly correlates with FA number and cell size and therefore might control cellular adhesion. To test this, we

blocked FBP synthesis in aldolase-deficient cells by simultaneously depleting PFK (Fig. 3b). Remarkably, codepletion of the two enzymes not only restored intracellular FBP levels (Fig. 3c) but also rescued the increased FA number and cell size (Fig. 3d–f). The same effect was observed when FBP levels were normalized by pharmacological treatment of aldolase-deficient cells with the upstream glycolytic inhibitor 2-DG (Fig. 3g,h and Extended Data Fig. 5), excluding the possibility that the phenotypic rescue arises from a non-catalytic function of PFK.

Collectively, these findings demonstrate that the glycolytic enzymes PFK and aldolase control FA dynamics by regulating the intracellular availability of FBP.

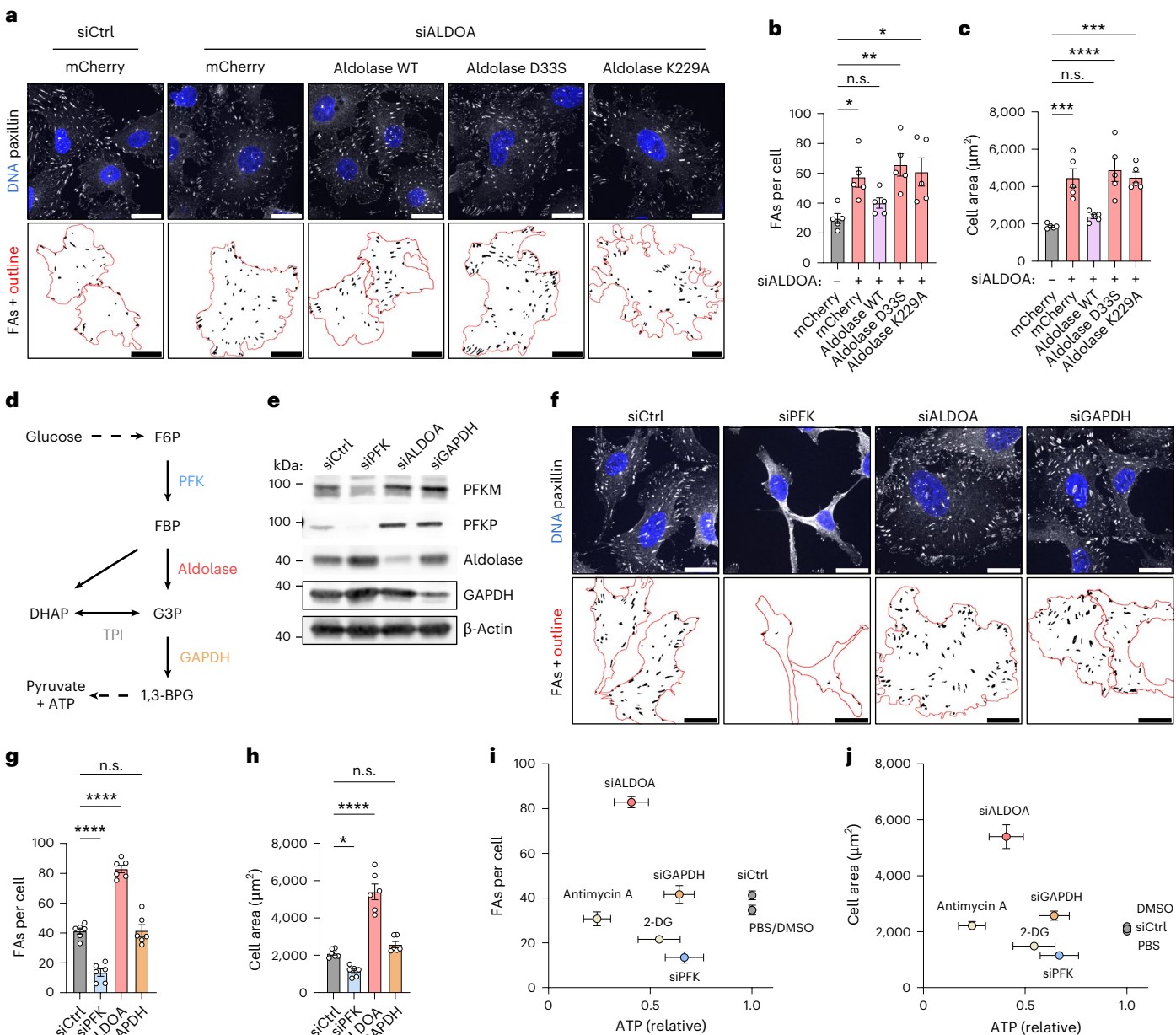

**Fig. 2 | Glycolytic enzymes control FA number and cell area. a–c**, The enzymatic activity of aldolase is essential to rescue phenotypes of aldolase knockdown cells. **a**, Representative confocal images of paxillin-labelled FAs in U-2 OS cells stably expressing HA-tagged mCherry or siRNA-resistant WT or catalytically inactive (D33S; K229A) aldolase and treated with indicated siRNAs. FA segmentation and cell outlines (red) are shown. **b,c**, A quantification of FAs per cell (**b**) and cell area (**c**) shown in **a**. The data represent mean ± s.e.m.; $n = 5$ independent experiments; one-way ANOVA with Dunnett's post-test (P values (for **b**): (siCtrl + mCherry versus siALDOA + mCherry) = 0.0221; (siCtrl + mCherry versus siALDOA + aldolase WT) = 0.6234; (siCtrl + mCherry versus siALDOA + aldolase D33S) = 0.0029; (siCtrl + mCherry versus siALDOA + aldolase K229A) = 0.0102; P values (for **c**): (siCtrl + mCherry versus siALDOA + mCherry) = 0.0005; (siCtrl + mCherry versus siALDOA + aldolase WT) = 0.737; (siCtrl + mCherry versus siALDOA + aldolase D33S) <0.0001; (siCtrl + mCherry versus siALDOA + aldolase K229A) = 0.0004). **d**, A simplified scheme of glycolysis highlighting aldolase and its neighbouring enzymes. The full arrows indicate a direct connection and the dashed arrows a substitute for missing steps. **e**, Efficient depletion of glycolytic enzymes. Immunoblot of U-2 OS cells

treated with indicated siRNAs. β-Actin was used as loading control. $N = 1$ independent experiment. **f–h**, Loss of PFK and aldolase affect FAs per cell and cell size in opposite manner, while GAPDH depletion has no effect. **f**, Representative confocal images of paxillin-labelled FAs in U-2 OS cells treated with indicated siRNAs. FA segmentation and cell outlines (red) are shown below. **g,h**, A quantification of FAs per cell (**g**) and cell area (**h**) shown in **f**. For **f–h**, the data represent mean ± s.e.m; $n = 6$ independent experiments; one-way ANOVA with Dunnett's post-test (P values (for **g**): (siCtrl versus siPFK) = <0.0001; (siCtrl versus siALDOA) <0.0001; (siCtrl versus siGAPDH) >0.9999; P values (for **h**): (siCtrl versus siPFK) = 0.0288; (siCtrl versus siALDOA) <0.0001; (siCtrl versus siGAPDH) = 0.3812). **i,j**, Decreased ATP levels are not the cause of increased FAs per cell and cell size. Relationship between ATP levels and FAs per cell (**i**) or cell area (**j**), for cells treated as indicated. The data represent mean ± s.e.m. $n = 3$–6 independent experiments. n.s., not significant; *$P < 0.05$, **$P < 0.01$, ***$P < 0.001$, ****$P < 0.0001$. Scale bars, 25 μm. F6P, fructose-6-phosphate; G3P, glyceraldehyde-3-phosphate; 1,3-BPG, 1,3-bisphosphoglycerate; GAPDH, glyceraldehyde-3-phosphate dehydrogenase; TPI, triose phosphate isomerase.

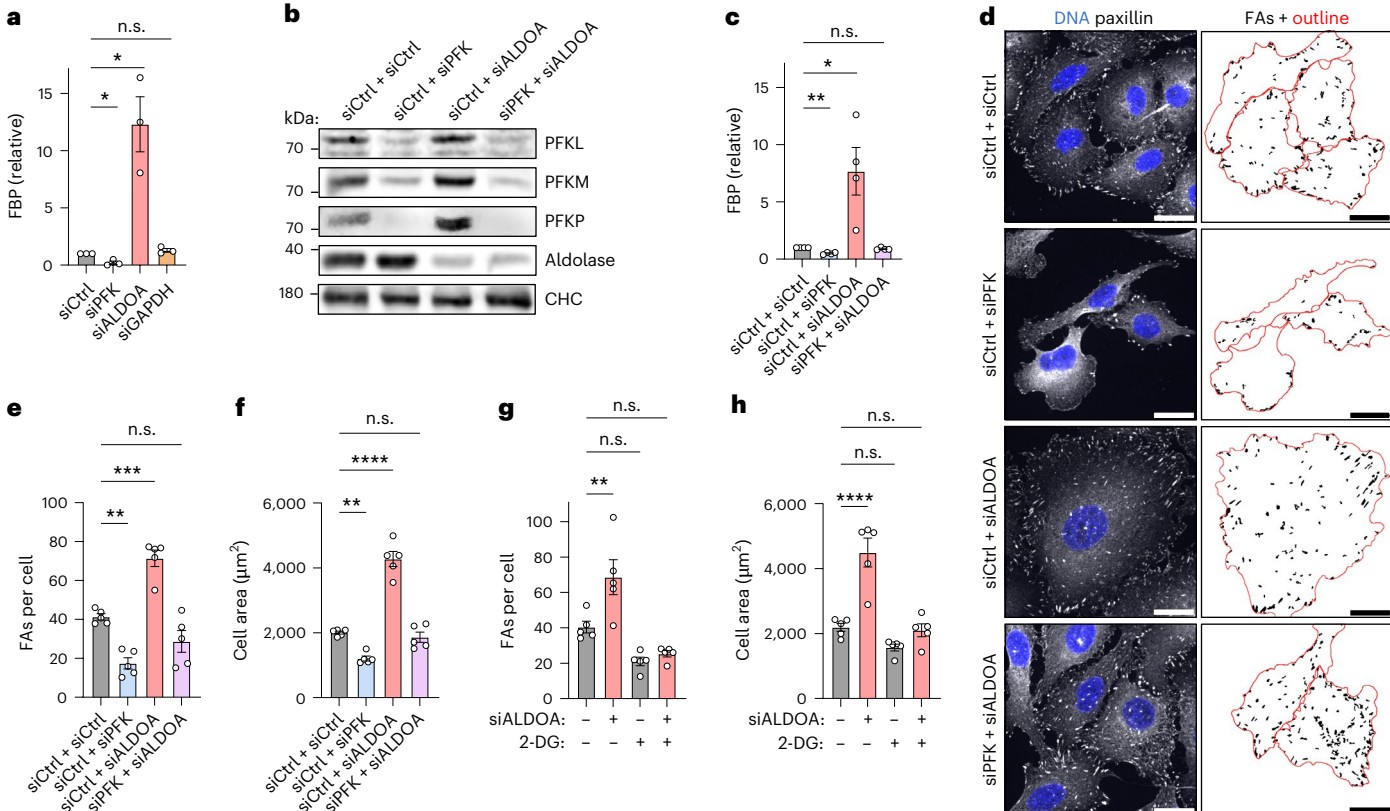

**Fig. 3 | Cell adhesion and cell area are controlled by the glycolytic metabolite FBP. a**, Loss of aldolase results in severely increased FBP levels, whereas PFK depletion lowers FBP. Relative FBP levels measured in U-2 OS cells treated with indicated siRNAs. Data were normalized to siCtrl and represent mean ± s.e.m.; n = 3 independent experiments; two-sided one-sample *t*-test (*P* values: (siPFK) = 0.0345; (siALDOA) = 0.0425; (siGAPDH) = 0.2188). **b**, Efficient codepletion of PFK and aldolase. Immunoblot of U-2 OS cells treated with indicated siRNAs. Clathrin heavy chain (CHC) was used as loading control; *N* = 1 independent experiment. **c**, Codepletion of PFK restores normal FBP levels in aldolase-knockdown cells. Relative FBP levels measured in U-2 OS cells treated with indicated siRNAs. Data were normalized to siCtrl and represent mean ± s.e.m.; *n* = 4 independent experiments; two-sided one-sample *t*-test (*P* values: (siCtrl + siPFK) = 0.0017; (siCtrl + siALDOA) = 0.0491; (siPFK + siALDOA) = 0.16). **d**–**f**, Co-depletion of PFK restores FA numbers and cell size in aldolase knockdown cells. **d**, Representative confocal images of paxillin-labelled FAs in U-2 OS cells treated with indicated siRNAs. FA segmentation and cell outlines (red) are shown on the right. **e**,**f**, A quantification of FAs per cell (**e**) and cell area (**f**) shown

in **d**. For **e**,**f**, data represent mean ± s.e.m.; *n* = 5 independent experiments; one-way ANOVA with Dunnett's post-test (*P* values (for **e**): (siCtrl + siCtrl versus siCtrl + siPFK) = 0.0015; (siCtrl + siCtrl versus siCtrl + siALDOA) = 0.0002; (siCtrl + siCtrl versus siPFK + siALDOA) = 0.1021; *P* values (for **f**): (siCtrl + siCtrl versus siCtrl + siPFK) = 0.0044; (siCtrl + siCtrl versus siCtrl + siALDOA) <0.0001; (siCtrl + siCtrl versus siPFK + siALDOA) = 0.8423). **g**,**h**, Inhibiting glycolytic flux rescues FAs per cell (**g**) and cell area (**h**) in aldolase-knockdown cells. A quantification of FAs per cell and cell area of U-2 OS cells treated with control (−) or ALDOA-specific siRNA (+) followed by 48 h treatment with PBS (−) or 25 mM 2-DG (+). Data represent mean ± s.e.m.; *n* = 5 independent experiments; one-way ANOVA with Dunnett's post-test (*P* values (for **g**): (siCtrl + PBS versus siALDOA + PBS) = 0.0056; (siCtrl + PBS versus siCtrl + 2-DG) = 0.0567; (siCtrl + PBS versus siALDOA + 2-DG) = 0.1605; *P* values (for **h**): (siCtrl+PBS versus siALDOA+PBS) <0.0001; (siCtrl + PBS versus siCtrl + 2-DG) = 0.2411; (siCtrl + PBS versus siALDOA + 2-DG) = 0.9888). Corresponding images are shown in Extended Data Fig. 5a. n.s., not significant; *P < 0.05, **P < 0.01, ***P < 0.001, ****P < 0.0001. Scale bars, 25 µm.

## FBP promotes de novo FA assembly, cytoskeletal reorganization and cell spreading

We wondered how FBP contributes to the observed changes, as an increased number of FAs can either result from increased FA assembly or reduced disassembly. To address this, we created U-2 OS cells stably expressing eGFP–paxillin and monitored FA dynamics using live-cell total internal reflection fluorescence (TIRF) microscopy (Supplementary Movies 1a,b and 2a,b). Detailed analyses of kinetic parameters showed that PFK depletion and, thus, FBP reduction decreased the rate of FA assembly and reduced the generation of FAs (Fig. 4a,b,d). By contrast, the rate of FA disassembly was unchanged (Fig. 4c), suggesting that FBP is specifically required for the formation of FAs. Consistently, we observed that elevated FBP levels in aldolase-depleted cells sufficed to increase FA assembly rate and FA formation (Fig. 4e,f,h). We also found a slight but significant increase in the disassembly rate of FAs under conditions of elevated cellular FBP (Fig. 4g), suggesting possible effects of FBP on FA maturation. Together, these results establish FBP as a positive regulator of FA formation.

FAs are physically and functionally linked to the actin cytoskeleton. Hence, we wondered whether FBP-mediated changes in FA dynamics correlate with altered cytoskeletal architecture. WT U-2 OS cells at steady state displayed parallel bundles of actin stress fibres, as expected (Fig. 5a). We observed a similar pattern in GAPDH-deficient cells indicating that ATP depletion does not affect the gross morphology of the actin cytoskeleton. By contrast, depletion of FBP elicited by PFK knockdown resulted in a strong reduction of visible stress fibres, whereas increased FBP concentrations in aldolase-depleted cells caused an overall reorganization of actin cytoskeletal architecture as evidenced by increased intersections of actin bundles and a larger area occupied by actin-rich lamellipodial protrusions (Fig. 5a). Therefore, we reasoned that the increased cell area caused by aldolase depletion might result from an elevated protrusive activity. To test this, we stained cells with fluorescent dyes labelling cytoplasm or plasma membrane and quantified the flat, protrusive peripheral parts of the cell using a custom-made image analysis pipeline (Methods; Extended Data Fig. 6a). Indeed, aldolase-deficient cells displayed an increased protrusive area compared to control cells

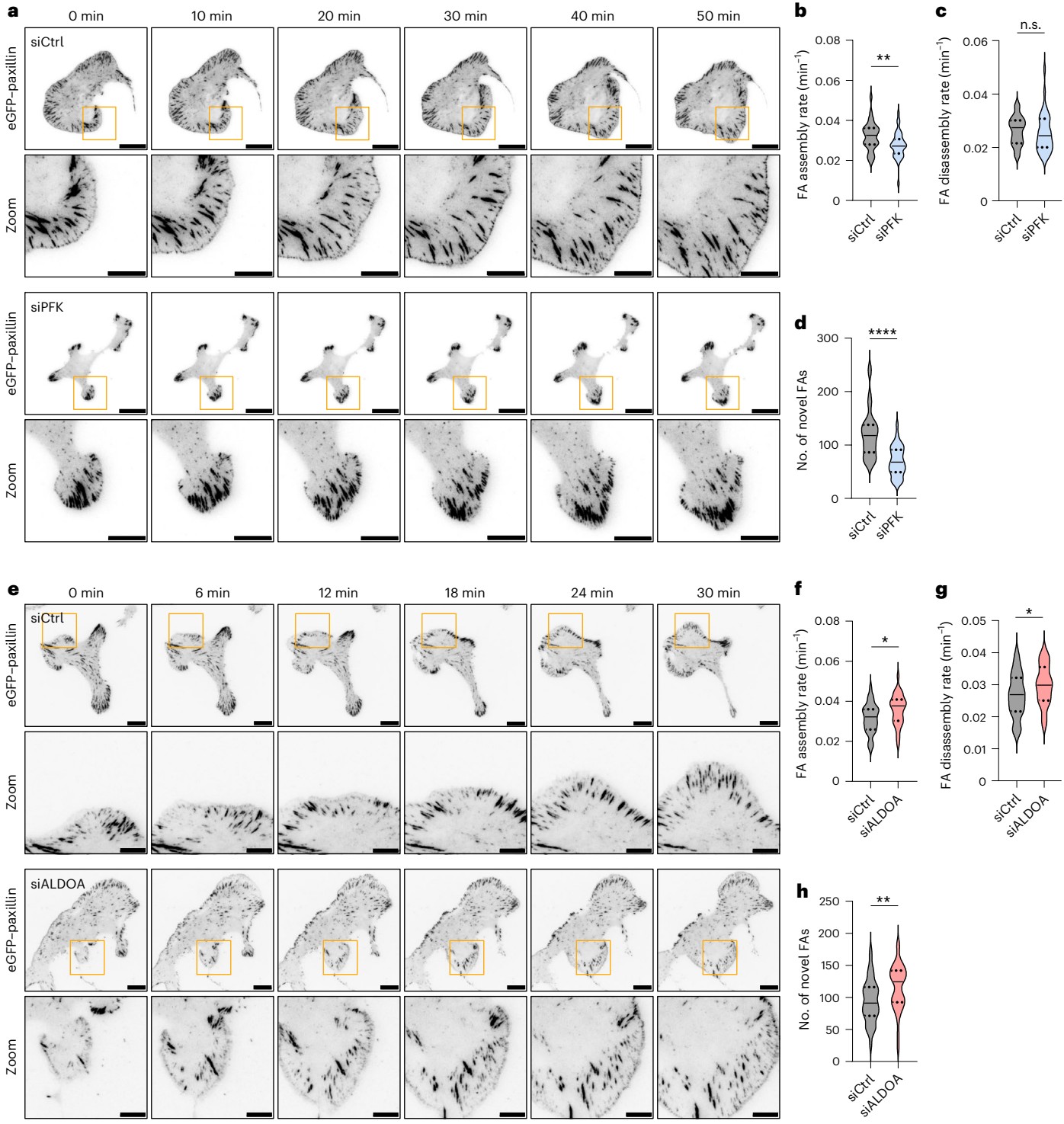

**Fig. 4 | FBP promotes FA assembly. a,e,** Reduced FBP levels associated with PFK depletion decrease FA assembly (**a**), whereas increased FBP levels upon aldolase knockdown increase the FA assembly rate (**e**). Representative TIRF microscopy time-lapse series of U-2 OS cells stably expressing eGFP–paxillin and treated with indicated siRNAs. Zoom-ins of the orange boxes are shown. **b,c,f,g,** A quantification of FA assembly rate (**b,f**) and disassembly rate (**c,g**). Data represent median and interquartile ranges. For **b** and **c**, n(siCtrl) = 34 and n(siPFK) = 39 cells from five independent experiments; for **f** and **g**, n(siCtrl) = 37 and n(siALDOA) = 35 cells from five independent experiments; two-sided Mann–Whitney test (P values (for **b**) = 0.0047; (for **c**) = 0.2941; (for **f**) = 0.0178; (for **g**) = 0.0492). **d,h,** A quantification of novel FAs formed over 4 h for siPFK (**d**) and siALDOA (**h**). Data represent median and interquartile ranges. For **d**, n(siCtrl) = 35 and n(siPFK) = 40 cells from five independent experiments; for **h**, n(siCtrl) = 37 and n(siALDOA) = 35 cells from five independent experiments; two-sided Mann-Whitney test (P values: (for **d**) <0.0001; (for **h**) = 0.0061). n.s., not significant; *P < 0.05, **P < 0.01, ****P < 0.0001. Scale bars, 25 µm and 10 µm for zoomed-in images.

or cells depleted of GAPDH or PFK (Fig. 5b and Extended Data Fig. 6b). In line with higher protrusive activity, aldolase depletion also resulted in accelerated cell spreading compared to control cells (Fig. 5c,d). Of importance, aldolase-deficient cells covered the same area at the initial

stages of spreading, suggesting that the observed increase in cell area is not a result of elevated cell volume or growth.

Collectively, our results demonstrate that accumulation of FBP elicits cellular reprogramming towards elevated FA-based adhesion by

reorganizing the F-actin cytoskeleton, thereby boosting FA assembly and cell spreading.

We wondered whether the large increase in FBP upon aldolase knockdown reflects a physiological scenario. To determine whether a less-pronounced elevation of FBP elicits a similar phenotype, we reduced the duration of our knockdown protocol to one round of siRNA transfection. Although this still led to a substantial decrease in aldolase A (Fig. 5e), the increase in FBP was less than half that detected upon two rounds of siRNA transfection (Fig. 5f, 2.6 nmol FBP per mg protein compared with 5.7 nmol FBP per mg protein). Nevertheless, we still observed a significant elevation of cell area under these conditions (Fig. 5g,h).

To explore whether similar FBP concentrations might occur in a physiological context, we evaluated FBP levels in U-2 OS cells under cellular conditions in which we expect FBP levels to be elevated, for example during cell spreading upon reseeding or during cell migration. As control condition, we measured FBP in stationary cells within a confluent cell layer where protrusive activity and FA assembly are intrinsically limited. Confluent cells contained 0.6 nmol FBP per mg protein, whereas migrating cells, which need localized protrusions for motility, displayed threefold higher FBP levels, that is, 1.8 nmol FBP per mg protein. Even more strikingly, we detected 5.4 nmol FBP per mg protein in spreading cells, which strongly protrude in all directions to enlarge their adhesive cell area (Fig. 5i). Importantly, the FBP levels quantified in migrating and spreading cells closely resemble the FBP levels measured upon mild (single knockdown) or near complete (double knockdown) aldolase depletion (Fig. 5f), respectively. These results demonstrate that FBP levels are dramatically increased under physiological conditions that require FA assembly and cell protrusion supporting the idea that FBP is a driver of these processes.

### FBP-mediated changes in FA dynamics and cellular protrusion depend on Rac1 activity

To elucidate the molecular mechanism by which FBP achieves this cellular reprogramming, we sought to identify FBP-binding proteins on a proteome-wide scale using limited proteolysis coupled to mass spectrometry (LiP–MS). This technique allows for the detection of subtle conformational changes of proteins within whole cell lysates upon the addition of small molecules[25,42]. Using U-2 OS cell lysates, we identified proteins responding to increasing concentrations of FBP (PRIDE repository identifier PXD043853; ref. 43) in the range found in mammalian cells[44]. Among the potential candidates, the small GTPase Rac1 stood out as a key regulator of actin dynamics, cell protrusion and

cell adhesion[45] and, consistently, exhibited a dose-dependent response to FBP (Fig. 6a). Mapping the responding peptide of Rac1 into its crystal structure revealed that conformational changes occurred in the P-loop (Fig. 6b), which coordinates phosphate-binding of the guanidine nucleotides[46]. This suggests that the GDP/GTP binding status of Rac1 and, thus, its activity are influenced by intracellular FBP concentrations. We probed this by performing GST-pull-down assays with the purified PBD domain of the Rac1 effector protein PAK, which specifically binds active Rac1-GTP. Elevated levels of active Rac1-GTP were captured from lysates of aldolase-deficient cells compared with controls, suggesting that FBP increases Rac1 activity (Fig. 6c). Conversely, active Rac1-GTP levels were reduced in PFK-deficient cells (Extended Data Fig. 7a). To test if Rac1 hyperactivity was responsible for the FBP-mediated changes in FA dynamics, we codepleted aldolase-deficient cells of Rac1 (Fig. 6d). Loss of Rac1 did not significantly affect the intracellular concentration of FBP (Fig. 6e). However, it resulted in a striking decrease in FA abundance and cell area (Fig. 6f–h), comparable to the loss of PFK (Fig. 2f–h). Moreover, Rac1 depletion in aldolase-deficient cells rescued the increased FA number not only in U-2 OS cells but also in hTERT RPE-1 cells (Extended Data Fig. 7b–e). Overexpression of dominant-negative mutant Rac1-T17N also rescued elevated FA numbers in aldolase-depleted cells (Fig. 6i,j and Extended Data Fig. 7f). Conversely, ectopic expression of constitutively active Rac1-Q61L led to significantly elevated FA numbers in control- or PFK-deficient cells (Fig. 6k,l and Extended Data Fig. 7g). We conclude that Rac1 activity is required for FBP regulation of FA assembly.

Because of its prominent role in actin organization, we wondered if Rac1 was also responsible for the FBP-mediated changes in cytoskeletal architecture. Indeed, depletion of Rac1 alone decreased actin stress fibres similarly to the depletion of PFK (Figs. 5a and 7a). Moreover, codepletion of Rac1 rescued the aberrant actin architecture of aldolase-depleted cells (Fig. 7a) suggesting that Rac1 is required for this phenotype. Similar codepletion experiments revealed that Rac1 was also necessary for the enhanced protrusion and accelerated spreading of cells suffering from elevated FBP levels in absence of aldolase (Fig. 7a–d and Extended Data Fig. 8).

Lastly, we probed whether Rac1 activation and Rac1-dependent cell spreading are not only promoted by the high levels of FBP accumulating upon aldolase silencing but can also occur under physiological conditions, that is, downstream of cellular glucose exposure. For this, we decreased basal Rac1 activity in the non-cancerous RPE-1 cell line by glucose starvation before testing cell spreading under a glucose

---

**Fig. 5 | Elevated FBP levels promote F-actin reorganization and cell protrusion.**
**a,b**, Increased FBP levels upon aldolase knockdown (KD) alter the organization of the actin cytoskeleton and elevate cellular protrusion. **a**, Representative confocal images of the phalloidin-labelled F-actin cytoskeleton in U-2 OS cells treated with indicated siRNAs. Zoom-ins of the orange boxes are shown. **b**, A quantification of the protrusive area of U-2 OS cells treated with indicated siRNAs. For **a** and **b**, data represent mean ± s.e.m.; $n = 5$ independent experiments; one-way ANOVA with Dunnett's post-test ($P$ values: (siCtrl versus siPFK) = 0.5203; (siCtrl versus siALDOA) <0.0001; (siCtrl versus siGAPDH) = 0.5458). Corresponding images are shown in Extended Data Fig. 6b. **c,d**, Aldolase depletion results in elevated cell spreading. **c**, Representative confocal images of phalloidin-labelled U-2 OS cells treated with indicated siRNAs. The cells were seeded and fixed after indicated time points. **d**, A quantification of cell spreading shown in **c**. For **c** and **d**, data represent mean ± s.e.m.; $n = 5$ independent experiments; one-way ANOVA with Dunnett's post-test ($P$ values: (siCtrl versus siPFK) = 0.2822; (siCtrl versus siALDOA) <0.0001; (siCtrl versus siGAPDH) = 0.9188). **e–h**, Less pronounced elevations in FBP levels also increase cell size. **e**, Efficient depletion of aldolase and GAPDH. Immunoblot of U-2 OS cells treated 1× or 2× with indicated siRNAs. Clathrin heavy chain (CHC) was used as loading control. The cropped lanes are from the same blot; $N = 1$ independent experiment. **f**, LC–MS/MS-based FBP measurements of U-2 OS cells treated 1× or 2× with indicated siRNAs. Data represent mean ± s.e.m.; $n = 3$ independent experiments; one-way ANOVA with Tukey's post-test ($P$ values: (1xkd-siCtrl versus

1xkd-siALDOA) = 0.0186; (1xkd-siCtrl versus 2xkd-siCtrl) >0.9999; (1xkd-siCtrl versus 2xkd-siALDOA) <0.0001; (1xkd-siCtrl versus 2xkd-siGAPDH) >0.9999; (1xkd-siALDOA versus 2xkd-siCtrl) = 0.018; (1xkd-siALDOA versus 2xkd-siALDOA) = 0.0036; (1xkd-siALDOA versus 2xkd-siGAPDH) = 0.0209; (2xkd-siCtrl versus 2xkd-siALDOA) <0.0001; (2xkd-siCtrl versus 2xkd-siGAPDH) >0.9999; (2xkd-siALDOA versus 2xkd-siGAPDH) <0.0001). **g**, A quantification of cell area shown in **h**. Data represent mean ± s.e.m.; $n = 3$ independent experiments; repeated-measures one-way ANOVA with Tukey's post-test ($P$ values: (1xkd-siCtrl versus 1xkd-siALDOA) = 0.0063; (1xkd-siCtrl versus 2xkd-siCtrl) = 0.2089; (1xkd-siCtrl versus 2xkd-siALDOA) = 0.0161; (1xkd-siCtrl versus 2xkd-siGAPDH) = 0.0105; (1xkd-siALDOA versus 2xkd-siCtrl) = 0.0018; (1xkd-siALDOA versus 2xkd-siALDOA) = 0.0267; (1xkd-siALDOA versus 2xkd-siGAPDH) = 0.0313; (2xkd-siCtrl versus 2xkd-siALDOA) = 0.0137; (2xkd-siCtrl versus 2xkd-siGAPDH) = 0.0172; (2xkd-siALDOA versus 2xkd-siGAPDH) = 0.0242). **h**, Representative confocal images of paxillin-labelled FAs in U-2 OS cells treated 1× or 2× with indicated siRNAs. FA segmentation and cell outlines (red) are shown below. **i**, Protruding cells display strongly elevated FBP levels. LC–MS/MS-based FBP measurements in confluent (24 h after dense plating), migrating (6 h after sparse plating) or spreading (1 h of sparse plating) U-2 OS cells. Data represent mean ± s.e.m.; $n = 3$ independent experiments; one-way ANOVA with Tukey's post-test ($P$ values: (immobile versus migrating) = 0.0046; (immobile versus spreading) <0.0001; (migrating versus spreading) <0.0001). n.s., not significant; *$P < 0.05$, **$P < 0.01$, ****$P < 0.0001$. Scale bars, 25 µm.

pulse. To ensure a sufficient energy supply, cells were kept in the presence of sodium pyruvate and the L-glutamine substitute GlutaMAX. In absence of glucose, cells spread very little in line with low Rac1 activity (Fig. 7e–h). Strikingly, the addition of 1 mM glucose resulted in an

increase in Rac1 activity as demonstrated by elevated GTP-Rac1 levels and efficient cell spreading (Fig. 7e–h). Collectively, these results demonstrate that FBP-mediated cell reprogramming towards increased adhesion is a consequence of stimulated Rac1 activity.

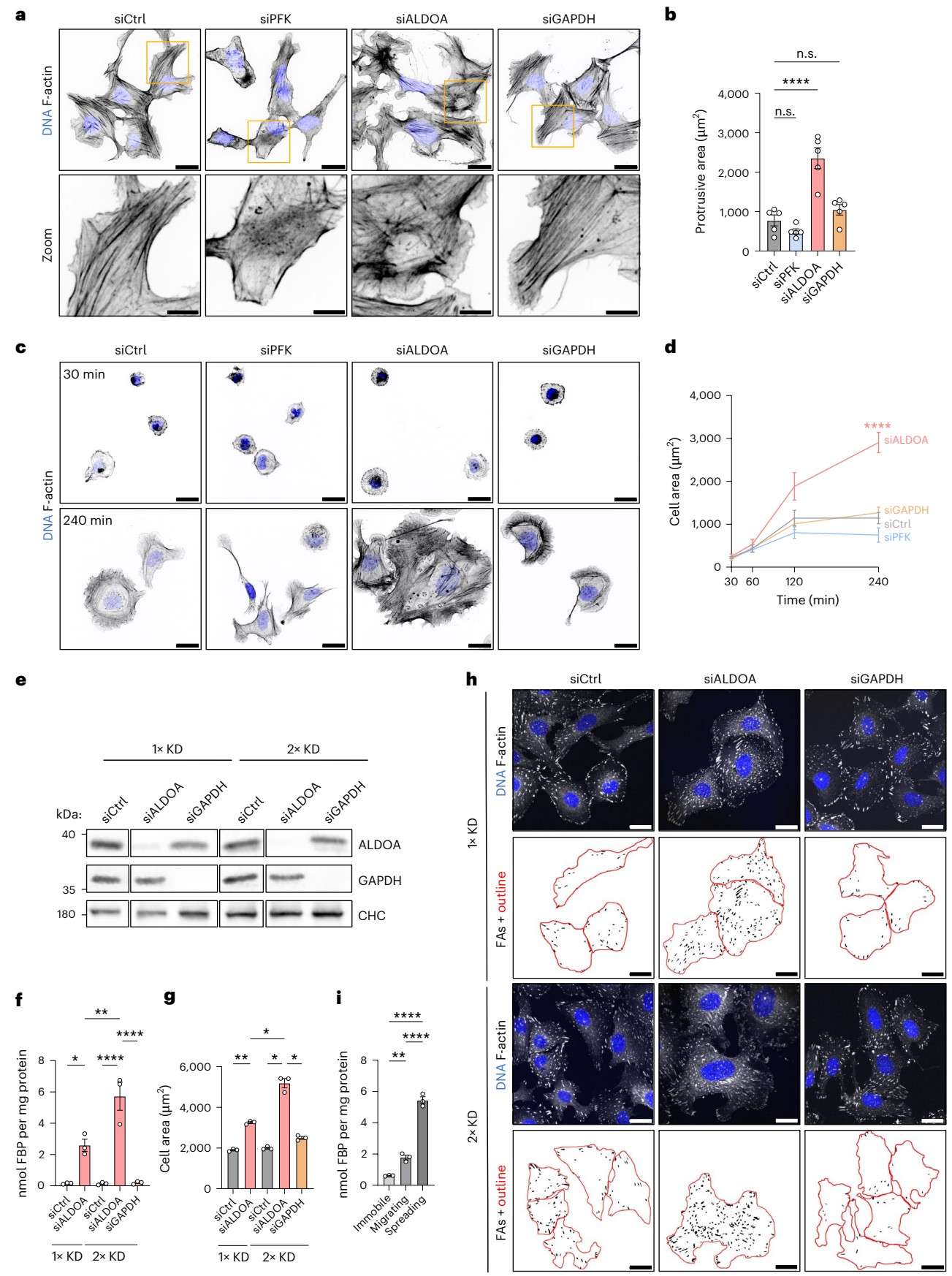

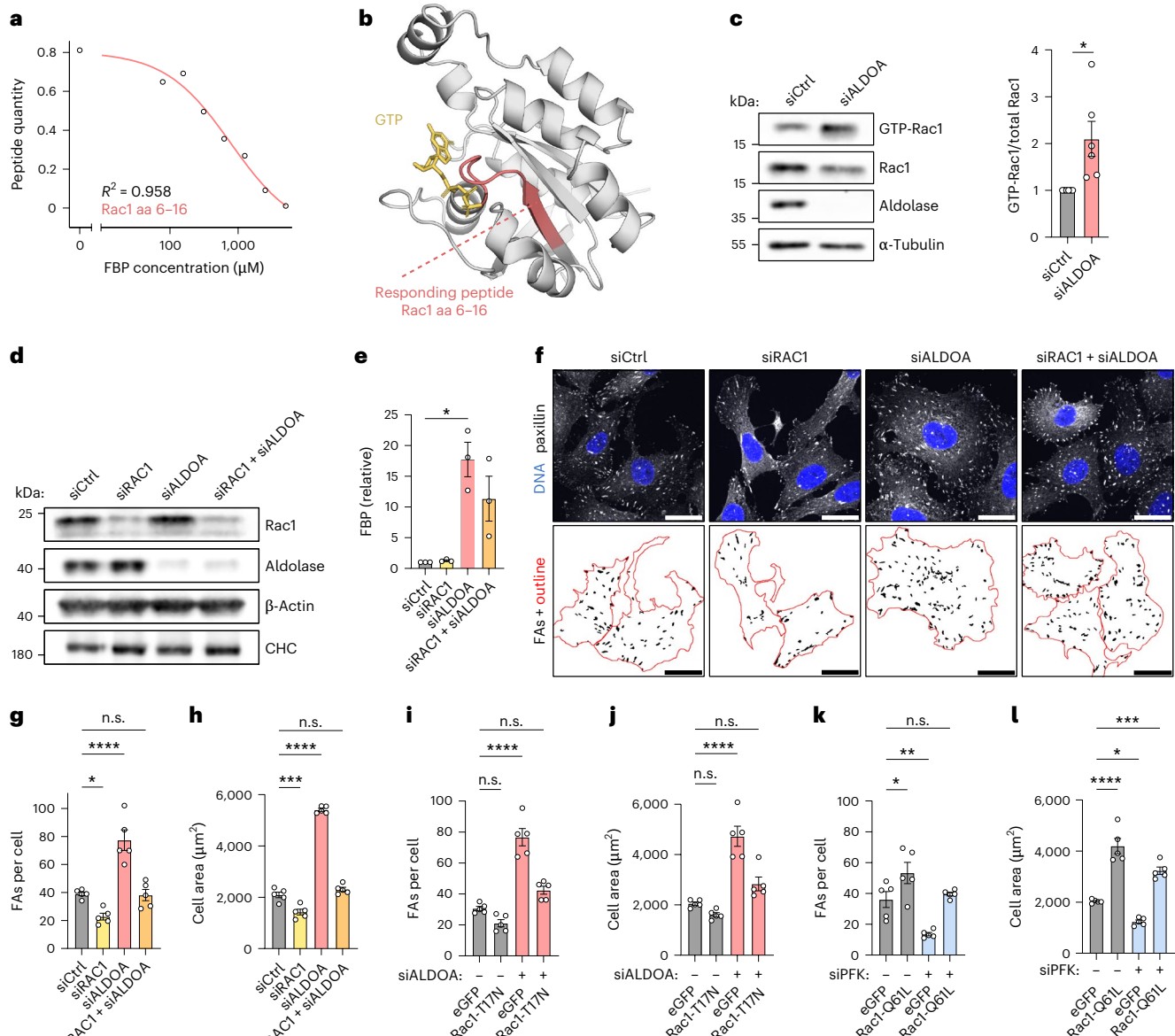

**Fig. 6 | FBP stimulates Rac1 activation leading to FA assembly and cell spreading. a,b,** Addition of FBP to whole cell lysate causes a conformational change in Rac1. **a,** Whole cell lysates of U-2 OS cells were subjected to LiP–MS. Peptide intensity (amino acids (aa) 6–16) of Rac1 in response to increasing FBP concentrations. **b,** Responding peptide (red, aa 6–16) highlighted in the GTP-bound structure of Rac1 (PDB ID: 1MH1). **c,** Increased Rac1 activity upon loss of aldolase. Left: representative immunoblot of pull-down assay using GST-PAK-PBD beads to detect active Rac1 in lysates from siRNA treated U-2 OS cells. α-Tubulin was used as a loading control. Right: a quantification of Rac1-GTP over total Rac1. Data were normalized to siCtrl and represent mean ± s.e.m.; $n$ = 5 independent experiments; two-sided one-sample $t$-test ($P$ value = 0.0326). **d,** Efficient Rac1 depletion. Representative immunoblot of U-2 OS cells treated with the indicated siRNAs. β-actin and clathrin heavy chain (CHC) were used as loading controls; $N$ = 3 independent experiments. **e,** Codepletion of Rac1 does not significantly lower FBP levels in aldolase-knockdown cells. Relative FBP levels measured in U-2 OS cells treated with indicated siRNAs. Data were normalized to siCtrl and represent mean ± s.e.m.; $n$ = 3 independent experiments; two-sided one-sample $t$-test ($P$ values: (siRAC1) = 0.1494; (siALDOA) = 0.027; (siRAC1 + siALDOA) = 0.105). **f–h,** Codepletion of Rac1 restores FA numbers and cell size in aldolase knockdown cells. **f,** Representative confocal images of paxillin-labelled FAs in U-2 OS cells treated with indicated siRNAs. FA segmentation and cell outlines (red) are shown below. **g,h,** A quantification of FAs per cell (**f**) and cell area (**g**) shown in **f**. For **f–h,** data represent mean ± s.e.m.; $n$ = 5 independent experiments; one-way ANOVA with Dunnett's post-test ($P$ values (for **g**): (siCtrl versus siRAC1) = 0.0481;

(siCtrl versus siALDOA) <0.0001; (siCtrl versus siRAC1 + siALDOA) = 0.9979; $P$ values (for **h**): (siCtrl versus siRAC1) = 0.0003; (siCtrl versus siALDOA) <0.0001; (siCtrl versus siRAC1 + siALDOA) = 0.2799). **i,j,** Expression of dominant-negative Rac1 in aldolase-depleted cells rescues increased FA numbers (**i**) and cell size (**j**). A quantification of FAs per cell and cell area of U-2 OS cells transfected with control (−) or ALDOA-specific (+) siRNAs in combination with either eGFP or myc-Rac1-T17N. Corresponding images are shown in Extended Data Fig. 7f. Data represent mean ± s.e.m.; $n$ = 5 independent experiments; one-way ANOVA with Dunnett's post-test ($P$ values (for **i**): (siCtrl + GFP versus siCtrl + T17N) = 0.1497; (siCtrl + GFP versus siALDOA + GFP) <0.0001; (siCtrl + GFP versus siALDOA + T17N) = 0.0633; $P$ values (for **j**): (siCtrl + GFP versus siCtrl + T17N) = 0.4683; (siCtrl + GFP versus siALDOA + GFP) <0.0001; (siCtrl + GFP versus siALDOA + T17N) = 0.0981). **k,l,** Expression of constitutively active Rac1 in PFK-depleted cells rescues reduced FA numbers and cell size. Quantifications of FAs per cell (**k**) and cell area (**l**) of U-2 OS cells transfected with control (−) or PFK-specific (+) siRNAs in combination with either eGFP or myc-Rac1-Q61L. Corresponding images are shown in Extended Data Fig. 7g. Data represent mean ± s.e.m.; $n$ = 5 independent experiments; one-way ANOVA with Dunnett's post-test ($P$ values (for **k**): (siCtrl + GFP versus siCtrl + Rac1-Q61L) = 0.0342; (siCtrl + GFP versus siPFK + GFP) = 0.0063; (siCtrl + GFP versus siPFK + Rac1-Q61L) = 0.8904; $P$ values (for **l**): (siCtrl + GFP versus siCtrl + Rac1-Q61L) <0.0001; (siCtrl + GFP versus siPFK + GFP) = 0.0116; (siCtrl + GFP versus siPFK + Rac1-Q61L) = 0.0004). n.s., not significant; *$P$ < 0.05, **$P$ < 0.01, ***$P$ < 0.001, ****$P$ < 0.0001. Scale bars, 25 µm.

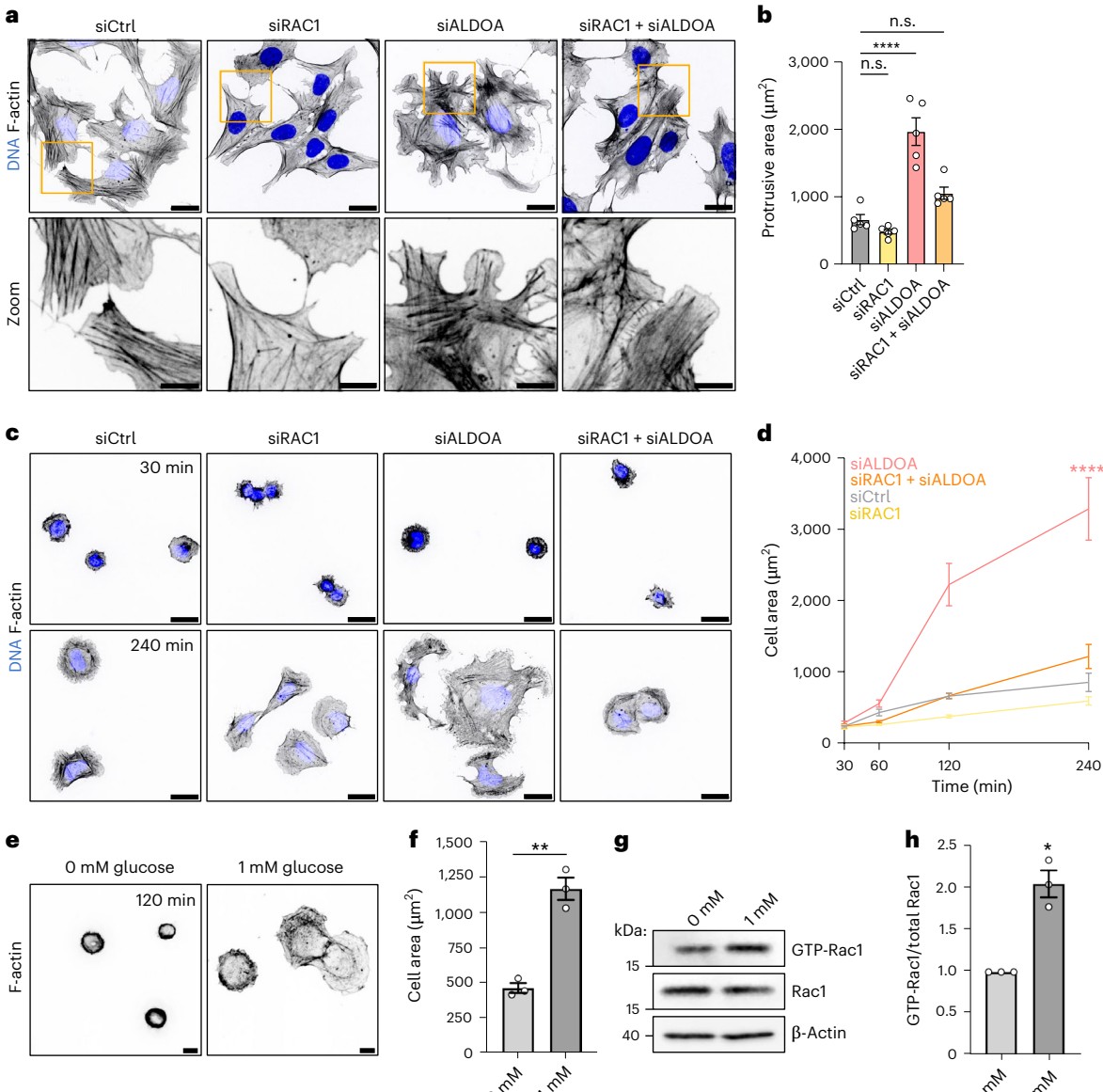

**Fig. 7 | Rac1 is required for FBP-mediated F-actin reorganization and cell protrusion. a,b**, Codepletion of Rac1 in aldolase-treated cells normalizes F-actin organization and cell protrusion. **a**, Representative confocal images of the phalloidin-labelled F-actin cytoskeleton in U-2 OS cells treated with indicated siRNAs. Zoom-ins of orange boxes are shown. **b**, A quantification of the protrusive area of U-2 OS cells treated with indicated siRNAs. Data represent mean ± s.e.m.; $n = 5$ independent experiments; one-way ANOVA with Dunnett's post-test ($P$ values: (siCtrl versus siRAC1) = 0.5925; (siCtrl versus siALDOA) <0.0001; (siCtrl versus siRAC1 + siALDOA) = 0.0825). Corresponding images are shown in Extended Data Fig. 8. **c,d**, Codepletion of Rac1 in aldolase-depleted cells restores cell spreading to normal levels. **c**, Representative confocal images of phalloidin-labelled U-2 OS cells treated with indicated siRNAs. The cells were seeded and fixed after indicated time points. **d**, A quantification of cell spreading shown in **c**. Data represent mean ± s.e.m.; $n = 5$ independent experiments; one-way ANOVA

with Dunnett's post-test ($P$ values: (siCtrl versus siRAC1) = 0.7944; (siCtrl versus siALDOA) <0.0001; (siCtrl versus siRAC1 + siALDOA) = 0.6024). **e,f**, Glucose triggers cell spreading. **e**, Representative images of phalloidin-labelled RPE-1 cells incubated with the indicated glucose concentrations. The cells were seeded and fixed after 2 h. **f**, A quantification of cell spreading shown in **e**. Data represent mean ± s.e.m.; $n = 3$ independent experiments; two-sided unpaired $t$-test ($P$ value = 0.0012). **g,h**, Increased Rac1 activity upon glucose. **g**, Representative immunoblot of a pull-down assay using GST-PAK-PBD beads to detect active GTP-Rac1 in lysates from untreated or glucose exposed RPE-1 cells. β-actin was used as a loading control. **h**, A quantification of GTP-Rac1 over total Rac1. The data were normalized to 0 mM glucose condition and represent mean ± s.e.m.; $n = 3$ independent experiments; two-sided one-sample $t$-test ($P$ value = 0.0228). n.s., not significant; *$P < 0.05$, **$P < 0.01$, ****$P < 0.0001$. Scale bars, 25 μm for **a** and **c**; 10 μm for **e**.

## FBP-mediated Rac1 activation depends on RCC2

FBP might bind to Rac1 directly to alter its activity or act indirectly via other regulatory factors that control Rac1 activity. We tested whether Rac1 activity is directly modulated by FBP via LiP–MS experiments using purified Rac1 in the presence of increasing concentrations of FBP. However, we detected no peptide responses in this setting. Although this does not rule out a direct binding of FBP to endogenous Rac1, it might suggest that FBP-mediated Rac1 activation is

controlled indirectly by other factors present in the cell lysate. To identify such factors, we screened the previously obtained list of putative FBP interactors from U-2 OS lysate for proteins linked to Rac1 activation. This analysis identified the Rac1 interactor RCC2, a metastasis suppressor previously shown to regulate Rac1 activity[47–49]. RCC2 displayed an FBP response (Fig. 8a) similar to that of Rac1 (Fig. 6a), and the FBP-mediated conformational changes in RCC2 localized to a loop facing the presumed interface for Rac1 interaction (Fig. 8b and

Extended Data Fig. 9a). Deletion of this loop was previously shown to hamper Rac1 binding[48]. Consistently, LiP–MS measurements of purified eGFP–RCC2 revealed an FBP-mediated conformational change within RCC2 (Fig. 8c and Extended Data Fig. 9b). These data identify RCC2 as a direct FBP binder. We also tested by LiP–MS whether the closely related molecule fructose-2,6-bisphosphate (F2,6BP), which is produced by PFK2 and serves as a regulator of PFK1 and fructose-1,6-bisphosphatase, can bind to RCC2. This analysis identified a single responding RCC2 peptide. Hence, an interaction cannot be excluded (Extended Data Fig. 9c). However, as F2,6BP levels are typically about one to two orders of magnitude lower than FBP levels, the contribution of F2,6BP to the total fructose bisphosphate pool is probably negligible[44,50].

Previous studies have yielded conflicting results as to whether RCC2 activates[47] or inhibits[48,49,51] Rac1 activity. To solve this, we silenced RCC2 in U-2 OS cells via siRNA (Fig. 8d). This led to an increase in FA abundance and elevated cell size comparable to the effects of aldolase knockdown (Fig. 8e–g). These data suggest an inhibitory function of RCC2 towards Rac1 activity. We confirmed this by codepletion of Rac1 and RCC2 in U-2 OS cells. Concomitant loss of Rac1 rescued the elevated FA number and increased cell size elicited by loss of RCC2 to levels observed in control cells (Fig. 8h–j and Extended Data Fig. 9d). We reasoned that FBP might activate Rac1 by interfering with the formation of an inhibitory RCC2–Rac1 complex. To test this, we overexpressed eGFP–RCC2 in cells suffering from elevated FBP levels. Strikingly, RCC2 overexpression suppressed the FBP-mediated FA increase and elevated cell size (Fig. 8k,l). Lastly, we probed whether FBP interferes with RCC2–Rac1 complex formation in a biochemical assay. For this, we added GST–Rac1 to eGFP–RCC2 immobilized on eGFP-trap beads. Indeed, the addition of FBP significantly reduced complex formation between RCC2 and Rac1 (Fig. 8m,n).

Accordingly, we propose a model in which FBP acts as signalling metabolite that elicits increased FA-based adhesion and cellular spreading. Molecularly, FBP appears to achieve this by disturbing an inhibitory RCC2–Rac1 complex resulting in the release and subsequent activation of Rac1. The ensuing Rac1 activity drives reorganization of the actin cytoskeleton, FA assembly and cell protrusion. Under conditions of low glycolytic activity, FBP is scarce and the RCC2–Rac1 complex remains intact, keeping Rac1 activation at a low level. This leads to a less developed F-actin cytoskeleton, decreased FA-based adhesion and smaller cell area (Extended Data Fig. 9e).

## Discussion

Here, we identify a signalling pathway that couples the metabolic cell state to cell adhesion and protrusion by capitalizing on the glycolytic intermediate FBP. We show that FBP acts as a signalling metabolite by tuning the activity of Rac1, a key factor in actin remodelling. As an upstream regulator of the actin nucleator Arp2/3, Rac1 promotes the formation of lamellipodia, which are essential for the initiation of nascent adhesions[45] and thus FAs. To allow for localized protrusions, Rac1 activity is regulated by RCC2[48,49,51,52]. RCC2–Rac1 complex formation prevents guanine nucleotide exchange factor (GEF)-mediated activation of Rac1, attenuating its activity outside protrusions[49]. However, how RCC2-mediated inhibition of Rac1 is regulated remained elusive. We demonstrate that FBP targets the RCC2–Rac1 complex, releasing Rac1 to be activated by GEFs and to promote actin polymerization, membrane protrusion and FA assembly. Thereby, a favourable metabolic state is directly translated into increased cell adhesion and protrusion.

FBP is produced by the PFK-catalysed rate-limiting commitment step of glycolysis and thus an ideal candidate to reflect the metabolic cell state. In fact, FBP levels correlate with glycolytic flux[53,54], and FBP is known to act as flux-signalling metabolite in multiple species ranging from bacteria to mammalian cells, transducing metabolic information into diverse cellular responses[55–57]. Metabolite sensing is indeed crucial for the coordination of cellular processes as exemplified by the importance of the two master regulators, mTORC1, a sensor of amino acid abundance, and AMPK, a sensor of ATP availability[58]. Interestingly, FBP does not only stimulate Rac1 activity but also negatively regulates AMPK by preventing aldolase from promoting the formation of a lysosomal complex containing and activating AMPK[39]. Thus, FBP is ideally positioned to coordinate the regulation of cell adhesion with the metabolic control exerted by AMPK and to reinforce Rac1-dependent cell protrusion by keeping AMPK, a negative regulator of integrin activity and cell spreading, inactive.

Previously, it has been shown that mechanotransduction and cytoskeletal dynamics can fine-tune metabolic pathways[23]. For example, aldolase is anchored to the actin cytoskeleton in an inactive state and is released upon Rac1-mediated actin remodelling, leading to

**Fig. 8 | FBP-mediated Rac1 activation depends on RCC2. a,b,** Addition of FBP to whole cell lysate causes a conformational change in RCC2. Whole cell lysates of U-2 OS cells were subjected to LiP–MS. **a,** Peptide intensity of RCC2 (amino acids (aa) 110–120) in response to increasing FBP concentrations. **b,** FBP-responsive peptides of Rac1 (red, aa 6–16) and RCC2 (pink, aa 110–120) highlighted in a putative Rac1-RCC2 complex using the structurally related Ran-RCC1 complex as a template (Rac1 PBD ID: 1MH1, RCC2 aa 89-522 PDB ID: 5GWN, Ran-RCC1 complex PDB ID: 1I2M). **c,** FBP directly acts on RCC2. Purified eGFP–RCC2 in combination with increasing FBP levels was subjected to LiP–MS. Peptide intensity of RCC2 aa 454–470, 57–68 and 55–67 in response to increasing FBP concentrations. **d,** Efficient depletion of RCC2. Representative immunoblot of U-2 OS cells treated with the indicated siRNAs. β-actin and clathrin heavy chain (CHC) were used as loading controls, N = 2 independent experiments. **e–g,** RCC2 deletion phenocopies loss of aldolase. **e,** Representative confocal images of paxillin-stained FAs in U-2 OS cells treated with indicated siRNAs. FA segmentation and cell outlines (red) are shown below. **f,g,** A quantification of FAs per cell (**f**) and cell area (**g**) shown in **e**. Data represent mean ± s.e.m.; n = 5 independent experiments; one-way ANOVA with Dunnett's post-test (P values (for **f**): (siCtrl versus siRCC2#1) <0.0001; (siCtrl versus siRCC2#2) = 0.0092; P values (for **g**): (siCtrl versus siRCC2#1) <0.0001; (siCtrl versus siRCC2#2) <0.0001).

**h,** Efficient codepletion of RCC2 and Rac1. Representative immunoblot of U-2 OS cells treated with the indicated siRNAs. β-actin and CHC were used as loading controls; N = 2 independent experiments. **i,j,** Codepletion of Rac1 in RCC2-knockdown cells restores FA number (**i**) and cell area (**j**). A quantification of FAs per cell and cell area of U-2 OS cells treated with indicated siRNAs. Data represent mean ± s.e.m.; n = 5 independent experiments; one-way ANOVA with

Dunnett's post-test (P values (for **i**): (siCtrl versus siRAC1) = 0.0025; (siCtrl versus siRCC2#2) <0.0001; (siCtrl versus siRAC1 + siRCC2#2) = 0.0548; P values (for **j**): (siCtrl versus siRAC1) = 0.007; (siCtrl versus siRCC2#2) <0.0001; (siCtrl versus siRAC1 + siRCC2#2) = 0.6939). Corresponding images are shown in Extended Data Fig. 9d. **k,l,** Expression of RCC2 in aldolase-depleted cells rescues FA numbers (**k**) and cell size (**l**). A quantification of FAs per cell and cell area of U-2 OS cells transfected with control (−) or ALDOA-specific (+) siRNAs in combination with indicated plasmids. Data represent mean ± s.e.m.; n = 5 independent experiments; one-way ANOVA with Tukey's post-test (P values (for **k**): (siCtrl + eGFP versus siCtrl + eGFP–RCC2) = 0.6828; (siCtrl + eGFP versus siALDOA + eGFP) = 0.0006; (siCtrl + eGFP versus siALDOA + eGFP–RCC2) = 0.261; (siCtrl + eGFP–RCC2 versus siALDOA + eGFP) <0.0001; (siCtrl + eGFP–RCC2 versus siALDOA + eGFP–RCC2) = 0.0357; (siALDOA + eGFP versus siALDOA + eGFP–RCC2) = 0.0263; P values (for **l**): (siCtrl + eGFP versus siCtrl + eGFP–RCC2) = 0.7046; (siCtrl + eGFP versus siALDOA + eGFP) <0.0001; (siCtrl + eGFP vesus siALDOA + eGFP–RCC2) = 0.1491; (siCtrl + eGFP–RCC2 versus siALDOA + eGFP) <0.0001; (siCtrl + eGFP–RCC2 versus siALDOA + eGFP–RCC2) = 0.0193; (siALDOA + eGFP versus siALDOA + eGFP–RCC2) = 0.001).

**m,n,** FBP impairs complex formation between RCC2 and Rac1. eGFP-RCC2 or eGFP as control were coupled to eGFP-trap beads and incubated in the absence or presence of FBP (10 mM) with purified GST-Rac1. **m,** Eluates from washed beads were analysed by immunoblotting with GFP-, Rac1- and CHC-specific antibodies. **n,** A quantification of Rac1 amount bound to RCC2 in the presence or absence of FBP. Data represent mean ± s.e.m.; n = 5 independent experiments; two-sided one-sample t-test (P value = 0.0029). n.s., not significant; *P < 0.05, **P < 0.01, ***P < 0.001, ****P < 0.0001. Scale bars, 25 µm.

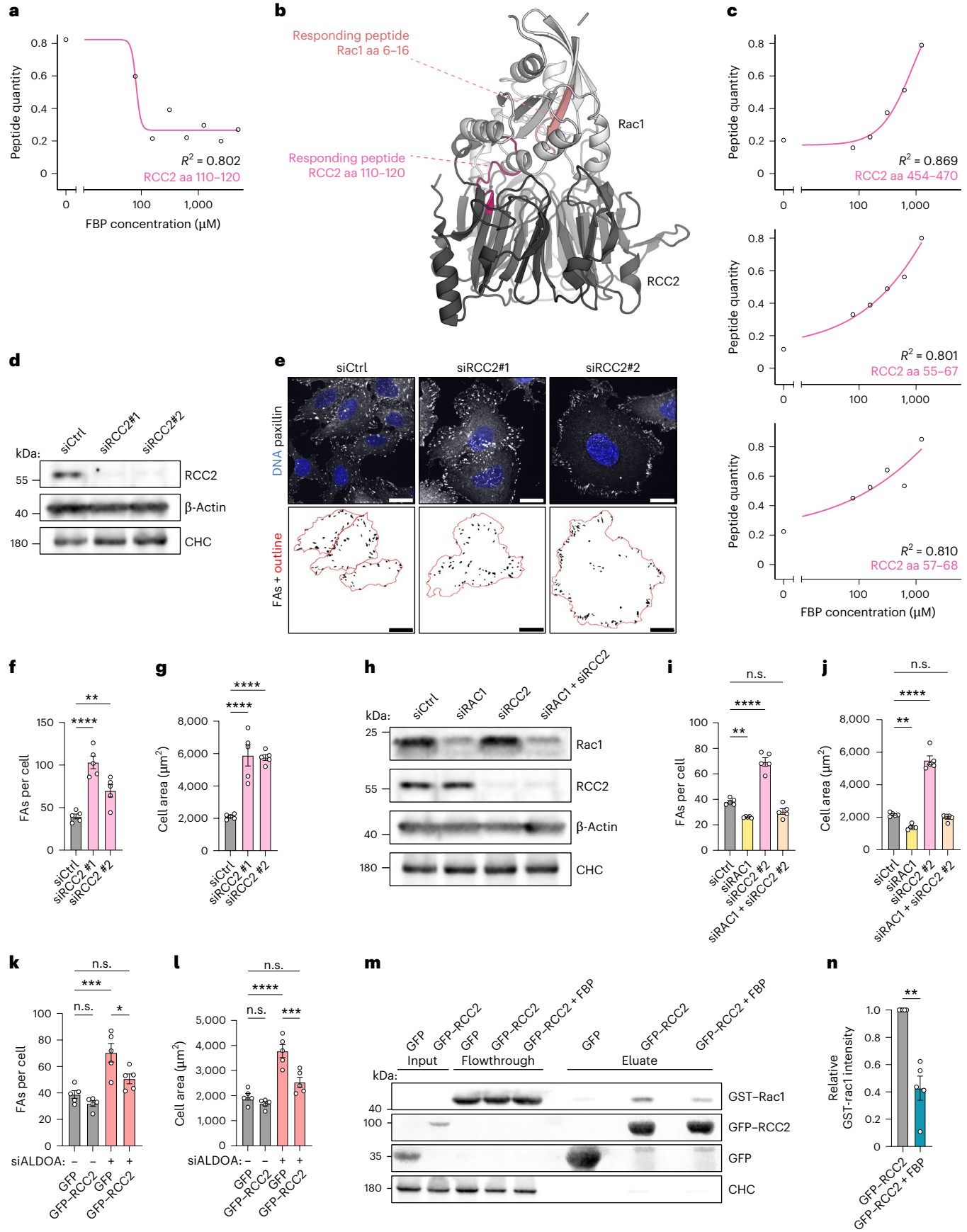

a concomitant increase in glycolytic energy production[59]. Thus, cytoskeletal dynamics can adapt metabolic pathways to varying energy demands. However, signalling routes acting in the reverse manner, feeding information about the metabolic state into cytoskeletal remodelling pathways, have remained elusive. With our finding that the metabolite FBP adapts actin dynamics to glycolytic flux, we close this gap in the regulatory loop demonstrating that metabolism and cytoskeletal dynamics reciprocally affect each other.

In addition to cytoskeletal dynamics influencing FBP production via aldolase regulation, another trigger for increased glycolysis might be an elevated glucose uptake during cell migration[60]. In line with our observation of elevated FBP levels in migrating and spreading cells, Kondo et al.[60] observed in a scratch wound assay that the cells close to the scratch display a higher rate of glucose uptake. This was traced back to an upregulation of the glucose transporters GLUT1 and GLUT4 in lower confluency states. Thus, the cellular sensing of available space for protrusion probably translates into an increased glucose uptake leading to enhanced glycolysis and elevated FBP levels. The increase in FBP in migratory and spreading cells conceivably promotes their Rac1-mediated protrusive activity. Consistent with our hypothesis of glycolytic flux as a requirement for cellular protrusion and migration, Kondo et al. observed that inhibiting glycolysis with 2-DG significantly impaired wound closure[60].

An important future question is in which organismic context FBP's signalling role is most relevant. Most differentiated cells primarily use oxidative phosphorylation to generate the necessary energy[61]. However, during the preplacental phase of development oxygen is scarce[62], and even in the adult body, stem cells niches are found in low oxygen environments[62], consistent with the observation that stem cells primarily use glycolysis and only switch to oxidative phosphorylation upon differentiation[63]. Interestingly, in the murine primitive streak, a gradient of glycolytic metabolism has been found to be crucial to instruct further development[64]. We speculate that FBP might support cellular adhesion and spreading of not yet differentiated cells in different developmental contexts.

In addition, it is known that also in the adult organism certain cell types switch their metabolism from oxidative phosphorylation to glycolysis even though the amount of ATP produced in this manner is considerably lower[65] and even when oxygen is available. This phenomenon, coined Warburg effect, has first been observed for cancer cells and is considered as one of the hallmarks of cancer progression[66,67]. Consistently, cancer cells have been found to have increased FBP levels[68], and the stable expression of oncogenic Ras leads to a striking upregulation of FBP[68]. Although it was initially hypothesized that the increase in aerobic fermentation of cancer cells resulted from impaired mitochondrial activity, growing evidence suggests that cancer cells might take up so much glucose that mitochondrial oxidation cannot keep up, leading to lactate production[69]. Interestingly, the acidification of the microenvironment caused by secreted lactate is believed to promote invasion[70]. Our findings suggest that enhanced glycolysis might additionally promote invasion by increasing cell protrusion through FBP generation.

This might especially be the case in tumours, which downregulate aldolases such as hepatocellular carcinomas where lower aldolase B levels are linked to a poor prognosis[71]. Similar to our results, decreased aldolase B levels cause an increase in FBP as shown for aldolase B-deficient mouse liver tumour cells[71]. In line with our finding of a protrusion-promoting effect of elevated FBP levels, Li et al.[71] show a two-fold higher migration of aldolase B silenced liver cancer cells in transwell assays, which might contribute to the frequently observed metastatic spread of hepatocellular carcinomas[72]. Of note, the situation might differ depending on tumour type and aldolase expression pattern. For example, for liver cancers high aldolase A levels have been associated with a poor patient survival[41], and the knockdown of aldolase A in liver cancer cells with their specific metabolic wiring

does not only lead to increased FBP levels, in line with our data, but also to energy exhaustion and strongly reduced cell viability due to imbalanced glycolysis[41].

The importance of FBP for tumour cells is also underlined by the fact that fructose-1,6-bisphosphatase 1, a gluconeogenic FBP-degrading enzyme, is a tumour suppressor and depleted in clear cell renal cell carcinoma[73], the most common form of kidney cancer, whereas nucleophosmin (NPM1) promotes pancreatic cancer progression by inhibiting fructose-1,6-bisphosphatase 1[74]. Because FBP is an allosteric activator of the end-step glycolytic enzyme pyruvate kinase, the increase in its level has been viewed as means to promote cancer cell glycolysis. However, in light of our results the elevated FBP levels might simultaneously stimulate adhesive and protrusive properties of tumour cells and therefore promote tumour cell survival and invasion.

In addition, activated endothelial cells and immune cells such as T cells and dendritic cells undergo glycolytic reprogramming upon activation[22,75–77]. While the metabolic switch probably fulfills multiple functions in these cells, its different benefits are not entirely clear. Our findings open up the possibility that upregulated glycolysis, among other benefits, might provide an increased FBP supply for Rac1 activation, promoting endothelial sprouting and immune cell migration.

In fact, our results might contribute to fill a gap in the current understanding of the molecular events downstream of elevated glycolysis in immune and endothelial cells. For example, vessel branching by endothelial cells was shown to depend on an upregulation of glycolysis via phosphofructokinase-2/fructose-2,6-bisphosphatase 3 (PFKFB3), which synthesizes F2,6BP, an allosteric activator of PFK-1 and the most potent stimulator of glycolysis[75,78]. In fact, PFKFB3 is essential for the necessary lamellipodia formation in endothelial cells[75]. Although the authors speculated that glycolysis might play a crucial role in cytoskeletal remodelling, the exact mechanism beyond providing a local supply of ATP for actin polymerization remained obscure. Our results suggest that a concomitantly increased FBP supply triggers Rac1-mediated actin remodelling and lamellipodia formation in endothelial cells to support vessel sprouting. Consistent with this, endothelial cells harbour RCC2–Rac1 complexes[52].

For activated dendritic cells, the switch to glycolysis is essential to retain their characteristic dendritic morphology and to migrate to lymph nodes for T cell activation[76]. At the same time, Rac GTPases play an important role in dendritic cell migration, as Rac1/2 deficient dendritic cells are not able to efficiently reach lymph nodes[79]. Thus, also here increased FBP levels after glycolytic reprogramming might well serve to promote cellular protrusion.

Finally, Rac1 has important functions in other cell types, thus, the connection between Rac1 and FBP might also be relevant there. In β-cells, for example, Rac1-dependent actin remodelling is crucial for the second phase of glucose-stimulated insulin secretion, which relies on the mobilization of insulin granules from behind an actin barrier[80]. At the same time, glycolysis is essential in β-cells for coupling glucose sensing to insulin secretion. FBP levels were shown to increase in line with glucose concentration in β-cells so that FBP can serve as read-out for increased glucose availability[81]. Thus, it will be interesting to test whether FBP produced in β-cells during glucose metabolism contributes to insulin secretion via Rac1 activation.

In contrast to Rac1, which has been studied in many cell types, there is little information on RCC2 function in regards to immune or β-cells. However, in line with our confirmation of the FBP-mediated phenotypes in four different cell lines (U-2 OS, MDA-MB-231, RPE-1 and HFF), RNA-based expression profiling demonstrates that RCC2 is widely expressed with above-median levels in dendritic, endothelial and β-cells[82] consistent with a potential FBP-orchestrated regulatory role in those cells, a hypothesis that remains to be tested.

In summary, our work reveals that the impact, which cellular reprogramming towards aerobic glycolysis exerts on cytoskeletal remodelling, probably goes beyond providing a fast local ATP supply.

Therefore, it will be interesting in the future to unravel the importance of FBP-based signalling in the different cellular systems, which rely on high levels of glycolysis and actin reorganization, and to resolve its spatiotemporal pattern.

## Online content

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

## Methods

Our research complies with all relevant ethical regulations.

### Plasmids

Retroviral transfer plasmids (HA-mCherry-pLIB-Neo, HA-ALDOA-WT-pLIB-Neo, HA-ALDOA-D33S-pLIB-Neo and HA-ALDOA-K229A-pLIB-Neo) for virus production were generated by restriction digest of the pLIB-IRES-Neo vector (gift from Martin Lehmann) with EcoRI and SalI followed by insertion with Gibson assembly. The corresponding plasmid templates were generated by subcloning an siRNA-resistant version (siRNAs #1 to #4, synthesized by Absea Biotechnologies) of human ALDOA into a pcDNA3.1(+) vector with an N-terminal HA-tag using restriction digest (EcoRI, XhoI) and Gibson assembly. Mutations were introduced by site-directed mutagenesis. The mCherry sequence was amplified from the pmCherry-N1 plasmid (Clontech). The retroviral transfer plasmid CMV-paxillin-pLIB-Puro was created by Marietta Bergmann. The retroviral packaging plasmid pCIG3 N was a gift from Jeremy Luban (Addgene #132941) and the retroviral envelope plasmid pMD2.G was a gift from Didier Trono (Addgene #12259). The plasmids pRK5-myc-Rac1-WT, pRK5-myc-Rac1-T17N and pRK5-myc-Rac1-Q61L were gifts from Gary Bokoch (Addgene #12985, #12984 and #12983). Human RCC2 was amplified from U-2 OS cDNA and subcloned into a peGFP-C1 vector (Clontech) using restriction digest (SmaI, EcoRI) and Gibson assembly. The peGFP-N1 plasmid (Clontech) was used as a control for overexpression experiments. To generate Rac1-pGEX-6P-1 for bacterial expression, the pGEX-6P-1 vector was digested with BamHI and NotI, the human Rac1 sequence was PCR-amplified from pRK5-myc-Rac1-WT and inserted using Gibson assembly. The baculovirus donor plasmid pFL-10His-eGFP-RCC2 was generated with Gibson assembly using RCC2-peGFP-C1 as a PCR template.

### siRNAs

All siRNAs used in this study are listed in the Supplementary Table in Tab 3. Unless indicated otherwise, siRNAs were synthesized by Merck with 3′-dTdT overhangs. Control siRNA as well as siRNAs against GAPDH and paxillin were obtained as pools consisting of four siRNAs.

### Antibodies

Primary antibodies used in this study are listed in the Supplementary Table in Tab 4. Secondary antibodies against mouse or rabbit IgG conjugated with Alexa Fluor 488, 568 or 647 (Thermo Fisher Scientific) were used for immunofluorescence experiments at a concentration of 1:200. Secondary antibodies against mouse or rabbit IgG coupled to IRDye 680RD or IRDye 800CW (LI-COR Biosciences) were used for immunoblotting at a concentration of 1:10,000.

### Inhibitors

Antimycin A (Abcam) was diluted in dimethylsulfoxide (DMSO) to a concentration of 1 mM, stored at −80 °C and freshly diluted to a working concentration of 1 µM in each experiment. 2-Deoxyglucose (Merck) was diluted in PBS to a concentration of 1 M, stored at −20 °C and freshly diluted to a working concentration of 25 mM in each experiment.

### Cell culture

The following cell lines were obtained from the American Type Culture Collection (ATCC), which authenticated them via short tandem repeat profiling: the human sarcoma cell line U-2 OS (ATCC: HTB-96; female), the human embryonic kidney cell line HEK293T (ATCC: CRL-3216; sex not specified), the human breast cancer cell line MDA-MB-231 (ATCC: HTB-26; female) and the telomerase-immortalized human retinal pigment epithelium cell line hTERT RPE-1 (ATTC: CRL-4000; female). Telomerase-immortalized HFF (male) were a kind gift of Professor Martin J. Humphries (University of Manchester, UK) and were validated by their fibroblast-like morphology. All cell lines except hTERT RPE-1 cells were cultured in DMEM (4.5 g l⁻¹ glucose,

Thermo Fisher Scientific) supplemented with 10% (v/v) FBS (Thermo Fisher Scientific), 100 U ml⁻¹ penicillin (Thermo Fisher Scientific) and 100 µg ml⁻¹ streptomycin (Thermo Fisher Scientific). hTERT RPE-1 cells were cultured in DMEM/F-12 (containing L-glutamine, HEPES buffer and sodium bicarbonate, Sigma-Aldrich) supplemented with 10% (v/v) FBS (Thermo Fisher Scientific), 100 U ml⁻¹ penicillin (Thermo Fisher Scientific), 100 µg ml⁻¹ streptomycin (Thermo Fisher Scientific) and 0.1 mg ml⁻¹ hygromycin B (Roth). All cells were routinely tested for *Mycoplasma* contamination.

### Genome-wide screening

Genome-wide screening was performed using the Dharmacon human ON-TARGETplus siRNA library (Horizon Discovery), which is arrayed on 66 × 384-well microplates and targets 18,091 genes with a pool of four siRNAs per target gene. For the reverse transfection on a Freedom EVO 200 workstation (Tecan), 5 µl of 250 nM siRNA per well, diluted in Opti-MEM (Thermo Fisher Scientific), were placed into a 384-well PhenoPlate (PerkinElmer) followed by the addition of 0.02 µl RNAiMAX (Thermo Fisher Scientific) in 5 µl Opti-MEM using a BioTek EL406 Washer Dispenser (Agilent). After 20 min of incubation at room temperature, 550 U-2 OS cells in 30 µl DMEM were seeded onto the transfection mixture using the BioTek EL406 Washer Dispenser resulting in a final concentration of 25 nM siRNA per well. In addition to mock-transfected cells, each plate contained control siRNAs targeting either paxillin or cofilin-1 + destrin, as well as a cell-death inducing siRNA mixture (AllStars Hs Cell Death siRNA, Qiagen) at a final concentration of 25 nM (see Extended Data Fig. 1a for the plate layout).

72 h post-transfection, the cells were fixed with cold 4% (w/v) paraformaldehyde (PFA) for 10 min at room temperature and washed with PBS. All steps of the fixation and immunofluorescence staining protocol were performed on a BioTek EL406 Washer Dispenser. Fixed cells were permeabilized and blocked for 1 h in goat serum dilution buffer (GSDB: 25% goat serum, 0.3% Triton X-100 in PBS). For immunostaining, a paxillin-specific antibody was used in a 1:500 dilution in GSDB at 4 °C overnight. After three washes with PBS, a goat anti-rabbit Alexa Fluor 568 conjugated secondary antibody diluted 1:600 in GSDB was incubated for 1 h at room temperature followed by three washes with PBS. Nuclei were stained with 20 µM Hoechst33342, and the microplates were sealed with sealing tape (Thermo Fisher Scientific).

Imaging was performed on an inverted microscope (Ti Eclipse, Nikon) equipped with a spinning disk unit (CSU-X1 Yokogawa), a laser combiner (405, 488, 561 and 641 nm, ALC Andor), a quad dichroic (400-410, 486-491, 560-570, 633-647, Semrock), a 40× Plan Apo 0.95 NA air objective (Nikon) and an EMCCD camera (AU-888, Andor). The microscope was operated with NIS Elements software (version 5.21.03, Nikon) at room temperature. The final pixel size was 0.16 µm. For sequential image acquisition, the following imaging parameters were used: excitation 405 nm (emission 450/50 nm) for Hoechst and excitation 561 nm (emission 600/50 nm) for Alexa Fluor 568 with 50 ms and 200 ms exposure time, respectively. Automated imaging in 384-well plates was performed with a perfect focus system (Nikon) for nuclei detection followed by autofocusing on the FA channel with a Piezo frame (Nanodrive, 200 µm range, MadCityLabs) using the NIS Elements JOBs Modules (Nikon).

Image analysis was performed with KNIME[83] version 3.7.0 with the KNIME Image Processing Community Extension[84] and the ImageJ and CellProfiler integration. The CellProfiler[85] version was 3.1.8. Image quality metrics were extracted with a CellProfiler quality metric pipeline[86] and used to exclude out-of-focus images, which were identified by a local focus score smaller than 0.03 in the nucleus channel.

The FA channel was processed with a median filter of radius 2, followed by a rolling ball background subtraction of radius 1 with the sliding paraboloid option turned on. Individual FAs were segmented by applying a global intensity-based threshold using Otsu's method on the processed FA channel. Image fields with more than 20,000 identified

structures were excluded from further analysis to remove remaining out-of-focus images. In addition, structures smaller than 50 pixels or larger than 5,000 pixels were removed from the analysis, as these are not in the size range of FAs and probably represented artefacts. The number of nuclei per field was determined using a Gaussian filter of radius 20 on the nucleus channel followed by local maxima detection with a noise level of 100. Image fields with two nuclei or fewer were excluded from analysis. The area of each FA, number of FAs per field and number of nuclei per field were extracted from the output of the image analysis. The number of FAs per cell was calculated by dividing the number of FAs per field by the number of nuclei per field.

To evaluate and validate the assay quality for each plate, the $Z'$-factor[87] was calculated from the parameter 'FAs per cell' of the mock- and siPXN-transfected wells. The $Z'$-factors of the 66 plates, which were included in the primary screen analysis, ranged from 0.278 to 0.853, with 54 plates (82%) having a $Z'$-factor >0.5. For plate-wise normalization of the sample data, $Z$-scores were calculated from the parameters 'FAs per cell' and 'FA area'. Hits were defined as genes with a $Z$-score >4.2 for 'FAs per cell' or a $Z$-score >3.5 for 'FA area'. Treatments that resulted in a low cell viability (<50 cells) or led to staining artifacts were excluded from further analysis. The resulting 280 candidate genes were chosen for the secondary screen.

For the secondary screen, the siRNA pools of the 280 selected candidate genes were picked from the library plates onto a new 384-well plate (reproducing the primary screen layout) and screened another two times following the same protocol used for the genome-wide screen. For comparison, the values of 'FAs per cell' and 'FA area' were normalized to mock-transfected controls, and the results of the two replicates were averaged. Hits were defined as genes with an average increase of 30% or higher compared with control conditions in either 'FAs per cell' or 'FA area'. All screening results are listed in the Supplementary Table in Tabs 1 and 2.

## Generation of stable cell lines

Retrovirus was generated by transfecting 70% confluent HEK293T cells grown in 10-cm dishes with a mixture of retroviral packaging plasmid (pCIG N), envelope plasmid (pMD2.G) and transfer plasmid of choice (based on pLIB-CMV-Puro/-Neo) using the calcium phosphate transfection method. On the following day, the medium was exchanged with fresh DMEM. Cell supernatants containing the virus were collected 48 h and 72 h post transfection and combined. To remove cell debris, the viral suspension was centrifuged for 5 min at 3,000$g$ and subsequently filtered through 0.45-μm pore filters.

Stable cell lines were generated by adding half of the viral suspension to 50% confluent U-2 OS cells grown in 10-cm dishes followed by an exchange of medium on the next day. Three days after infection, cells were selected by incubating in DMEM containing either 1.5 μg ml$^{-1}$ puromycin (Thermo Fisher Scientific) or 500 μg ml$^{-1}$ G418 (InvivoGen) for 1 week. Note that G418 selection was performed in the absence of penicillin and streptomycin. The expression of the transgene was confirmed by immunoblot analysis or fluorescence microscopy, if applicable. U-2 OS cells stably expressing eGFP–paxillin were further separated into single clones using fluorescence-activated cell sorting to obtain a homogeneous expression.

## siRNA and plasmid transfection

For knockdown experiments with all cell lines except hTERT RPE-1 cells, siRNAs at a concentration of 50 nM were transfected on two consecutive days using the RNAiMAX transfection reagent (Thermo Fisher Scientific) according to the manufacturer's instructions. On the first day, cells were reverse transfected by freshly seeding them onto a transfection mixture containing siRNA and RNAiMAX diluted in Opti-MEM. On the following day, the medium was replaced with fresh DMEM, and cells were forward transfected by adding the same transfection mixture to the adherent cells. The medium was exchanged

the next day and cells were processed 96 h after the first transfection. hTERT RPE-1 cells were transfected with INTERFERin (Polyplus) according to the manufacturer´s instructions. To obtain less severely silenced cells for the liquid chromatography coupled with tandem mass spectrometry (LC−MS/MS)-based FBP measurements, only one round of knockdown was performed. For this, cells were transfected on the first day as described above, followed by a medium exchange on the following day, and processed after 72 h.

For simultaneous overexpression and knockdown, cells were seeded in 24-well plates. On the following day, siRNAs at a concentration of 50 nM and 300 ng of plasmid DNA were cotransfected using the Lipofectamine2000 transfection reagent (Thermo Fisher Scientific) according to the manufacturer's instructions. The medium was exchanged the next day, and the cells were further processed 72 h after transfection.

## Immunofluorescence, confocal imaging and image analysis

Cells grown on Geltrex-coated (0.3 mg ml$^{-1}$, Thermo Fisher Scientific) glass cover slips were briefly washed with PBS and fixed with cold 4% (w/v) PFA for 10 min at room temperature. Then cells were briefly washed three times with PBS and simultaneously blocked and permeabilized with GSDB for 30 min. Afterwards, cover slips were incubated with primary antibodies diluted in GSDB for 1 h at room temperature. After three brief washes with PBS, cover slips were incubated with fluorescently labelled secondary antibodies diluted in GSDB for 30 min at room temperature. For F-actin labelling, phalloidin-Alexa Fluor 488 (Thermo Fisher Scientific) was added 1:50 to the secondary antibody solution. Finally, the cells were washed another three times with PBS and mounted on glass slides using Immu-Mount (Thermo Fisher Scientific) supplemented with 1 mg ml$^{-1}$ DAPI (Thermo Fisher Scientific). Cover slides were left to dry, and the images were acquired with the confocal microscope described earlier or with a similar inverted microscope (Ti2-E, Nikon) equipped with a CSU-W1 SORA spinning disk unit. For sequential image acquisition, the following imaging parameters were used: excitation 405 nm (emission 450/50 nm) for Hoechst, excitation 488 nm (emission 525/50 nm) for eGFP/Alexa Fluor 488, excitation 561 nm (emission 600/50 nm) for Alexa Fluor 568 and excitation 641 nm (emission 700/75 nm) for Alexa Fluor 647 with 50−200 ms exposure time.

The number of FAs per cell was analysed in Fiji[88] (version 1.54f) using a semi-automated macro. In brief, regions of interest were drawn manually around cells that were completely in the field of view. The paxillin channel (FAs) was then automatically processed using a median filter of radius 2, followed by rolling ball background subtraction of radius 1 and with the sliding paraboloid setting enabled. The resulting image was automatically thresholded using Otsu's method and filtered for objects with a size ranging from 50 to 500 pixels and a circularity between 0.00 and 0.95. The number of cells was determined by processing the nuclear channel with a Gaussian blur of radius 20, followed by maxima detection with a prominence of 100 and edge exclusion enabled. Finally, the number of FAs per image divided by the number of nuclei per image. The cell area was quantified manually by drawing regions of interest around complete cells in the F-actin channel or a channel containing a cytoplasmic marker. Examples of the resulting FA masks as well as the manually drawn cell outlines are depicted in the corresponding figures.

## Cell death assay

SiRNA-treated cells were seeded into Geltrex-coated wells of a μ-Slide 8 well (Ibidi). After 4 h, the cells were incubated for 30 min in the cell culture incubator with 1 μM of the cell-permeable far-red DNA stain SiR-Hoechst and 0.1 μM of the membrane-impermeant DNA dye SYTOX Green (Invitrogen), which only stains cells with compromised plasma membranes. Control cells treated with 0.5% Triton X-100 for 2 min before dye addition to impair membrane integrity were used

as positive control. Samples were imaged by epifluorescence microscopy. The stained nuclei were manually counted, and the number of SYTOX-positive nuclei (dead cells) was divided by the number of SiR-Hoechst-positive nuclei (all cells) to calculate the percentage of dead cells per condition.

## Cell spreading

Cells grown in six-well plates were briefly washed with PBS and detached with TrypLE dissociation reagent (Thermo Fisher Scientific) for 5 min at 37 °C. Subsequently, cells were collected in fresh DMEM and seeded onto Geltrex-coated cover slips. After 30, 60, 120 and 240 min of adhesion and spreading, cells were fixed, and F-actin was labelled as described above. Images were acquired with an inverted spinning disk confocal microscope as described above, and the cell area was automatically quantified using a custom-made Fiji macro. In brief, cell masks were created from the F-actin channel by processing with a Gaussian blur of radius 4 followed by automated thresholding using the Triangle algorithm. The cell masks were measured with the 'Analyze Particles' function excluding objects smaller than 500 pixels and subsequently divided by the number of nuclei, which was determined by processing the nuclear channel with a Gaussian blur of radius 20 followed by maxima detection with a prominence of 100.

## FA dynamics

U-2 OS cells stably expressing eGFP–paxillin were seeded on Geltrex-coated eight-well glass bottom imaging chambers (Ibidi) 4 h before the experiment. Afterwards, the medium was exchanged with Fluorobrite DMEM supplemented with 10% (v/v) fetal bovine serum and 4 mM L-glutamine.

Total internal reflection fluorescence (TIRF) microscopy was performed with an inverted microscope (Ti Eclipse, Nikon) equipped with an incubation chamber with temperature and $CO_2$ control (Okolab), a laser combiner (405, 488, 561 and 642 nm, Omicron), a Notch Dichroic (Di01-R405/488/561/635, Semrock), a 60× APO TIRF 1.49 NA oil objective (Nikon) and a back illuminated sCMOS camera (Prime95B, Photometrics). The microscope was operated with NIS Elements software (Nikon) at 37 °C and 5% $CO_2$. The final pixel size was 0.122 μm. EGFP fluorescence was excited at 488 nm (emission 525/40) with an exposure time of 70 ms every 2 min for 4 h. All conditions of one experiment were imaged in parallel with a perfect focus system (Nikon) using the NIS Elements JOBs Modules (Nikon).

After acquisition, movies underwent a manual quality control to exclude dying cells and cells that left the field of view from the analysis. If necessary, movies were drift-corrected with the 'Correct 3D drift' plugin[89] in Fiji as follows: only pixels with grey values of 120 were considered, and the maximum shift corrected in $x$ and $y$ was set to 20 pixels. Subpixel drift correction, multi-time scale computation and edge enhancements were turned off. To reduce file size and black padding caused by drift correction, an automatic segmentation-based approach was used to restrict the movie to the area that contained the moving cells. To this end, a maximum intensity projection was created, a Gaussian blur of radius 15 was applied and an automatic intensity threshold using the Triangle algorithm was used to create a segmentation mask. Any holes in the segmentation mask were filled and the largest connected component was retained using the 'Keep largest region' function of the MorphoLibJ plugin[90]. The resulting mask was then used to define a crop area, which was automatically applied to the movie stack. Residual black padding was removed by manual cropping.

FA dynamics were analysed using the FA analysis server[91]. For tracking, no median filter was applied to the individual frames. The segmentation threshold was set to 2, and FA splitting was turned off. Minimum FA size was set to ten pixels to exclude most very small and short-lived structures from the analysis. The parameters FA assembly rate and disassembly rate were extracted from the output data using a custom-made Python script (Python version 3.8.3). The number

of novel FAs was computed based on the tracking matrices of the FA analysis server: the matrices were filtered for tracks with unclear starts (tracks present in the first frame) and tracks that emerged from merge or split events. Afterwards, the number of starting tracks per frame was counted and averaged over each time-lapse movie.

## Protrusive area measurements

Cells grown in six-well plates were briefly washed with PBS and detached with TrypLE dissociation reagent for 5 min at 37 °C. Subsequently, cells were collected in fresh DMEM, seeded onto Geltrex-coated eight-well imaging chambers (Ibidi) and left to adhere and spread for 4–6 h. Then, cytoplasm and cell membranes were stained by a 30 min incubation with CellTracker Green CMFDA Dye (1:1,000, Thermo Fisher Scientific) and Cell Mask Deep Red Plasma Membrane Stain (1:1,000, Thermo Fisher Scientific) in DMEM. The remaining dye was washed off with PBS, and the medium was exchanged with Fluorobrite (Thermo Fisher Scientific) supplemented with 10% (v/v) FBS and 4 mM L-glutamine.

Phase contrast and epifluorescence images were acquired with an inverted microscope (Ti Eclipse, Nikon) equipped with an incubation chamber with temperature and $CO_2$ control (Okolab), an LED lamp (CoolLED, pE4000), a 40× APO Plan 0.95 NA Ph2 air objective (Nikon) and a back illuminated sCMOS camera (Prime95B, Photometrics). The microscope was operated with NIS Elements software (Nikon) at 37 °C and 5% $CO_2$. The final pixel size was 0.275 μm. For sequential image acquisition, the following parameters were used: excitation 470 nm (emission: 525/50) for CellTracker CMFDA and excitation 635 nm (emission 690/50 nm) for CellMask Deep Red with 50 ms and 100 ms exposure time, respectively.

Protrusive cell area was analysed using an automated Fiji macro based on the following principle: total cell area can be visualized using a plasma membrane dye, and elevated parts of the cell can be differentiated from flat, protrusive parts of the cell because they accumulate more cytoplasmic fluorescence when imaged with an epifluorescence microscope. To this end, the plasma membrane channel was processed with a Gaussian blur of radius 1, and the combined cell outline was segmented using the Huang2 threshold. Similarly, elevated cell areas were segmented by processing the cytoplasmic channel with a Gaussian blur of radius 1 followed by a Huang2 threshold. In addition, a copy of the cytoplasmic channel was processed by a Gaussian blur of radius 25 followed by a maxima detection with a prominence of 100 to determine the cell count and to create valleys for the watershed segmentation. Individual cells were segmented using the 'Marker-controlled Watershed' plugin from the MorphoLibJ library[90] with the unprocessed, inverted cytoplasmic channel as 'input', the combined cell outline as 'mask' and cytoplasmic maxima as 'marker'. The area of the resulting masks was measured, and the corresponding elevated area was subtracted to calculate the protrusive area (see Extended Data Fig. 6a for a workflow example).

## Cell lysis and immunoblot analysis

Cells were washed once with cold PBS and subsequently scraped into lysis buffer (20 mM HEPES pH 7.4, 130 mM NaCl, 2 mM $MgCl_2$, 1% Triton X-100, 1 mM phenylmethylsulfonyl fluoride (PMSF), Protease Inhibitor Cocktail (Merck), Phosphatase Inhibitor Cocktail 2 and 3 (Merck)). The cells were briefly vortexed, incubated on ice for 15 min and subsequently centrifuged for 8 min at 17,000$g$ at 4 °C. Cleared lysates were collected, and the protein concentration was determined using the Bradford assay. Finally, lysates were adjusted to 1× Laemmli buffer and heated for 10 min at 95 °C.

Equal amounts of protein (between 15 and 25 μg) were resolved by SDS–polyacrylamide gel electrophoresis and subsequently transferred to polyvinylidene fluorid membranes with low background fluorescence (PVDF-FL) (Merck) using a wet tank transfer system. Membranes were briefly washed with TBS + 0.1% Tween-20 (TBS-T) and blocked with intercept (TBS) blocking buffer (LI-COR Biosciences) for

1 h at room temperature. Afterwards, membranes were incubated with primary antibodies (diluted in 1:1 TBS-T/blocking buffer) overnight at 4 °C. On the next day, unbound antibodies were washed off three times with TBS-T for 5 min, and membranes were incubated in fluorescently labelled secondary antibodies for 30 min at room temperature. After another three washes with TBS-T, fluorescent images were acquired with an Odyssey Fc Imaging System (LI-COR Biosciences). Where required, band intensities were analysed with Image Studio Lite (version 5.2 or 6.1, LI-COR Biosciences) and normalized to loading controls.

#### Effector pull-down assays

Cells were washed once with cold PBS and subsequently scraped in GTPase lysis buffer (50 mM TRIS pH 7.2, 500 mM NaCl, 10 mM $MgCl_2$, 1 mM PMSF, Protease Inhibitor Cocktail, Phosphatase Inhibitor Cocktail), briefly vortexed and centrifuged for 2 min at 17,000g at 4 °C. Cleared lysates were collected and incubated with 25 µg of GST-PAK-PBD beads (Cytoskeleton) for 1 h at 4 °C under constant rotation. Afterwards, the beads were pelleted by centrifugation, and the unbound supernatant was discarded. Beads were washed with GTPase washing buffer (25 mM TRIS pH 7.2, 150 mM NaCl, 10 mM $MgCl_2$, 1 mM PMSF, Protease Inhibitor Cocktail, Phosphatase Inhibitor Cocktail) and briefly pelleted by centrifugation. The supernatant was discarded. Bound proteins were eluted from the beads by incubation in 2× Laemmli buffer for 10 min at 95 °C. Finally, samples were subjected to immunoblot analysis as described above.

#### ATP measurements

Equal amounts of cells were seeded onto black 96-well plates (PerkinElmer) and left to rest for 4–24 h. ATP concentrations were determined with the ATPlite Luminescence ATP Detection Assay System (PerkinElmer) according to the manufacturer's instructions using a TECAN Infinite MPLEX plate reader. Measurements were adjusted to protein concentration determined by the Bradford assay.

#### FBP measurements

FBP levels were either measured using a fluorometric FBP assay kit (Abcam) or by high-performance anion-exchange LC–MS/MS. The fluorometric assay was performed according to the manufacturer's instructions. In brief, cells were washed once with cold PBS and lysed with F1,6BP assay buffer, briefly vortexed and incubated for 15 min on ice. Lysates were cleared by centrifuging for 10 min at 10,000g at 4 °C. A small fraction of the lysate was used to determine protein concentration with the Bradford assay. The remaining lysate was deproteinized by centrifugation through an Amicon 10 kDa MWCO Spin Column (Merck) for 35 min at 14,000g at 4 °C. The supernatant was collected and used for FBP measurements. Reactions were performed in black 96-well plates, and fluorescence intensity was measured with a TECAN Infinite MPLEX plate reader. Measurements were adjusted for background fluorescence and normalized to protein concentration.

To generate samples of confluent cells for LC–MS/MS-based FBP measurements, $2 \times 10^6$ cells were seeded into a 6-cm dish and processed 24 h later at a confluency of 100%. For the migrating and spreading condition, two million cells each were seeded into a 10-cm dish and processed 1 h later to analyse cells in the process of spreading respectively 6 h later to measure FBP in migrating cells. For sample preparation, cells were washed two times with ice-cold PBS and then scraped into ice-cold PBS. After 5 min centrifugation at 400g and 4 °C, the supernatant was removed, and cell pellets were frozen in liquid nitrogen for storage.

For extraction, cell pellets were left in 350 µl chloroform/methanol (3:7, v/v) for 2 h at −20 °C under occasional vortexing. After addition of 350 µl ice-cold water and vortexing, the samples were centrifuged for 10 min at 13,000g and 4 °C, and the upper aqueous phase was transferred into a new tube. The chloroform phase was re-extracted with 300 µl ice-cold water and, after centrifugation as above, the second aqueous phase was removed and combined with the first. The combined aqueous phases were dried under continuous nitrogen flow and reconstituted with 350 µl water. High-molecular weight contaminants were removed from the extracts by centrifugal filtration through MultiScreen PCR-96 Filter Plates (Merck Millipore) for 0.5 h at 2,500g and 15 °C, before the extracts were used for metabolite analysis. In parallel, the residual chloroform phase from the extraction, including the insoluble residue from the interface, was evaporated to dryness under continuous nitrogen flow, resuspended in 200 µl 0.1 M NaOH and heated at 98 °C for 30 min to solubilize the protein. The protein content of each sample was measured using the Coomassie Blue dye-binding assay[92], with bovine serum albumin as the standard, for later normalization of the measured FBP.

For metabolite analysis, the filtered extracts were simultaneously diluted 1:10 with water and spiked with 2.5 pmol of $[^{13}C_6]$FBP (CLM-8962-PK; Cambridge Isotope Laboratories). FBP was measured by high-performance anion-exchange LC–MS/MS, using an ICS 5000+ chromatograph (Thermo Scientific Dionex) fitted with an AS50 autosampler (operating at 4 °C), a 2× 50-mm AG11-HC-4 µm guard-column and a 2× 250-mm Ion Pac AS11-HC-4-µm column (Thermo Scientific Dionex) in series and coupled to a QTrap 6500 triple quadrupole mass spectrometer (Sciex). The ICS 5000+ chromatograph was equipped with an eluent generator fitted with EGC 500 KOH eluent cartridges (Thermo Scientific Dionex); and 5% (v/v) methanol (degassed online) was passed through a 9× 75-mm IonPack ATC-HC 500 trap column (Thermo Scientific Dionex) to remove anionic contaminants before entering the eluent cartridges. The column was equilibrated for 12 min with 5 mM KOH containing 5% (v/v) methanol and kept at a constant 20 °C. Aliquots (50 µl) of calibration standards containing 0.03–12.5 pmol FBP (F6803; Merck KGaA) or the spiked cell extracts were injected onto the column, and anionic compounds were eluted with the following programme: (1) 0–5 min, isocratic 5 mM KOH; (2) 5–21 min, linear gradient from 5 to 30 mM KOH; (3) 21–30 min, linear gradient from 30 to 50 mM KOH; (4) 30–34 min, linear gradient from 50 to 100 mM KOH; and (5) 34–39 min, isocratic 100 mM KOH. The retention time of FBP was 37.9 min, and there was baseline separation from glucose-1,6-bisphosphate (retention time of 36.4 min) and F2,6BP (retention time of 35.7 min). $K^+$ ions were removed from the postcolumn eluent by passage through a 2-mm AERS 500 ion suppressor (Thermo Scientific Dionex), operating at 10 mA (0–10 min), 20 mA (10–29.5 min) and 50 mA (from 29.5 min onwards), in external water mode, before entry into the mass spectrometer.

The QTrap 6500 was operated in multiple reactions monitoring mode, with an electrospray ionization source in negative mode, and centroid data acquisition. Nitrogen was used as the curtain gas, ion source gas 1 and 2 and collision gas. Ion spray voltage was −4,200 V and the temperature was set to 480 °C. The FBP declustering potential was −45 V. Selection of FBP (parent ion) in the first quadrupole was at m/z of 339.1. In the second quadrupole, collision energies of −20 V, −11 V and −40 V were used to generate the three product ions: 176.9 m/z (putative formula $H_3P_2O_7^-$), 97 m/z ($H_2PO_4^-$) and 241 m/z (putative formula $C_6H_{10}O_8P^-$), respectively, which were detected in the third quadrupole. The characteristic ratios of the three product ions confirmed the identity of the FBP peak, and the $H_3P_2O_7^-$ product ion (m/z 176.9) was used for quantification of FBP. For the $[^{13}C_6]$FBP internal standard, selection of the parent ion in the first quadrupole was at 345.1 m/z, and the product ions detected in the third quadrupole were at 176.9 m/z, 97 m/z and 247 m/z. FBP was quantified by comparison of the integrated MS-Q3 signal peak area with a calibration curve obtained using authentic FBP standards, and the signal from the $[^{13}C_6]$FBP internal standard was used to correct for matrix and ion suppression effects. Peak integration and calculations were done using MultiQuant software (Sciex).

#### Protein expression and purification

To express GST-fused Rac1, *E. coli* BL21 (DE3) were grown in 2×YT medium at 160 rpm at 37 °C to an optical density at 600 nm of 0.8–0.9.

The culture was cooled to 20 °C, and protein expression was induced by applying IPTG to a final concentration of 0.5 mM. The cells were collected at 20 h post induction. To purify GST–Rac1, the cells were resuspended in lysis buffer (50 mM TRIS pH 7.5, 150 mM NaCl, 5 mM DTT, 0.5 mM EDTA). The lysate was bound to GST-Bind resin (Novagen), washed three times with lysis buffer and cleaved off the beads with GST-fused HRV3-C protease. After cleavage, the flow-through fractions were collected and concentrated with an Amicon 10 kDa MWCO Spin Column (Merck) for further purification. The final purification step was performed by size-exclusion chromatography (Superdex 75, Cytiva) with a size-exclusion chromatography buffer (20 mM TRIS pH 7.5, 150 mM NaCl, 5 mM DTT).

His$_{10}$–eGFP-tagged RCC2 full-length was expressed in Sf21 insect cells, using SF900-II serum-free medium (Thermo Fisher Scientific). Sf21 cells (800 ml) grown to a density of $1$–$1.5 × 10^6$ cells ml$^{-1}$ were infected with 10 ml amplified baculovirus encoding the His$_{10}$–eGFP-tagged RCC2. Cells were collected around 72 h post virus infection. The cell viability was around 70%. Cell pellets were stored frozen at −20 °C until purification. To purify His10–eGFP-tagged RCC2 full-length, cell pellets from each 200 ml culture were resuspended in 35 ml lysis buffer (50 mM TRIS pH 8.0, 300 mM NaCl, 10 mM imidazole, 1 mM DTT, 0.5% Triton X-100, 1 tablet per 50 ml of protease inhibitor cocktail), sonicated for 1 min (1 s pulse on, 5 s pulse off) and centrifuged for 20 min at 87,200$g$ at 4 °C. Each 50 ml of supernatant were incubated with 0.5 ml Ni-NTA beads (Merck) on a rotating wheel for 1 h at 4 °C. Beads were collected in a plastic open column (Bio-Rad), washed with 10 ml lysis buffer and then washed with 50 ml wash buffer (50 mM TRIS pH 7.5, 300 mM NaCl, 20 mM imidazole, 1 mM DTT). RCC2 was eluted with 4-5 ml elution buffer (20 mM TRIS pH 7.5, 300 mM NaCl, 300 mM imidazole, 5 mM DTT). Proteins were further purified on a Superdex 200 gel filtration column (Cytiva) at 4 °C with SEC buffer (20 mM TRIS pH 7.5, 300 mM NaCl). Protein samples were concentrated to about 3 mg ml$^{-1}$ for Rac1 and 2.5 mg ml$^{-1}$ for His$_{10}$–eGFP-tagged RCC2 full-length. Samples were flash frozen in liquid nitrogen and stored at −80 °C.

## Limited proteolysis mass spectrometry

A total of $5 × 10^7$ U-2 OS cells were lysed in 100 mM HEPES, 150 mM KCl, 1 mM MgCl$_2$, pH 7.6 with a syringe needle. Residual cell debris was collected by centrifugation for 4 min at 16,000$g$ at 4 °C. Protein concentration was determined by bicinchoninic acid assay (Pierce BCA Protein Assay Kit, Thermo Fisher Scientific) according to the manufacturer's instructions. Endogenous metabolites were removed by size exclusion, and lysates were exposed to the designated FBP concentrations (5 mM, 2.5 mM, 1.25 mM, 0.625 mM, 0.3125 mM, 0.156 mM and 0.078 mM) for 10 min at room temperature. Samples were treated with proteinase K (PK) from *Tritirachium album* (Sigma-Aldrich) in a 1:100 (enzyme to substrate, E:S) ratio and incubated for 4 min at 25 °C. For experiments involving FBP or F2,6BP addition to RCC2, 50 µg purified RCC2 were exposed individually to the designated concentrations of the two metabolites (1.25 mM, 0.625 mM, 0.3125 mM, 0.156 mM, 0.078 mM and 0 mM). For these experiments with recombinant RCC2, PK was added in a 1:10,000 E:S ratio. The PK digestion was stopped by 5 min of incubation at 98 °C and the subsequent addition of sodium deoxycholate (DOC) to a final concentration of 5%. Samples were treated with 5 mM Tris(2-carboxyethyl) phosphine (TCEP) for 30 min at 37 °C and then with 20 mM iodoacetamide for 40 min in the dark at room temperature. The concentration of DOC in the alkylated protein samples was diluted to 1% with 0.1 M ammonium bicarbonate before protease digestion. The samples were digested first by Lysyl endopeptidase (Wako Chemicals) for 4 h, and then by sequence-grade trypsin (Promega) for 16 h, both in 1:100 E:S ratio at 37 °C. The digestion was stopped by adding formic acid to a final concentration of 2%. DOC in the peptide solution was removed by centrifuging twice for 10 min at 20,000$g$. The peptide solution was cleaned-up with C18 columns, either Sep Pak C18 columns (Waters) or

BioPure SPN columns (NEST Group). Peptides were eluted by 35% acetonitrile containing 0.1% formic acid and dried in a vacuum concentrator. The dried peptide was reconstituted in 3% acetonitrile containing 0.1% formic acid before injection into the mass spectrometer.

Samples were analysed with an Orbitrap Q Exactive HF-X mass spectrometer (Thermo Fisher Scientific) equipped with a nano-electrospray ion source and a nano-flow liquid chromatography (LC) system (Easy-nLC 1000, Thermo Fisher Scientific). The LC system was operated with buffer A (3% acetonitrile containing 0.1% formic acid) and buffer B (90% acetonitrile containing 0.1% formic acid). Peptides were separated with a 120-min nonlinear gradient through a fused silica capillary column (20 cm, 75-µm inner diameter), in-house packed with 1.9-µm C18 beads (Dr. Maisch Reprosil-Pur 120). Samples were analysed in data-independent acquisition (DIA) mode. The DIA method contained a survey MS1 scan from 350 to 1,650 $m/z$ at a resolution of 120,000 followed by the acquisition of MS2 in 40 DIA isolation windows. MS2 was acquired at a resolution of 30,000. The 40 DIA windows were segmented with a 0.5 $m/z$ overlap in between. In the case of the LiP experiment on purified RCC2, peptides were analysed on an Orbitrap Exploris 480 mass spectrometer (Thermo Fisher Scientific) equipped with a nano-electrospray ion source and a nano-flow LC system (Vanquish Neo UHPLC System, Thermo Fisher Scientific). Buffer A, buffer B and the analytical column in the LC configuration stayed the same as previously described, but the separation LC gradient was 30 min. A DIA method was used for analysing the samples. The DIA method contained a survey MS1 scan from 350 to 1,650 $m/z$ at a resolution of 60,000 followed by the acquisition of MS2 in 12 DIA isolation windows. MS2 was acquired at a resolution of 30,000. The 12 DIA windows were with variable overlaps and widths.

To construct the spectra library, a data-dependent acquisition method was used to analyse the fractionated samples and five other randomly selected samples. The data-dependent acquisition method contained a survey MS1 scan from 350 to 1,650 $m/z$ at a resolution of 60,000 followed by the acquisition of MS2 of top 15 at a resolution of 15,000. The mass spectrometry proteomics data have been deposited to the ProteomeXchange Consortium via the PRIDE[93] partner repository with the dataset identifier PXD043853 (ref. 43).

The responding peptides were modelled into the crystal structures of Rac1 (PBD ID: 1MH1)[46] and RCC2 (PBD ID: 5GWN)[48] using PyMOL[94] (version 3.9). The Rac1–RCC2 complex was modelled by aligning the crystal structures of Rac1 and RCC2 into an existing cocrystal of Ran and RCC1 (PBD ID: 1I2M)[95] as previously described[48]. In addition, peptide responses of purified RCC2 were modelled into a predicted structure (AF-Q9P258-F1) generated by AlphaFold[96].

## Biochemical competition assay

HEK293T cells, transfected to express either GFP–RCC2 or eGFP, were lysed in a buffer containing 50 mM Tris–HCl pH 8.0, 150 mM NaCl, 2 mM EDTA, 0.5% NP-40, 1 mM PMSF and protease inhibitor cocktail (Merck). After centrifugation at 17,000$g$ for 5 min at 4 °C, 10 mM FBP were added to the supernatant of the FBP-sample, and the lysates were left to incubate for 30 min on a rotator at 4 °C. Afterwards, all samples were incubated with 30 µl of magnetic GFP-trap beads (ChromoTek) for 3 h at 4 °C under constant rotation. A total of 30 µg GST-Rac1 (bacterially expressed and purified as described in the 'Protein expression and purification' section using Pierce Glutathione-Agarose and eluted with 50 mM Tris–HCl pH 8.0, 150 mM NaCl and 5 mM reduced glutathione) were added to all lysates and left to rotate for 1 h at 4 °C. The beads were then washed five times for 5 min in wash buffer containing 50 mM Tris–HCl pH 8.0, 150 mM NaCl, 2 mM EDTA and 0.1% NP-40 before elution. Eluted proteins were analysed by immunoblotting.

## Cell cycle profiling via flow cytometry

On the fourth day of the knockdown protocol (see 'siRNA and plasmid transfection'), 1.5 million U-2 OS cells treated with siCtrl, siPFK,

siALDOA or siGAPDH siRNA were seeded into 10-cm dishes to obtain a confluency of approximately 70% on the following day where cell cycle profiling was conducted as follows:

A total of 10 μM EdU (Merck) were added to the cells which were left for 30 min at 37 °C to label cells in the S phase. After trypsinization, the cells were washed in PBS, centrifuged at 400g for 5 min at room temperature and fixed and permeabilized in commercial fixation/permeabilization solution (1:3 fixation/permeabilization concentrate in fixation/permeabilization diluent, Invitrogen) at room temperature for 15 min. Then, 1 ml PBS was added, and after centrifugation at 400g for 5 min at room temperature, the cell pellet was resuspended in 100 μl commercial Perm wash buffer (1:10 permeabilization buffer in $H_2O$, Invitrogen). EdU-containing cells were fluorescently labelled with Eterneon-Red (baseclick GmbH) via incubation with ClickIt Reaction Mix (1 μM Eterneon-Red, 10 mM Tris pH 8.8, 500 μM $CuSO_4$ and 100 mM ascorbic acid in PBS) for 20 min. After two washes in Perm wash buffer, the cells were stained with 0.2 μg $ml^{-1}$ DAPI in PBS for at least 5 min. Eterneon-Red and DAPI intensities were measured via flow cytometry.

In the Attune NxT Flow Cytometer (Invitrogen), 20,000 cells were measured per sample. Subsequent analysis was performed with the FlowJoTM v10.8 Software (BD Life Sciences). Unspecific signals were excluded, and the actual cell population was defined via gating of the forward and side scatter pulse areas. Cells with an Eterneon-Red pulse area above 350 were classified as cells in the S phase. Cells with a pulse area below 350 were further categorized as cells in either G0/G1 phase (low DAPI pulse area) or G2/M phase (high DAPI pulse area), as follows: borders were defined around the two visible subpopulations. The centre between the two inner borders was set as the threshold between low and high DAPI pulse areas.

### Testing Rac1 activity in response to glycolytic flux

hTERT RPE-1 cells were transferred to glucose-and FBS-free DMEM (Gibco) supplemented with 1 mM sodium pyruvate (Gibco) and 1× GlutaMAX (Gibco) as alternative energy sources (glucose-free medium). After 6 h at 37 °C and 5% $CO_2$, the cells were detached with TrypLE Express (Gibco), collected in glucose-free medium supplemented with 0.1% BSA and washed twice with this medium. Cells were counted and either resuspended in glucose-free medium or in the same medium containing 1 mM glucose. For the spreading assay, 10,000 cells were seeded onto poly-L-lysine (Sigma)–coated μ-slide eight-well glass bottom chambers (Ibidi), where they were allowed to adhere for 2 h. The cells were then fixed and stained with Alexa Fluor 568 phalloidin and DAPI (as described in 'Immunofluorescence, confocal imaging and image analysis'). Images were acquired using a Nikon spinning disk confocal microscope using a 20× objective for acquiring images for quantification and a 60× objective for taking representative images. The cell area was quantified using CellProfiler[85,97], applying propagation-based segmentation to delineate phalloidin-labelled cell boundaries using the DAPI signal as the primary seed[98]. A minimum of 250 cells was analysed per condition.

For the determination of Rac1 activity, $1 × 10^6$ cells were seeded into 6-cm dishes and left to adhere for 2 h at 37 °C and 5% $CO_2$. Cells were then processed for effector pull-down assay as described in 'Effector pull-down assays'.

### Statistics and reproducibility

No statistical method was used to predetermine sample sizes, but our sample sizes are similar to those reported in previous publications[20,24]. Cell culture dishes were allocated in a randomized manner to treatment conditions. No data were excluded from the analyses. The investigators were usually not blinded to allocation during experiments and outcome assessment. However, image analysis was performed in an automated fashion to avoid bias. Experiments were successfully replicated independently. Data were collected in Microsoft Excel (version 16.104.1). Statistical analyses were performed with GraphPad Prism (version 9 and 10).

All graphs depict mean ± s.e.m., except for Fig. 4b–d,f–h, which show median and interquartile ranges. All measurements were taken from distinct samples. The statistical significance between two experimental groups was assessed using either two-sided Student's t-test or Mann–Whitney test. For comparisons of more than two groups, a one-way analysis of variance (ANOVA) or an repeated-measures ANOVA, each followed by either a Dunnett's or Tukey's post-test was used, as indicated in the figure legends. For the data in Extended Data Figs. 3 and 7, a mixed-effects analysis was used. For normalized data, a one-sample t-test was used. Data were tested for normal distribution with the Shapiro–Wilk normality test and met the assumptions of the statistical tests used. The number of independent experiments (n) is indicated in the corresponding figure legends. The level of significance is indicated in the figures as follows: *$P < 0.05$, **$P < 0.01$, ***$P < 0.001$, ****$P < 0.0001$.

### Reporting summary
Further information on research design is available in the Nature Portfolio Reporting Summary linked to this article.

### Data availability
Mass spectrometry data have been deposited in ProteomeXchange with the accession code PXD043853 (ref. 43). Data on the genome-wide screen can be found in the Supplementary Table in Tabs 1 and 2. All other data supporting the findings of this study are available from the corresponding author on reasonable request. Source data are provided with this paper.

### Code availability
The code that was generated for image analysis including Python jupyter notebooks, Fiji image processing and analysis scripts and KNIME workflows has been published via Zenodo at https://doi.org/10.5281/zenodo.18327217 (ref. 99) and are made available under the MIT license.

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

## Acknowledgements

We thank C. Bahnik, M. Mühlbauer, D. Löwe, S. Zillmann and S. Kleissle for technical assistance. We thank G. L. Dornan for help with PyMol. We thank H.-P. Rahn for help with single cell sorting. We thank E. Richling, S. Stegmüller and Y. Schermer for assistance with sample preparation for FBP measurements. We thank Z. Storchová and A. Becker for assistance with cell cycle profiling. This research was supported by a stipend of the Sonnenfeld Foundation to LH and by grants of the Deutsche Forschungsgemeinschaft (DFG, German Research Foundation) (grant nos. 516983053/ MA4735/4-1; 461336323/ INST248/342-1 FUGG to T.M.). R.F. and J.E.L. were financially supported by the Max Planck Society.

## Author contributions

L.H. and T.M. conceived the project and designed the experiments. L.H. performed most microscopy and biochemical experiments with U-2 OS cells. M.D. performed microscopy, biochemical experiments and cell cycle profiling with U-2 OS and other cell lines. F.L. assisted with biochemical experiments and performed spreading assays with RPE-1 cells. K.L., J.P.v.K., M.L., C.S. and L.H. designed and conducted the genome-wide screen. L.H. and C.S. analysed the data. C.S. performed the analysis of FA dynamics. Y.H.H. and I.P. performed and analysed the LiP–MS. W.T.L. and V.H. provided purified proteins. R.F. and J.E.L. contributed LC–MS/MS-based FBP measurements. L.H., V.H. and T.M. prepared the manuscript with input and approval from all authors.

## Funding

## Competing interests

The authors declare no competing interests.

## Additional information

**Extended data** is available for this paper at https://doi.org/10.1038/s41556-026-01911-1.

**Correspondence and requests for materials** should be addressed to Tanja Maritzen.

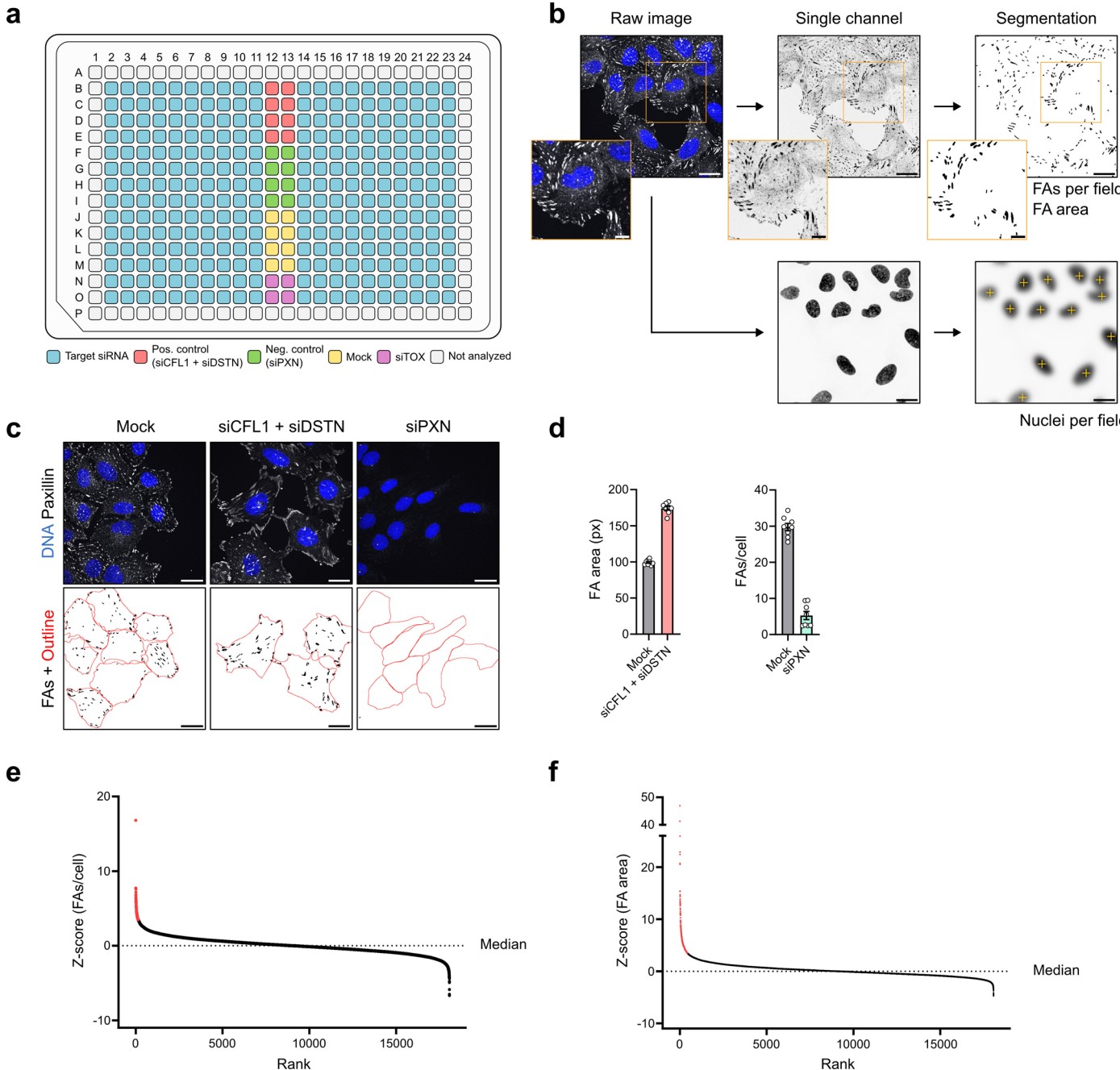

**Extended Data Fig. 1 | Genome-wide screen for FA regulators. a**, Scheme of the plate layout for the genome-wide screen. **b**, Image analysis workflow for the genome-wide screen. FA number and area (top) and nuclei number (bottom) were quantified using an automated KNIME pipeline. Detailed descriptions can be found in the Methods section. Boxed areas show zooms. **c, d**, Validation of positive and negative controls. **c**, Representative confocal images of paxillin-labeled FAs in U-2 OS cells treated with siRNAs targeting either cofilin (CFL1) and destrin (DSTN) or paxillin (PXN), which served as assay quality controls. FA segmentation and cell outlines (red) are shown below. **d**, Quantification of FA area (in pixels) or FAs per cell shown in **c**. Data represent mean ± SEM; n = 8 wells of one plate. **e, f**, Scatter plots of all siRNAs ranked according to their Z-score in either FAs per cell or FA area. Dotted lines represent median values for each plot. Red dots indicate siRNAs that were considered hits based on their Z-scores (>4.2 for FAs per cell or >3.5 for FA area). Scale bars, 25 μm. Screen data can be found in the Supplementary Table in Tabs 1 and 2.

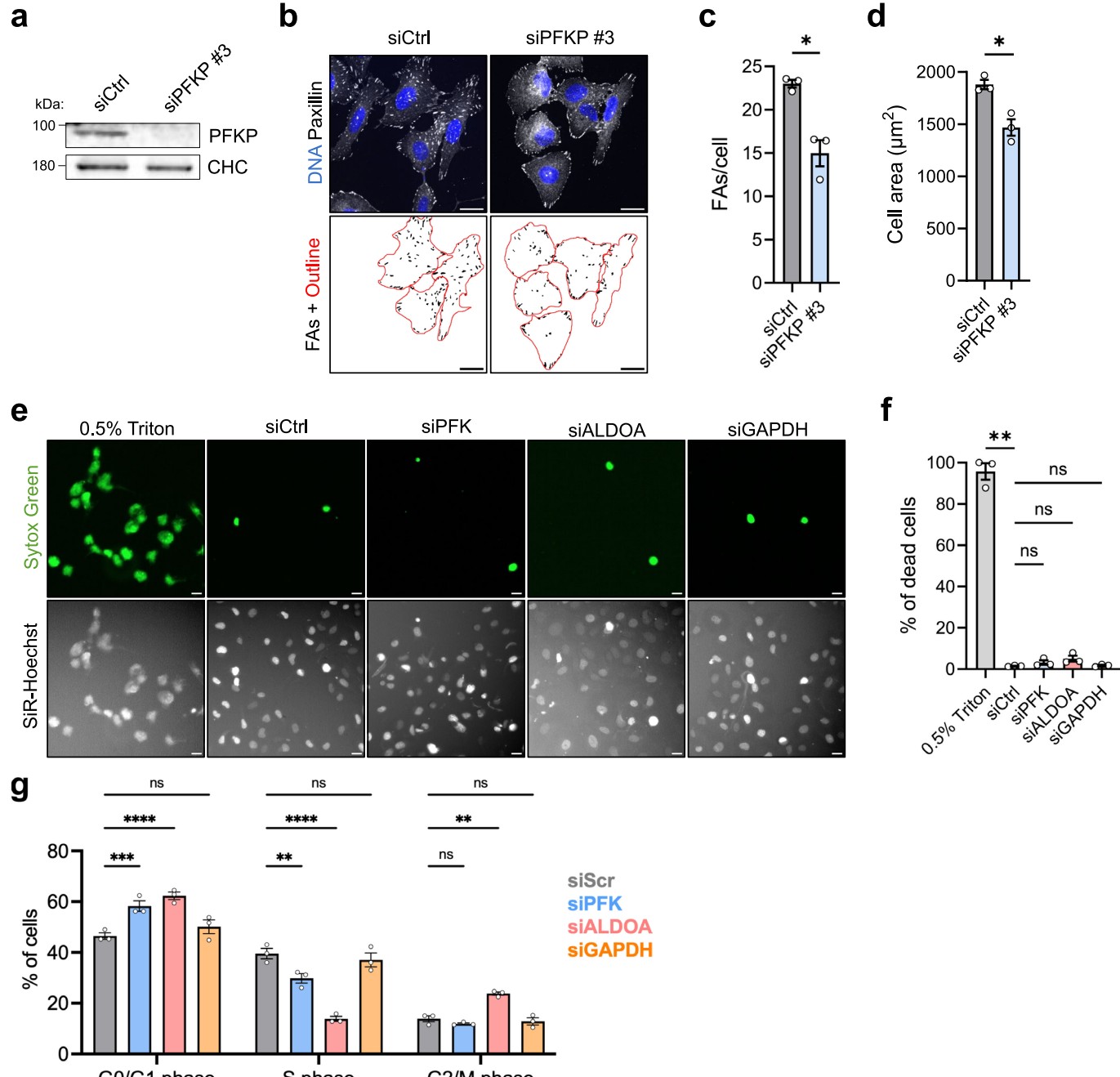

**Extended Data Fig. 2 | FA phenotypes of glycolytic enzyme depletion are neither off-target effects nor a consequence of compromised cell viability or altered cell cycle distribution. a-d**, Loss of PFK (isoform PFKP) in U-2 OS cells by an alternative siRNA also decreases FA numbers and cell size. **a**, Efficient depletion of PFK in U-2 OS cells with an siRNA targeting PFKP. Immunoblot of U-2 OS cells treated with indicated siRNAs. CHC was used as loading control. N = 1 independent experiment. **b**, Representative confocal images of paxillin-labeled FAs in U-2 OS cells treated with indicated siRNAs. FA segmentation and cell outlines (red) are shown below. Scale bars, 25 μm. **c**, **d**, Quantification of FAs per cell and cell area shown in **b**. Data represent mean ± SEM; n = 3 independent experiments; two-sided paired t-test (P value (**c**) = 0.0175; P value (**d**) = 0.0429). **e-f**, U-2 OS cells depleted of aldolase, PFK or GAPDH do not show increased cell death. Cells treated with the indicated siRNAs were stained with the cell-permeable DNA stain SiR-Hoechst and the membrane-impermeable DNA dye SYTOX Green which can only enter cells with compromised cell membrane integrity. Cells treated with the detergent Triton X-100 to impair membrane integrity were used as positive control. **e**, Representative epifluorescence images

of SiR-Hoechst- and SYTOX-labeled U-2 OS cells treated with the indicated siRNAs. Scale bars, 25 μm. **f**, Quantification of percentage of SYTOX-positive cells. Data represent mean ± SEM; n = 3 independent experiments; RM one-way ANOVA with Dunnett's post-test (P values: (siCtrl vs. 0.5% Triton) = 0.0039; (siCtrl vs. siPFK) = 0.5294; (siCtrl vs. siALDOA) = 0.2328; (siCtrl vs. siGAPDH) = 0.798). **g**, U-2 OS cells depleted of aldolase are to a lower extent in S phase. Cells treated with the indicated siRNAs were analyzed by FACS for their cell cycle distribution after 30 min incubation with 10 μM EdU to label cells in S phase (gating strategy illustrated in Supplementary Figure 1). Quantification of the percentage of cells in the indicated cell cycle phases. Data represent mean ± SEM; n = 3; two-way ANOVA with Dunnett's post-test (P values: G0/G1 phase: (siScr vs. siPFK) = 0.0002; (siScr vs. siALDOA) = <0,0001; (siScr vs. siGAPDH) = 0.3297; S phase: (siScr vs. siPFK) = 0.0015; (siScr vs. siALDOA) = <0,0001; (siScr vs. siGAPDH) = 0.6219; G2/M phase: (siScr vs. siPFK) = 0.7589; (siScr vs. siALDOA) = 0,0012; (siScr vs. siGAPDH) = 0.9552). Ns, not significant; *p < 0.05, **p < 0.01, ***p < 0.001, ****p < 0.0001.

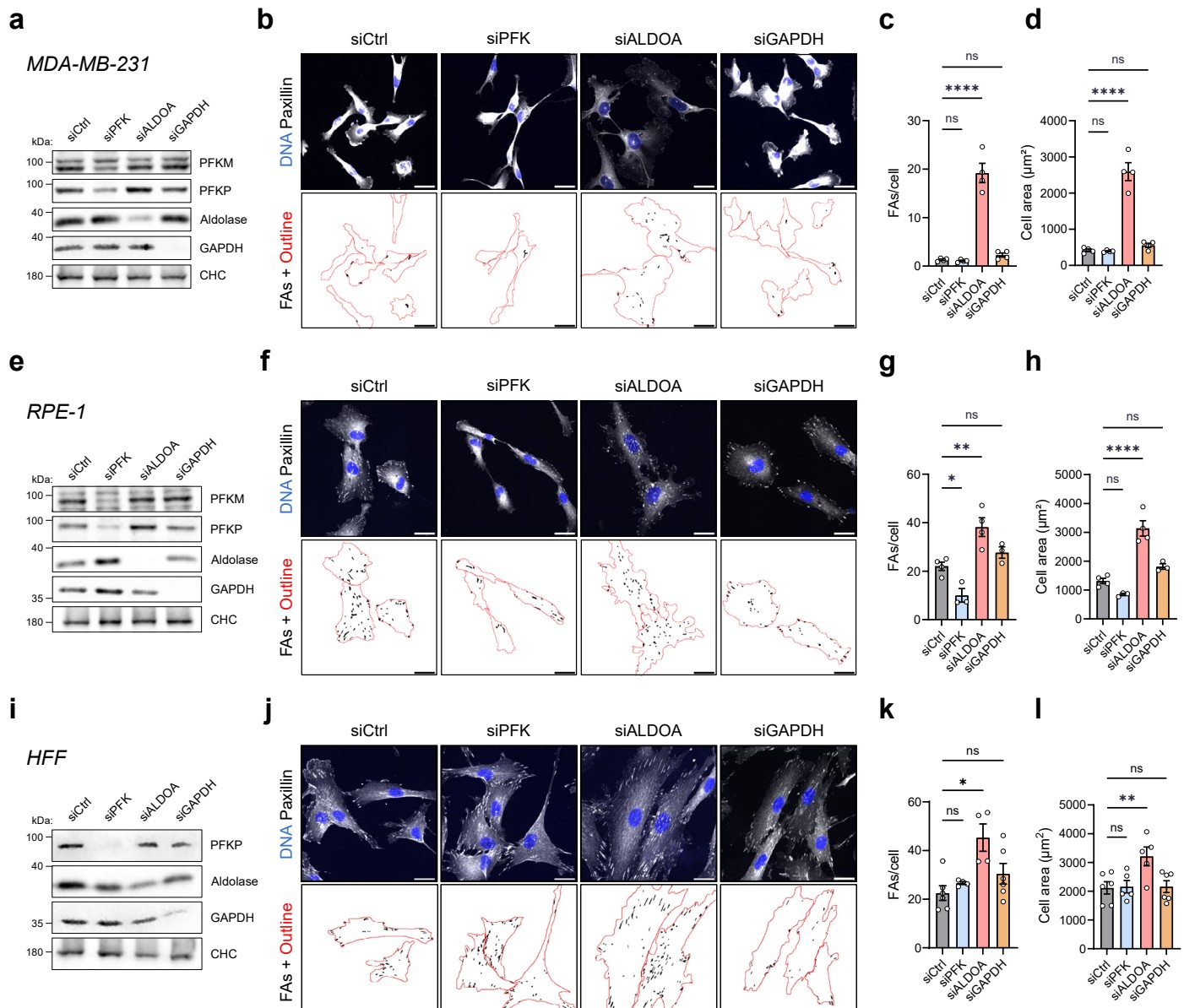

**Extended Data Fig. 3 | Glycolytic control of FAs and cell area is reproducible in multiple cell lines. a**, Efficient depletion of glycolytic enzymes in MDA-MB-231 cells. Immunoblot of MDA-MB-231 cells treated with indicated siRNAs. CHC was used as loading control. N = 1 independent experiment. **b-d**, In the breast cancer cell line MDA-MB-231, loss of aldolase causes a striking increase in FA number and cell area. **b**, Representative confocal images of paxillin-labeled FAs in MDA-MB-231 cells treated with indicated siRNAs. FA segmentation and cell outlines (red) are shown below. **c, d**, Quantification of FAs per cell and cell area shown in **b**. Data represent mean ± SEM; n = 4 independent experiments; One-way ANOVA with Dunnett's post-test (*P* values (**c**): (siCtrl vs. siPFK) = 0.9969; (siCtrl vs. siALDOA) = <0.0001; (siCtrl vs. siGAPDH) = 0.8386; *P* values (**d**): (siCtrl vs. siPFK) = 0.9978; (siCtrl vs. siALDOA) = <0.0001; (siCtrl vs. siGAPDH) = 0.7936). **e**, Efficient depletion of glycolytic enzymes in RPE-1 cells. Immunoblot of RPE-1 cells treated with indicated siRNAs. CHC was used as loading control. N = 1 independent experiment. **f-h**, In the non-malignant cell line RPE-1 aldolase depletion also causes an increased FA number and cell size. **f**, Representative confocal images of paxillin-labeled FAs in RPE-1 cells treated with indicated siRNAs. FA segmentation and cell outlines (red) are shown below. **g, h**, Quantification of FAs per cell and cell area shown in **f**. Data represent

mean ± SEM; n = 4 (siCtrl, siALDOA) and n = 3 (siPFK, siGAPDH) independent experiments; One-way ANOVA with Dunnett's post-test (*P* values (**g**): (siCtrl vs. siPFK) = 0.0407; (siCtrl vs. siALDOA) = 0.0048; (siCtrl vs. siGAPDH) = 0.4303; *P* values (**h**): (siCtrl vs. siPFK) = 0.1966; (siCtrl vs. siALDOA) = <0.0001; (siCtrl vs. siGAPDH) = 0.1624). **i**, Efficient depletion of glycolytic enzymes in HFF cells. Immunoblot of HFF cells treated with indicated siRNAs. CHC was used as loading control. N = 1 independent experiment. **j-l**, In the non-malignant cell line HFF aldolase depletion also causes an increased FA number and cell size. **j**, Representative confocal images of paxillin-labeled FAs in HFF cells treated with indicated siRNAs. FA segmentation and cell outlines (red) are shown below. **k, l**, Quantification of FAs per cell and cell area shown in **j**. Data represent mean ± SEM; n = 6 (k, l: siCtrl, siGAPDH) or 5 (l: siPFK, siALDOA) or 4 (k: siPFK, siALDOA) independent experiments; statistical testing by mixed-effects model (REML) with Dunnett's multiple comparison test (*P* values (k): (siCtrl vs. siPFK) = 0.6632; (siCtrl vs. siALDOA) = 0.0303; (siCtrl vs. siGAPDH) = 0.169; *P* values (l): (siCtrl vs. siPFK) = 0.9945; (siCtrl vs. siALDOA) = 0.005; (siCtrl vs. siGAPDH) = 0.9613). Ns, not significant; *p < 0.05, **p < 0.01, ****p < 0.0001. Scale bars, 25 µm.

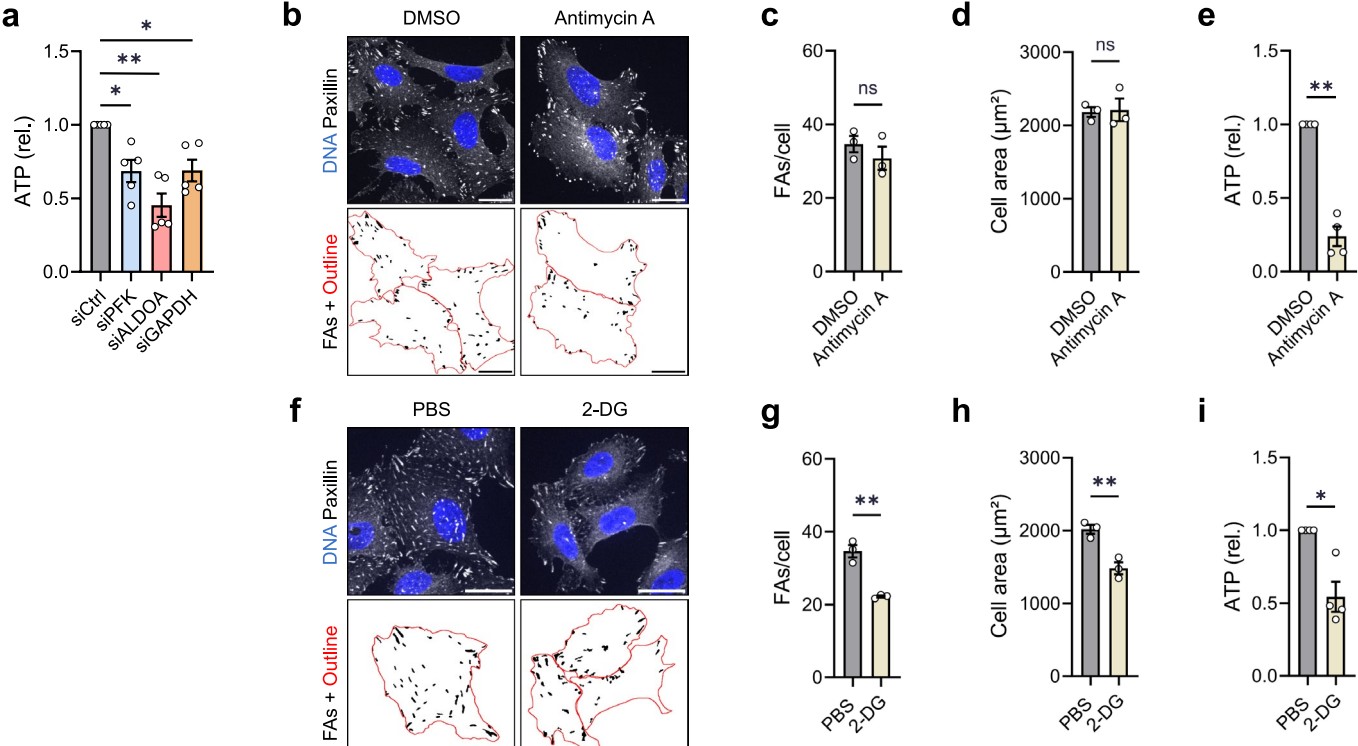

**Extended Data Fig. 4 | Glycolytic control of FAs and cell area is independent of ATP. a**, Loss of PFK, aldolase or GAPDH results in lower ATP levels. Relative ATP levels measured in U-2 OS cells treated with indicated siRNAs. Data were normalized to siCtrl and represent mean ± SEM; n = 5 independent experiments; two-sided One sample t-test (*P* value (siPFK) = 0.014; (siALDOA) = 0.0024; (siGAPDH) = 0.0136). **b-e**, Lower ATP levels due to inhibition of oxidative phosphorylation do not affect FA numbers or cell area. **b**, Representative confocal images of paxillin-labeled FAs in U-2 OS cells treated with DMSO or 1 µM Antimycin A for 48 h. FA segmentation and cell outlines (red) are shown below. **c**, **d**, Quantification of FAs per cell and cell area shown in **b**. Data represent mean ± SEM; n = 3 independent experiments; two-sided unpaired Student's t-test (*P* value (**c**) = 0.3716; *P* value (**d**) = 0.8734). **e**, Relative ATP levels measured in U-2

OS cells treated with DMSO or 1 µM Antimycin A for 48 h. Data were normalized to DMSO and represent mean ± SEM. n = 3 independent experiments; two-sided One sample t-test (*P* value = 0.0014). **f-i**, Lower ATP levels due to inhibition of glycolysis phenocopy loss of PFK. **f**, Representative confocal images of paxillin-labeled FAs in U-2 OS cells treated with PBS or 25 mM 2-deoxy-D-glucose (2-DG) for 48 h. FA segmentation and cell outlines (red) are shown below. **g**, **h**, Quantification of FAs per cell and cell area shown in **f**. Data represent mean ± SEM; n = 3 independent experiments; two-sided unpaired Student's t-test (*P* value (**g**) = 0.002; *P* value (**h**) = 0.0069). **i**, Relative ATP levels measured in U-2 OS cells treated with PBS or 25 mM 2-DG for 48 h. Data were normalized to PBS and represent mean ± SEM; n = 3 independent experiments; two-sided One sample t-test (*P* value = 0.0216). Ns, not significant; *p < 0.05, **p < 0.01. Scale bars, 25 µm.

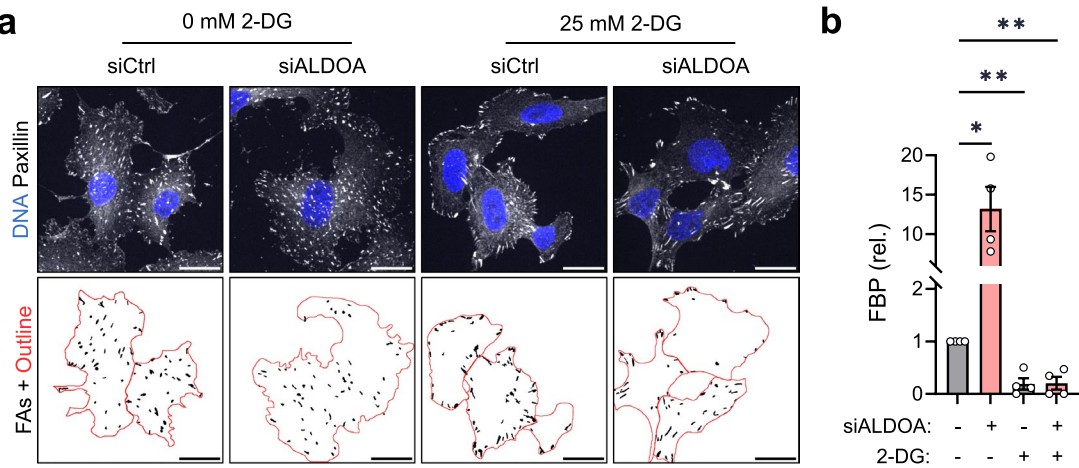

**Extended Data Fig. 5 | Inhibition of upstream glycolysis rescues aldolase loss-of-function phenotypes. a**, Inhibiting glycolysis lowers FA number and cell size to normal levels in aldolase knockdown cells. Representative confocal images of paxillin-labeled FAs in U-2 OS cells treated with indicated siRNAs followed by 48 h treatment with PBS or 25 mM 2-deoxy-D-glucose (2-DG). FA segmentation and cell outlines (red) are shown below. Corresponding quantification is shown in Fig. 3g, h. **b**, Inhibiting glycolysis rescues elevated FBP level in aldolase-depleted cells. Relative FBP levels measured in U-2 OS cells treated with control (-) or aldolase-specific siRNA (ALDOA, +) followed by 48 h treatment with PBS (-) or 25 mM 2-DG. Data were normalized to siCtrl + PBS and represent mean ± SEM; n = 4 independent experiments; two-sided One sample t-test (*P* values: (siALDOA) = 0.0227; (siCtrl+2DG) = 0.005; (siALDOA+2-DG) = 0.007). Ns, not significant; *p < 0.05, **p < 0.01. Scale bars, 25 μm.

**a**

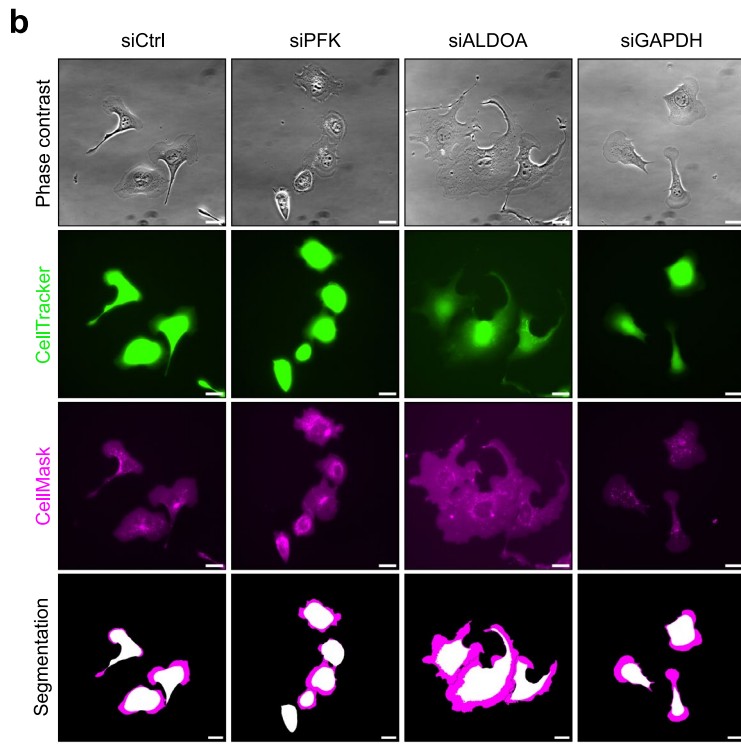

Input, plasma membrane (CellMask)

Cell area

*Gaussian blur, Huang2 threshold*

Individual cell area

Elevated area (white) + Protrusive area (purple)

Input, cytosol (CellTracker)

*Inversion*

CellTracker (inverted)

*Marker-controlled watershed*

Maxima

*Gaussian blur, Maxima detection*

Elevated area

Phase contrast (for comparison)

*Gaussian blur, Huang2 threshold*

**b**

siCtrl | siPFK | siALDOA | siGAPDH

Phase contrast

CellTracker

CellMask

Segmentation

**Extended Data Fig. 6 | See next page for caption.**

**Extended Data Fig. 6 | FBP regulates protrusive area. a**, Example of the image analysis workflow used to quantify protrusive area using an automated Fiji macro. Total cell area was visualized via the plasma membrane dye CellMask. Elevated cell regions were detected by their greater accumulation of the cytoplasmic fluorescent CellTracker dye in comparison to the very thin cellular protrusions. A detailed description of the image analysis procedure can be found in the Methods section. **b**, Representative phase contrast and epifluorescent images of U-2 OS cells treated with indicated siRNAs. Segmentation is shown below. Corresponding quantification is shown in Fig. 5b. Scale bars, 25 μm.

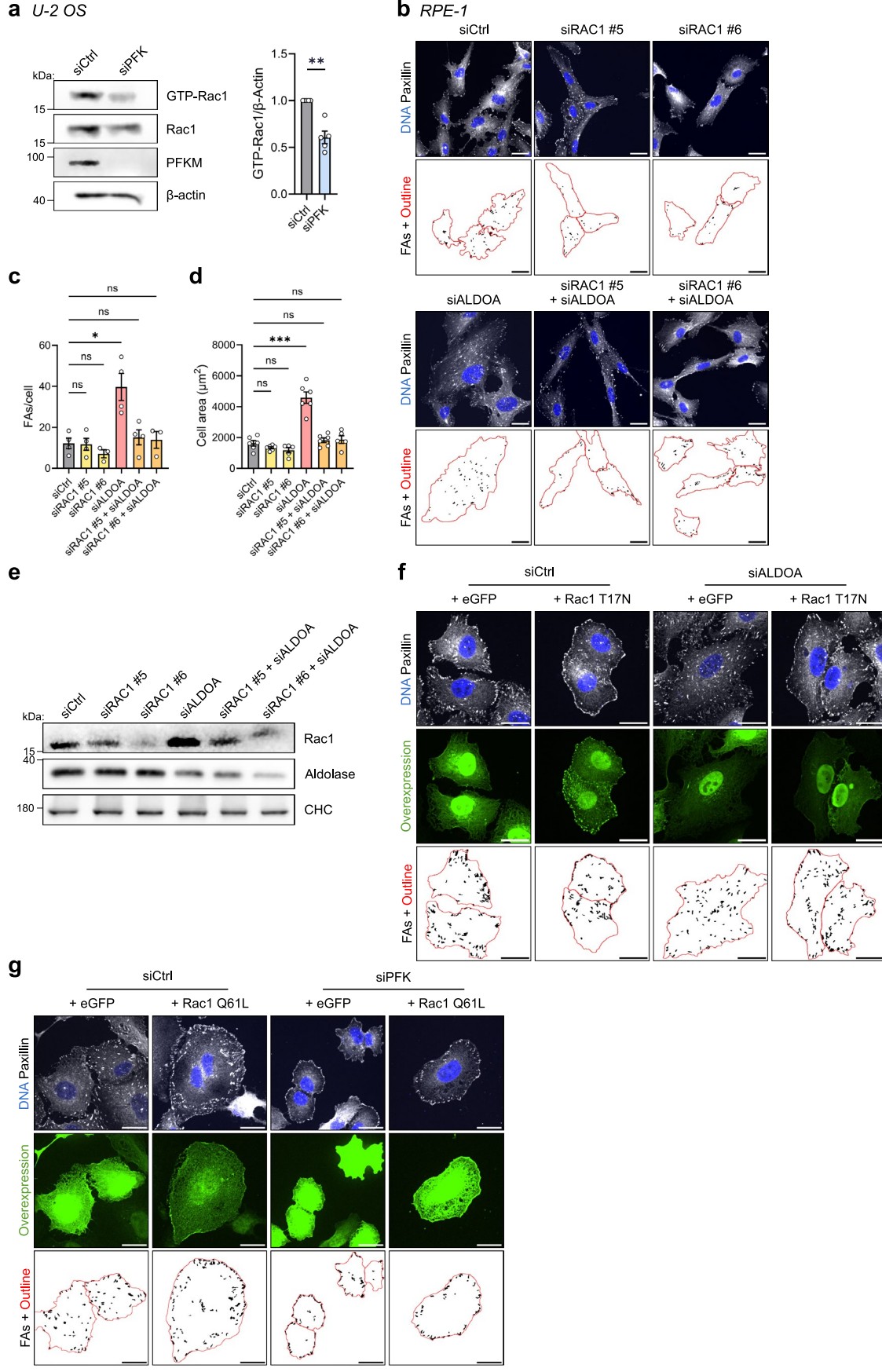

**Extended Data Fig. 7 | See next page for caption.**

**Extended Data Fig. 7 | FBP stimulates Rac1 activation leading to FA assembly and cell spreading. a**, PFK depletion decreases the level of active Rac1. Left: Representative immunoblot of pulldown assay using GST-PAK-PBD beads to detect active Rac1 in lysates from U-2 OS cells treated with indicated siRNAs. β-actin was used as loading control. Right: Quantification of GTP-Rac1 over β-actin. Data were normalized to siCtrl and represent mean ± SEM; n = 5 independent biochemical experiments; two-sided One sample t-test (*P* value = 0.0043). **b-e**, Co-depletion of Rac1 restores FA numbers and cell size upon aldolase knockdown also based on alternative Rac1-targeting siRNAs in RPE-1 cells. **b**, Representative confocal images of paxillin-labeled FAs in RPE-1 cells treated with indicated siRNAs. FA segmentation and cell outlines (red) are shown below. **c, d**, Quantification of FAs per cell and cell area shown in **b**. Data represent mean ± SEM; n = 3 (c: siRAC1#6, siRAC1#6 + siALDOA), 4 (c, all others), 5 (d: siRAC1#6, siRAC1#6 + siALDOA) or 6 (d, all others) independent experiments; statistical testing by mixed-effects model (REML) with Dunnett's multiple comparison test (*P* values (**c**): (siCtrl vs. siRAC1#5) = 0.6235; (siCtrl vs. siRAC1#6) = 0.4587; (siCtrl vs. siALDOA) = 0.0332; (siCtrl vs. siRAC1#5

+ siALDOA) = 0.3327; (siRAC1#6 + siALDOA) = 0.9142; *P* values (**d**): (siCtrl vs. siRAC1#5) = 0.3583; (siCtrl vs. siRAC1#6) = 0.1119; (siCtrl vs. siALDOA) = 0.0006; (siCtrl vs. siRAC1#5 + siALDOA) = 0.8081; (siCtrl vs. siRAC1#6 + siALDOA) = 0.2114). Ns, not significant. **e**, Efficient Rac1 and aldolase depletion in RPE-1 cells. Immunoblot of RPE-1 cells treated with indicated siRNAs. CHC was used as loading control. N = 1 independent experiment. **f**, Expression of dominant-negative Rac1 in aldolase-depleted cells rescues FA number and cell size. Representative confocal images of paxillin-labeled FAs in U-2 OS cells transfected with control or ALDOA-specific siRNAs in combination with either eGFP or myc-Rac1-T17N. FA segmentation and cell outlines (red) are shown below. Corresponding quantification is shown in Fig. 6i, j. **g**, Expression of constitutively active Rac1 in PFK depleted cells elevates FA number and cell size to normal levels. Representative confocal images of paxillin-labeled FAs in U-2 OS cells transfected with control or PFK-specific siRNAs in combination with either eGFP or myc-Rac1-Q61L. FA segmentation and cell outlines (red) are shown below. Corresponding quantification is shown in Fig. 6k, l. Ns, not significant; *p < 0.05, **p < 0.01, ***p < 0.001. Scale bars, 25 µm.

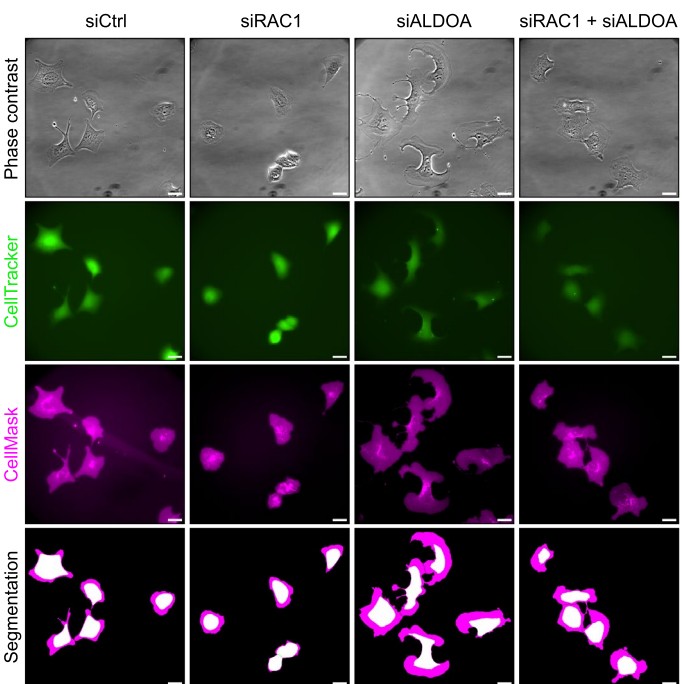

**Extended Data Fig. 8 | Rac1 is required for FBP-mediated cell protrusion.**
Co-depletion of Rac1 in aldolase knockdown cells decreases cell protrusion to normal levels. Representative phase contrast and epifluorescent images of U-2 OS cells treated with indicated siRNAs and labeled with CellTracker and CellMask (see also Extended Data Fig. 6). Segmentation is shown below. Corresponding quantification is shown in Fig. 7b. Scale bars, 25 μm.

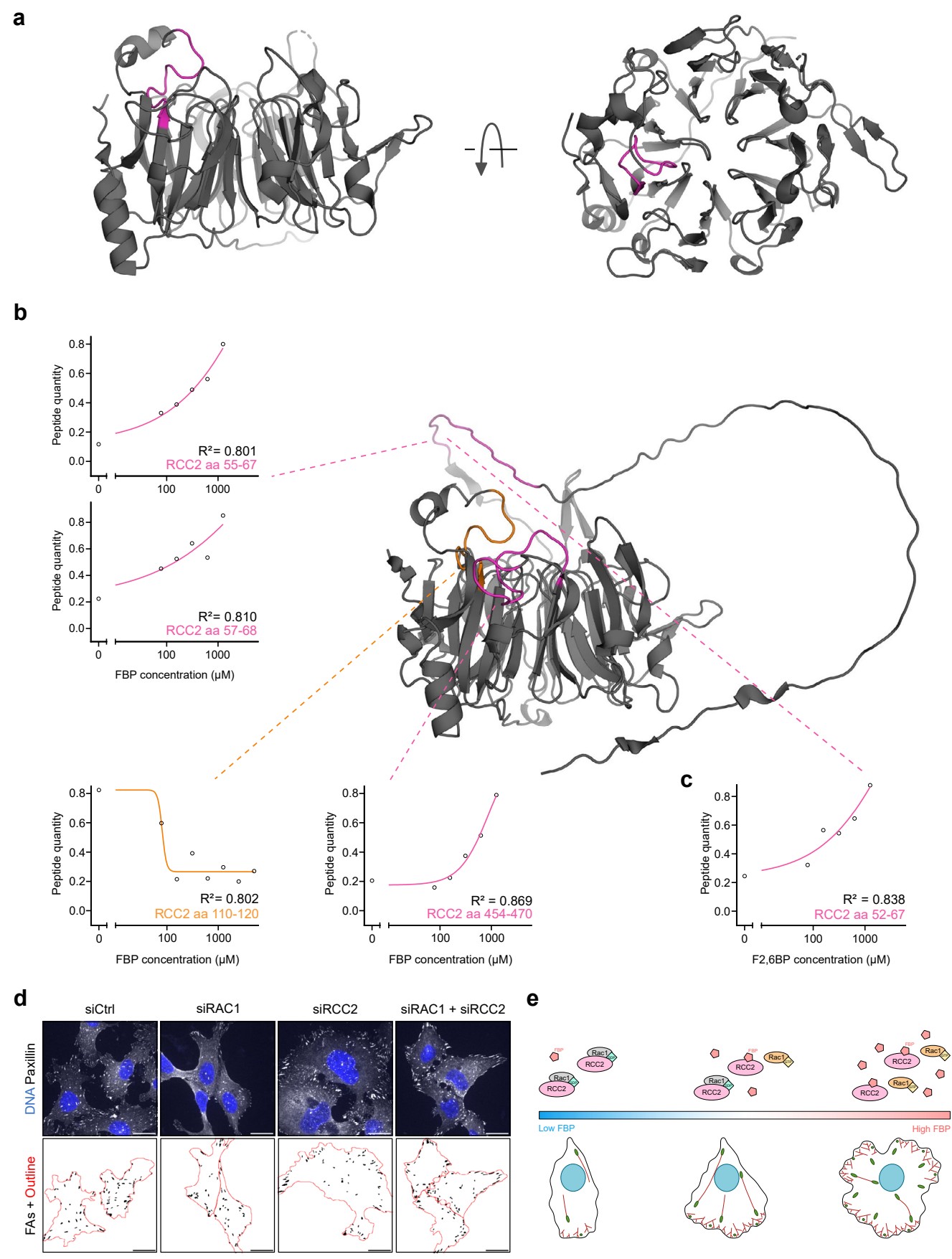

**Extended Data Fig. 9 | See next page for caption.**

**Extended Data Fig. 9 | FBP-mediated Rac1 activation depends on RCC2.**
**a**, Additional views of FBP-responsive peptide (pink, aa 110-120) highlighted in the structure of RCC2 (aa 89-522, PDB ID: 5GWN) (see also Fig. 8b). **b**, Overview of FBP-responsive peptides identified from whole cell lysate (orange, aa 110-120) and purified RCC2 (pink, aa 55-67, aa 57-68 and aa 454-470) highlighted in the predicted structure of full-length RCC2 by AlphaFold (AF-Q9P258-F1).
**c**, Overview of the F2,6BP-responsive peptide identified from purified RCC2 (pink, aa 52-67). **d**, Co-depletion of Rac1 in RCC2 knockdown cells restores FA number and cell area. Representative confocal images of paxillin-labeled FAs

in U-2 OS cells treated with indicated siRNAs. FA segmentation and cell outlines (red) are shown below. Corresponding quantification is shown in Fig. 8i, j.
**e**, Model for the regulation of cell adhesion and protrusion by FBP: under low FBP concentrations, the inhibitory RCC2–Rac1 complex is intact, preventing Rac1 activation and thus resulting in decreased cell protrusion and adhesion. Increased FBP levels destabilize the RCC2–Rac1 complex, likely by direct binding of FBP to RCC2, leading to Rac1 activation, cell spreading and adhesion. Green dots represent FAs, red lines represent F-actin, and blue circles represent nuclei. Scale bars, 25 μm.

|---|---|

# Reporting Summary

## Statistics

For all statistical analyses, confirm that the following items are present in the figure legend, table legend, main text, or Methods section.

| n/a | Confirmed | |
|---|---|---|
| ☐ | ☒ | The exact sample size (*n*) for each experimental group/condition, given as a discrete number and unit of measurement |
| ☐ | ☒ | A statement on whether measurements were taken from distinct samples or whether the same sample was measured repeatedly |
| ☐ | ☒ | The statistical test(s) used AND whether they are one- or two-sided<br>*Only common tests should be described solely by name; describe more complex techniques in the Methods section.* |
| ☒ | ☐ | A description of all covariates tested |
| ☐ | ☒ | A description of any assumptions or corrections, such as tests of normality and adjustment for multiple comparisons |
| ☐ | ☒ | A full description of the statistical parameters including central tendency (e.g. means) or other basic estimates (e.g. regression coefficient) AND variation (e.g. standard deviation) or associated estimates of uncertainty (e.g. confidence intervals) |
| ☐ | ☒ | For null hypothesis testing, the test statistic (e.g. *F*, *t*, *r*) with confidence intervals, effect sizes, degrees of freedom and *P* value noted<br>*Give P values as exact values whenever suitable.* |
| ☒ | ☐ | For Bayesian analysis, information on the choice of priors and Markov chain Monte Carlo settings |
| ☒ | ☐ | For hierarchical and complex designs, identification of the appropriate level for tests and full reporting of outcomes |
| ☒ | ☐ | Estimates of effect sizes (e.g. Cohen's *d*, Pearson's *r*), indicating how they were calculated |

*Our web collection on statistics for biologists contains articles on many of the points above.*

## Software and code

Policy information about availability of computer code

| Data collection | Quantitative values from Western Blots were acquired using the Licor Odyssey software Image Studio Lite (version 5.2 and 6.1). Fluorescence images from Nikon microscopes were acquired with the Nikon NIS Elements software (version 5.21.03). Analysis of microscopy images was performed with KNIME (version 3.7.0), ImageJ/Fiji (version 1.54f), CellProfiler (version 3.1.8) and Python (version 3.8.3). Peptides were modelled using PyMOL (version 3.9). Data were collected in Microsoft Excel files (version 16.104.1). |
|---|---|
| Data analysis | All statistical tests were performed using GraphPad Prism (version 9 and 10). |

For manuscripts utilizing custom algorithms or software that are central to the research but not yet described in published literature, software must be made available to editors and reviewers. We strongly encourage code deposition in a community repository (e.g. GitHub). See the Nature Portfolio guidelines for submitting code & software for further information.

## Data

Policy information about availability of data

All manuscripts must include a data availability statement. This statement should provide the following information, where applicable:
- Accession codes, unique identifiers, or web links for publicly available datasets
- A description of any restrictions on data availability
- For clinical datasets or third party data, please ensure that the statement adheres to our policy

Mass spectrometry data have been deposited in ProteomeXchange with the primary accession code PXD043853 [https://www.ebi.ac.uk/pride/archive/projects/

PXD043853]. Data on the genome-wide screen can be found in the Supplementary table. Numerical source data have been provided in Source Data. All other data supporting the findings of this study are available from the corresponding author on reasonable request.

## Research involving human participants, their data, or biological material

Policy information about studies with human participants or human data. See also policy information about sex, gender (identity/presentation), and sexual orientation and race, ethnicity and racism.

| | |
|---|---|
| Reporting on sex and gender | Sex and gender were not considered in our study design since our study is based on human cell lines and investigates basic cellular processes that are regarded as independent of sex and gender. The sex of the used cell lines was reported as far as it was known. |
| Reporting on race, ethnicity, or other socially relevant groupings | We do not use socially constructed or socially relevant categorization variables in our study. |
| Population characteristics | Our study does not involve human research participants. |
| Recruitment | Our study did not involve the recruitment of participants. |
| Ethics oversight | The study protocol did not need to be approved by an ethics committee using exclusively standard cell lines. |

Note that full information on the approval of the study protocol must also be provided in the manuscript.

# Field-specific reporting

Please select the one below that is the best fit for your research. If you are not sure, read the appropriate sections before making your selection.

☒ Life sciences  ☐ Behavioural & social sciences  ☐ Ecological, evolutionary & environmental sciences

For a reference copy of the document with all sections, see nature.com/documents/nr-reporting-summary-flat.pdf

# Life sciences study design

All studies must disclose on these points even when the disclosure is negative.

| | |
|---|---|
| Sample size | Sample sizes were not chosen based on pre-specified effect size, as the effect sizes were not known beforehand, but had to be discovered as part of the study. Instead, a sufficient number of independent experiments was carried out (as detailed in the figure legends) to allow for statistical testing. |
| Data exclusions | No samples were excluded from analysis. |
| Replication | Experiments were replicated independently. All attempts of replication were successful. |
| Randomization | Cell culture dishes were allocated in a randomized manner to treatment conditions. |
| Blinding | The genome-wide screen with its automated analysis pipeline was objective by design. For subsequent experiments, blinding during the experiment was often not applicable as most knockdowns (PFK, Aldolase, RCC2, Rac1) caused obvious morphological changes in the cells. To avoid potential bias in imaging experiments, cells for analysis were chosen based solely on their nuclear staining and image analysis was performed in an automated fashion. Cells for Western blots were not collected or processed blindly since knowledge of the characteristic of each sample was necessary for data generation. |

# Reporting for specific materials, systems and methods

We require information from authors about some types of materials, experimental systems and methods used in many studies. Here, indicate whether each material, system or method listed is relevant to your study. If you are not sure if a list item applies to your research, read the appropriate section before selecting a response.

## Materials & experimental systems

| n/a | Involved in the study |
|---|---|
| ☐ | ☒ Antibodies |
| ☐ | ☒ Eukaryotic cell lines |
| ☒ | ☐ Palaeontology and archaeology |
| ☒ | ☐ Animals and other organisms |
| ☒ | ☐ Clinical data |
| ☒ | ☐ Dual use research of concern |
| ☒ | ☐ Plants |

## Methods

| n/a | Involved in the study |
|---|---|
| ☒ | ☐ ChIP-seq |
| ☐ | ☒ Flow cytometry |
| ☒ | ☐ MRI-based neuroimaging |

# Antibodies

| | |
|---|---|
| Antibodies used | A complete list of the antibodies used in this study can be found in the Supplementary Table and under the following point. |
| Validation | Paxillin (Abcam, ab32084): specificity was validated by knockdown in this study, see Extended Fig. 1c; Aldolase (Proteintech, 11217-1-AP): specificity was validated by knockdown in this study, see Fig. 2e, Fig. 3b, Fig. 5e, Fig. 6c, Fig. 6d, etc.; PFKM (Proteintech, 55028-1-AP): specificity was validated by knockdown in this study, see Fig. 2e, Fig. 3b; PFKL (Santa Cruz, sc-393713): specificity was validated by knockdown in this study, see Fig. 2e, Fig. 3b; PFKP (Cell Signaling, 8164): specificity was validated by knockdown in this study, Fig. 2e, see Fig. 3b; GAPDH (Merck, G8795): specificity was validated by knockdown in this study, see Fig. 2e, Extended Fig. 3a, e, i; Rac1 (BD Transduction, 610650): specificity was validated by knockdown in this study, see Fig. 6d, Fig. 8h; RCC2 (Novus Biological, NB110-40618): specificity was validated by knockdown in this study, see Fig. 8d, Fig. 8h; Clathrin heavy chain (Abcam, ab21679): specificity was validated by knockdown in https://doi.org/10.1371/journal.pone.0003115; Alpha-tubulin (Merck, T5168): specificity was demonstrated by immunofluorescence and Western blot using lysates from various cell lines according to the manufacturer's website (https://www.sigmaaldrich.com/DE/de/product/sigma/t5168); beta-actin (Merck, A5441): specificity was validated by knockdown in https://doi.org/10.1091/mbc.e11-06-0582; c-Myc (Merck, M4439): specificity was demonstrated by Western blot and ChiP according to manufacturer's website (https://www.sigmaaldrich.com/DE/de/product/sigma/m4439); GFP (Abcam, ab13970): Specificity was demonstated by Western blot and Immunofluorescence after overexpression according to manufacturer's website (https://www.abcam.com/en-us/products/primary-antibodies/gfp-antibody-ab13970) [see also Supplementary Table 4] |

# Eukaryotic cell lines

Policy information about cell lines and Sex and Gender in Research

| | |
|---|---|
| Cell line source(s) | The following cell lines were obtained from the American Type Culture Collection (ATCC): the human sarcoma cell line U-2 OS (ATCC: HTB-96; female), the human embryonic kidney cell line HEK293T (ATCC: CRL-3216; sex not specified), the human breast cancer cell line MDA-MB-231 (ATCC: HTB-26; female), and the telomerase-immortalized human retinal pigment epithelium cell line hTERT RPE-1 (ATTC: CRL-4000; female). Telomerase-immortalized human foreskin fibroblasts (HFF; sex not specified) were a kind gift of Prof. Martin J. Humphries (University of Manchester, UK). |
| Authentication | The cell lines were authenticated by the vendor ATCC by short tandem repeat (STR) marker profiling . The only non-commercial cell line used (HFF) which we obtained from another lab was authenticated based on their typical fibroblast morphology. |
| Mycoplasma contamination | Cell lines were regularly tested for mycoplasma contamination and were not contaminated. |
| Commonly misidentified lines (See ICLAC register) | N/A |

# Flow Cytometry

## Plots

Confirm that:

☒ The axis labels state the marker and fluorochrome used (e.g. CD4-FITC).

☒ The axis scales are clearly visible. Include numbers along axes only for bottom left plot of group (a 'group' is an analysis of identical markers).

☒ All plots are contour plots with outliers or pseudocolor plots.

☒ A numerical value for number of cells or percentage (with statistics) is provided.

## Methodology

| | |
|---|---|
| Sample preparation | 1.5 million U-2 OS cells were seeded into 10 cm dishes. On the next day, their cell cycle was analyzed via the following steps where all centrifugation steps were performed at room temperature and 400 g for 5 min. 10 µM EdU (Merck) were added to the cells for 30 min at 37°C to label the cells in S Phase. After trypsination, the cells were washed in PBS, centrifuged, and |

fixed and permeabilized in Fixation/Permeabilization solution (1:3 Fixation/Permeabilization Concentrate in Fixation/Permeabilization Dilution, InvitrogenTM) at room temperature for 15 min. 1 ml PBS was added, and after centrifugation the cell pellet was resuspended in 100 µl Perm Wash (1:10 Permeabilization Buffer (InvitrogenTM) in H2O). EdU-containing cells were fluorescently labeled with Eterneon-Red (baseclick GmbH) via incubation in the ClickIt Reaction Mix (1 µM Eterneon-Red, 10 mM Tris pH 8.8, 500 µM CuSO4, and 100 mM Ascorbic Acid, in PBS) for 20 min. After two washes in Perm Wash, the cells were stained with 0.2 µg/ml DAPI in PBS for at least 5 min. Eterneon-Red and DAPI intensities were measured via flow cytometry.

| Instrument | Attune™ NxT Flow Cytometer (InvitrogenTM) |
| --- | --- |

| Software | FlowJoTM v10.8 Software (BD Life Sciences) |
| --- | --- |

| Cell population abundance | 20000 cells were measured per condition. The supplementary figure exemplifying the gating strategy also contains information on the abundance of the relevant cell populations within the sorted fractions. |
| --- | --- |

| Gating strategy | The cell population was first defined via the forward and side scatter pulse areas. Via gating of the forward scatter area and forward scatter height, doublets were excluded. After plotting the area of the DAPI signal vs. the area of the EdU signal, the main cell cluster was defined as the final subset of cells. Cells with an Eterneon-Red pulse area above 350 were classified as cells in S phase. Cells with a pulse area below 350 were further categorized as cells in either G0/G1 phase (low DAPI pulse area) or G2/M phase (high DAPI pulse area), as follows: Borders were manually defined around the two visible subpopulations as exemplified in the supplementary figure on the gating strategy. The center between the two inner borders was set as the threshold between low and high DAPI pulse areas. |
| --- | --- |

☒ Tick this box to confirm that a figure exemplifying the gating strategy is provided in the Supplementary Information.

