## [Peer Review File · Nature Cell Biology]

Fructose-1,6-bisphosphate couples glycolytic activity to cell adhesion

Corresponding Author: Professor Tanja Maritzen

Version 0:

Decision Letter:

*Please delete the link to your author homepage if you wish to forward this email to co-authors.

Dear Professor Maritzen,

Thank you for submitting your manuscript, "The signaling metabolite fructose-1,6-bisphosphate couples glycolytic activity to cell adhesion and protrusion" to Nature Cell Biology, and please accept my apologies for the delay in sharing our decision with you. The manuscript has now been seen by 3 referees, who are experts in metabolism (Referee #1); adhesion (Referee #2); and metabolism (Referee #3). As you will see from their comments (attached below), they found this work of potential interest but have raised substantial concerns, which in our view would need to be addressed with considerable revisions before we can consider publication in Nature Cell Biology.

Nature Cell Biology editors discuss the referee reports in detail within the editorial team, including the chief editor, to identify key referee points that should be addressed with priority, as opposed to requests that are overruled as being beyond the scope of the current study. To guide the scope of the revisions, I have listed these points below. Our standard revision period is six months and we are committed to providing a fair and constructive peer-review process, so please feel free to contact me if you would like to discuss any of the referee comments further or if you anticipate any delays or issues addressing the reviews.

I should stress that the referees' concerns point to a premature dataset and concerns regarding the effects of glycolytic flux, FBP on Rac-RCC2 and adhesions would need to be addressed experimentally thoroughly, as reconsideration of the study for this journal and re-engagement of referees will depend on the strength of these revisions. In particular, it would be essential to address the following points:

A- The reviewers questioned the degree to which the phenotypes could be relevant to cell health or physiological contexts. Enzyme knockdowns are not physiologically accepted models to study metabolic fluxes – it is essential to follow Rev#3's recommendations to track glycolytic flux and implicate it in the mechanism proposed. The reviewers additionally asked for measurements of the levels of metabolites as per standard methods in the field and clarifications with regards to the isomers and enzymes at work. These points should be addressed experimentally in full:

Rev#1: "First of all, FBP Aldolase exists as three major isoforms including ALDOA, B and C. These three isoforms share large homological sequences thus thus difficult to distinguish experimentally. However, the distinct tissue and cell type-specific expressions are important for their enzymatic and non-enzymatic functions. In the genome-wide siRNA screen, how did the authors exclude ALDOB or C isoforms in this study?"

"Secondly, Fructose-1,6-bisphosphate (F-1,6-BP) and Fructose-2,6-bisphosphate (F-2,6-BP) are regional-isomers with distinct biological functions. A fluorometric assay kit was used to quantify the amount of F-1,6-BP. It is recommended to use mass spectrometry coupled with liquid chromatography to confirm the identity and quantity of these metabolites. It is such a critically important evidence underpinning the proposed mechanism. Chemically characterised standards of these two isomers must be used for confirmation."

Rev#2 point #3

Rev#3 point #1 in full (ABCD), #2AB, #3

B- Strengthening the mechanism linking FBP to Rac-RCC2 complex formation and Rac activity will be essential:

Rev#2 – please do investigate the impact on Rac activity as suggested by the rev in their general comments; Rev#2 points #1-2

Rev#3 point #4

C- We agree with the reviewers' requests for validation of key data in at least one other cell model and for controls:

Rev#2 point #4
Rev#3 point #5

D- Finally, please pay close attention to our guidelines on statistical and methodological reporting (listed below), as failure to do so may delay the reconsideration of the revised manuscript. In particular, please provide:

- a Supplementary Figure including unprocessed images of all gels/blots in the form of a multi-page pdf file. Please ensure that blots/gels are labeled and the sections presented in the figures are clearly indicated.
- a Supplementary Table including all numerical source data in Excel format, with data for different figures provided as different sheets within a single Excel file. The file should include source data giving rise to graphical representations and statistical descriptions in the paper and for all instances where the figures present representative experiments of multiple independent repeats, the source data of all repeats should be provided.

We would be happy to consider a revised manuscript that would satisfactorily address these points, unless a similar paper is published elsewhere, or is accepted for publication in Nature Cell Biology in the meantime.

- ensure that it conforms to our format instructions and publication policies (see below and <https://www.nature.com/nature/for-authors>).
- provide a point-by-point rebuttal to the full referee reports verbatim, as provided at the end of this letter.
- provide the completed Reporting Summary (found here <https://www.nature.com/documents/nr-reporting-summary.pdf>). This is essential for reconsideration of the manuscript will be available to editors and referees in the event of peer review. For more information see <http://www.nature.com/authors/policies/availability.html> or contact me.

Nature Cell Biology is committed to improving transparency in authorship. As part of our efforts in this direction, we are now requesting that all authors identified as 'corresponding author' on published papers create and link their Open Researcher and Contributor Identifier (ORCID) with their account on the Manuscript Tracking System (MTS), prior to acceptance. ORCID helps the scientific community achieve unambiguous attribution of all scholarly contributions. You can create and link your ORCID from the home page of the MTS by clicking on 'Modify my Springer Nature account'. For more information please visit www.springernature.com/orcid.

This journal strongly supports public availability of data. Please place the data used in your paper into a public data repository, or alternatively, present the data as Supplementary Information. If data can only be shared on request, please explain why in your Data Availability Statement, and also in the correspondence with your editor. Please note that for some data types, deposition in a public repository is mandatory - more information on our data deposition policies and available repositories appears below.

Link Redacted

We hope that you will find our referees' comments and editorial guidance helpful. Please do not hesitate to contact me if there is anything you would like to discuss. Thank you again for considering our journal for your work.

Best wishes,

Melina

Melina Casadio, PhD
Senior Editor, Nature Cell Biology
ORCID ID: <https://orcid.org/0000-0003-2389-2243>

Reviewers' Comments:

Reviewer #1:

Remarks to the Author:

The manuscript by Hoffmann et al identified that Fructose-1,6-bisphosphate acts as a signalling molecule to couple cell glycolytic activity to cell adhesion and protrusion. Overall the study is well designed and executed. The insights from this study is likely to have significant impact on our understanding of the relationship between cell morphology to the metabolic state. There are some minor concerns and suggestions to further improve this study.

First of all, FBP Aldolase exists as three major isoforms including ALDOA, B and C. These three isoforms share large homological sequences thus difficult to distinguish experimentally. However, the distinct tissue and cell type-specific expressions are important for their enzymatic and non-enzymatic functions. In the genome-wide siRNA screen, how did the authors exclude ALDOB or C isoforms in this study?

Secondly, Fructose-1,6-bisphosphate (F-1,6-BP) and Fructose-2,6-bisphosphate (F-2,6-BP) are regional-isomers with distinct biological functions. A fluorometric assay kit was used to quantify the amount of F-1,6-BP. It is recommended to use mass spectrometry coupled with liquid chromatography to confirm the identity and quantity of these metabolites. It is such a critically important evidence underpinning the proposed mechanism. Chemically characterised standards of these two isomers must be used for confirmation.

Thirdly, what's the upstream signal responsible for the F-, 1,6-BP and aldolase-mediated cell adhesion and protrusion?

Lastly, ALDOA and ALDOB play a distinct role in different cancer types. For example, in liver tissue and hepatocellular carcinoma (HCC), ALDOB is the major isoform and is significantly down regulated during the progression of HCC. Thus how does accumulation of FBP due to the ALDOB down regulation may affect cancer cell adhesion with the mechanism identified in this study? Please comment in the discussion.

Reviewer #2:

Remarks to the Author:

The manuscript by Hoffman et al. describes a striking new link between flux in the glycolytic pathway, the activation of the small GTPase Rac and the ability of cells to generate focal adhesions and actin-based protrusions. Whilst others have shown that glycolytic enzymes can interact with F-actin, this study differs in that it describes a potential interaction between a specific metabolite, fructose-1,6 bisphosphate (FBP) and RCC2, a regulator of Rac activity. The study manipulates flux through the glycolytic pathway by knocking down key enzymes, beginning with aldolase which they identify in an siRNA screen for regulators of adhesion. Knockdown of aldolase results in an accumulation of FBP and an increase in FA number/cell spreading, whilst knockdown of PFK, the enzyme upstream of FBP, or inhibition of glycolytic flux with 2-deoxyglucose both result in a decrease in FAs and cell spreading. The authors then use another 'global' approach, limited proteolysis mass spectrometry, to reveal that Rac and the Rac binding protein RCC2 change conformation in the presence of FBP. They further demonstrate that RCC2 is a potential target of FBP binding, and that RCC2 can suppress the ability of Rac and aldolase knockdown to promote adhesion formation and cell protrusion. The study is highly innovative in ideas and approaches and is generally very well controlled revealing an exciting new connection between metabolism and Rac signalling to the cytoskeleton/adhesion formation. My main concern is that the influence FBP on Rac-RCC2 complex formation, and Rac activity, is not addressed and hence whilst the mechanism inferred is very novel, it seems less than completely proven at this stage.

Specific comments:

1) The authors very elegantly show that Rac and RCC2 conformations are altered upon increasing concentrations of FBP. They used combined knockdown of Rac/RCC2, LiP-MS on lysates and purified proteins along with modelling of potential interaction sites to suggest that binding of FBP to RCC2 prevents RCC2-Rac binding, relieving suppression of Rac activity which in turn influences adhesion/spreading. However they do not address the fundamental interaction at the heart of this mechanism: RCC2-Rac complex formation and the influence of FBP on it. With the tools they have in hand it should be relatively straightforward to show this, and also the direct impact on Rac activity with purified proteins.

2) In figure 8K/M the authors express eGFP-RCC2 in aldolase knockdown cells and claim 'Strikingly, ectopic RCC2 expression sufficed to suppress the FBP-mediated FA increase and elevated cell size'. However there is no indication that aldolase knockdown cells respond differently to the expression of eGFP or eGFP-RCC2. I feel that this is a very important comparison to analyse statistically.

3) The authors manipulate glycolytic flux to modulate FBP levels. However, there is no indication of the physiological relevance of the levels reached, and/or the levels required to suppress RCC2-Rac regulation. Whilst some of this could be local regulation, the authors do suggest that in certain contexts glycolytic flux could be altered (metastatic cancer cells, endothelial cells, immune cells etc). Do the levels of FBP found in these cells correspond to levels that might impact RCC2-mediated Rac regulation?

4) The authors control for off target effects of aldolase knockdown very well. However, for other knockdowns this doesn't

seem to have been a consideration. It is important that for some key experiments PFK and Rac are depleted using an alternative siRNA.

Minor comments:

Figure 8B modelling: use of colour could help the reader discern specific binding interfaces

Extended data 7: b and c switched in legend?

Reviewer #3:

Remarks to the Author:

Hoffman et al. utilized a genome-wide loss-of-function RNAi screen to identify aldolase A as an important regulator of focal adhesion assembly and cell morphogenesis. Aldolase A generates DHAP and G3P from fructose 1,6 bisphosphate (FBP). The authors argue that changes in glycolytic flux (specifically aldolase A activity) change the concentration of FBP, and that FBP subsequently binds to RCC2. Binding of FBP to RCC2 prevents RCC2 binding to Rac1, thereby promoting actin reorganization, increased focal adhesions, and protrusive activity.

Although aldolase A is known to be released from the actin cytoskeleton upon Rac1 mediated actin remodeling to increase glycolytic flux, this manuscript proposes that increased glycolytic flux can also act in a reciprocal way to influence cytoskeletal dynamics. This would be a novel function for FBP and an exciting way for metabolism to drive a cell adhesion phenotype. In this referee's opinion, it is an interesting mechanism that would be of broad interest to a general cell biological audience. That said, several major key experiments are still needed to substantiate the claim and to verify its physiological relevance.

1. One important question relates to the physiological relevance of FBP activation. In the cell line studied here by the authors, upon knock down of aldolase A, FBP increases by a staggering 10-15 fold. It is questionable whether FBP would ever increase anywhere close to that magnitude in a naturally occurring setting without knockdown. When comparing proliferating and non-proliferating cancer cells, for example, some studies report only a modest change in FBP levels of a few percent.

A. The authors should measure the actual concentrations of FBP (currently they only show relative changes after aldolase A knockdown).

B. The authors claim that FBP levels changes as glycolytic flux changes, however, they never show this. They should manipulate glycolytic flux and quantify changes in FBP in the same cell line in which this study was conducted. This could be accomplished by administering 2DG at different levels. Currently, evidence that FBP links metabolic flux to focal adhesions is missing.

C. How closely do the changes in FBP levels under physiological conditions (eg, during proliferation, hypoxia, nutrient availability, etc) mimic quantitative changes in FBP under which this study was conducted? Aldolase A is unlikely to be unexpressed in wildtype cells. Its activity is regulated by mechanisms that provide less dynamic range.

D. Do changes in FBP levels less than 15-fold (which I suspect will reflect most of physiology) still impact focal adhesions?

2. As a control, the authors knockdown PFK and GAPDH. These are relatively crude biochemical assays in that they are not especially specific. Cancer cells generally do not proliferate without glycolysis. I imagine knocking down PFK would cause the cells to become less viable, or even die, which would result in less focal adhesions. Less focal adhesions due to cell death or cells being unhealthy would mean the data cannot be used to support the authors' mechanism.

A. Did the authors take changes in cell viability and proliferation rate into account when quantifying the number of focal adhesions with each condition? Do cells show any changes in proliferation, invasion, attachment, etc?

B. Why doesn't GAPDH lead to similar phenotype as aldolase A knockout? GAPDH is immediately downstream of aldolase

A. It should impede glycolytic flux. Further, it should lead to the accumulation of FBP just as aldolase A knockdown does. Quantitation of FBP would be helpful here to interpret these findings.

3. Specificity of mechanism to fructose 1,6 bisphosphate.

A. It is well established that F2,6BP increases glycolytic flux through PFK2. Knocking down aldolase A could increase both F1,6BP and F2,6BP production. The authors only performed assays with F1,6BP. Is binding of FBP to RCC2 specific to F1,6BP, or could they have actually elucidated a mechanism for F2,6BP? Do limited proteolysis coupled to MS provide the same results when performed with F2,6BP as F1,6BP?

4. Can the authors introduce a mutation into RCC2 that prevents FBP binding but enables its WT activity? Given that FBP and Rac1 could bind in the same region, this may not be possible. If feasible, however, it would provide compelling evidence for the proposed mechanism. Can the authors incubate RCC2 with Rac1 and increasing concentrations of FBP, followed by a co-immunoprecipitation analysis?

5. While I do not think the authors need to do any in vivo experiments to prove their mechanism, I do agree that it would be useful to obtain data from more than a single cell line to assess broad applicability and reproducibility.

Minor:

1. Statistics: From statistical lines in bar graphs, it is sometimes unclear which groups are being compared. Some bars look significant, but do not include any statistical test or p-value. Some plots appear to only have n=2, which is underpowered. All ANOVAs were conducted using Dunnett's post-test. Although this test is appropriate, a Tukey's post hoc test might be more beneficial.
2. The legends for Supplemental Figure 8B and C appear to have been accidentally flipped.
3. Some prior work has indicated that FBP is membrane permeable (doi: 10.1007/s11010-012-1279-x, doi: 10.1023/b:mcbi.0000021356.89867.0d, doi: 10.1152/ajpheart.1994.267.6.h2325). Although surprising, if true, this might be a useful approach to the authors to test mechanism.
4. Figure captions use the notation N=3. The authors should clarify what this means (eg, how many cells/plates/etc.).
5. Although it was once thought that cancer cells switch their metabolism from oxphos to glycolysis as the authors state in the Discussion, increasing data suggests that oxphos increases to its maximum rate of activity and glycolysis is an overflow channel (doi: 10.1016/j.tcb.2023.03.013).

REFERENCES – are limited to a total of 70 for Articles, Resources, Technical Reports; and 40 for Letters. This includes references in the main text and Methods combined. References must be numbered sequentially as they appear in the main text, tables and figure legends and Methods and must follow the precise style of Nature Cell Biology references. References only cited in the Methods should be numbered consecutively following the last reference cited in the main text. References

only associated with Supplementary Information (e.g. in supplementary legends) do not count toward the total reference limit and do not need to be cited in numerical continuity with references in the main text. Only published papers can be cited, and each publication cited should be included in the numbered reference list, which should include the manuscript titles. Footnotes are not permitted.

Methods should be written concisely, but should contain all elements necessary to allow interpretation and replication of the results. As a guideline, Methods sections typically do not exceed 3,000 words. The Methods should be divided into subsections listing reagents and techniques. When citing previous methods, accurate references should be provided and any alterations should be noted. Information must be provided about: antibody dilutions, company names, catalogue numbers and clone numbers for monoclonal antibodies; sequences of RNAi and cDNA probes/primers or company names and catalogue numbers if reagents are commercial; cell line names, sources and information on cell line identity and authentication. Animal studies and experiments involving human subjects must be reported in detail, identifying the committees approving the protocols. For studies involving human subjects/samples, a statement must be included confirming that informed consent was obtained. Statistical analyses and information on the reproducibility of experimental results should be provided in a section titled "Statistics and Reproducibility".

All Nature Cell Biology manuscripts submitted on or after March 21 2016 must include a Data availability statement as a separate section after Methods but before references, under the heading "Data Availability". For Springer Nature policies on data availability see <http://www.nature.com/authors/policies/availability.html>; for more information on this particular policy see <http://www.nature.com/authors/policies/data/data-availability-statements-data-citations.pdf>. The Data availability statement should include:

- Accession codes for primary datasets (generated during the study under consideration and designated as "primary accessions") and secondary datasets (published datasets reanalysed during the study under consideration, designated as "referenced accessions"). For primary accessions data should be made public to coincide with publication of the manuscript. A list of data types for which submission to community-endorsed public repositories is mandated (including sequence, structure, microarray, deep sequencing data) can be found here <http://www.nature.com/authors/policies/availability.html#data>.
- Unique identifiers (accession codes, DOIs or other unique persistent identifier) and hyperlinks for datasets deposited in an approved repository, but for which data deposition is not mandated (see here for details <http://www.nature.com/sdata/data-policies/repositories>).
- At a minimum, please include a statement confirming that all relevant data are available from the authors, and/or are included with the manuscript (e.g. as source data or supplementary information), listing which data are included (e.g. by figure panels and data types) and mentioning any restrictions on availability.
- If a dataset has a Digital Object Identifier (DOI) as its unique identifier, we strongly encourage including this in the Reference list and citing the dataset in the Methods.

We recommend that you upload the step-by-step protocols used in this manuscript to the Protocol Exchange. More details can be found at www.nature.com/protocolexchange/about.

All imaging data should be accompanied by scale bars, which should be defined in the legend.

Cropped images of gels/blots are acceptable, but need to be accompanied by size markers, and to retain visible background signal within the linear range (i.e. should not be saturated). The boundaries of panels with low background have to be demarked with black lines. Splicing of panels should only be considered if unavoidable, and must be clearly marked on the figure, and noted in the legend with a statement on whether the samples were obtained and processed simultaneously. Quantitative comparisons between samples on different gels/blots are discouraged; if this is unavoidable, it should only be performed for samples derived from the same experiment with gels/blots were processed in parallel, which needs to be stated in the legend.

Figures should be provided at approximately the size that they are to be printed at (single column is 86 mm, double column is 170 mm) and should not exceed an A4 page (8.5 x 11"). Reduction to the scale that will be used on the page is not necessary, but multi-panel figures should be sized so that the whole figure can be reduced by the same amount at the smallest size at which essential details in each panel are visible. In the interest of our colour-blind readers we ask that you avoid using red and green for contrast in figures. Replacing red with magenta and green with turquoise are two possible

colour-safe alternatives. Lines with widths of less than 1 point should be avoided. Sans serif typefaces, such as Helvetica (preferred) or Arial should be used. All text that forms part of a figure should be rewritable and removable.

The total number of Supplementary Figures (not including the "unprocessed scans" Supplementary Figure) should not exceed the number of main display items (figures and/or tables (see our Guide to Authors and March 2012 editorial <http://www.nature.com/ncb/authors/submit/index.html#suppinfo>; <http://www.nature.com/ncb/journal/v14/n3/index.html#ed>). No restrictions apply to Supplementary Tables or Videos, but we advise authors to be selective in including supplemental data.

GUIDELINES FOR EXPERIMENTAL AND STATISTICAL REPORTING

REPORTING REQUIREMENTS – We are trying to improve the quality of methods and statistics reporting in our papers. To that end, we are now asking authors to complete a reporting summary that collects information on experimental design and reagents. The Reporting Summary can be found here <https://www.nature.com/documents/nr-reporting-summary.pdf>. If you would like to reference the guidance text as you complete the template, please access these flattened versions at <http://www.nature.com/authors/policies/availability.html>.

Version 1:

Decision Letter:

*Please delete the link to your author homepage if you wish to forward this email to co-authors.

Dear Professor Maritzen,

Your revised manuscript, "The signaling metabolite fructose-1,6-bisphosphate couples glycolytic activity to cell adhesion and protrusion", has been seen by 3 referees, two of the original referees and one new referee 4 to replace referee 3. The reviewers are experts in metabolism (referee 1); adhesion (referee 2); and metabolism (referee 4). As you will see from their comments (attached below) they find this work of potential interest, but have raised substantial concerns, which in our view would need to be addressed with considerable revisions before we can consider publication in Nature Cell Biology. Please note that these concerns must all be addressed in full and that these revisions will be the final time we approach the referees.

Nature Cell Biology editors discuss the referee reports in detail within the editorial team, including the chief editor, to identify key referee points that should be addressed with priority, and requests that are overruled as being beyond the scope of the current study. To guide the scope of the revisions, I have listed these points below. We are committed to providing a fair and constructive peer-review process, so please feel free to contact me if you would like to discuss any of the referee comments further.

In particular, it would be essential to:

- address Reviewer #4 concerns 1, 2, 4, and 5 in full, with new experiments as necessary.
- All other referee concerns pertaining to strengthening existing data by providing controls, methodological details, clarifications and textual changes should also be addressed.
- Finally please pay close attention to our guidelines on statistical and methodological reporting (listed below) as failure to do so may delay the reconsideration of the revised manuscript. In particular please provide:
 - a Supplementary Figure including unprocessed images of all gels/blots in the form of a multi-page pdf file. Please ensure that blots/gels are labeled and the sections presented in the figures are clearly indicated.

We would be happy to consider a revised manuscript that would satisfactorily address these points, unless a similar paper is published elsewhere, or is accepted for publication in Nature Cell Biology in the meantime.

In contrast, although we agree with referee 4 that point 3 regarding ALDOA binding mutants would provide valuable insights, we consider this point to be beyond the scope of the present study. Thus, addressing it experimentally will not be necessary for reconsideration of the manuscript at this journal.

- ensure that it conforms to our format instructions and publication policies (see below and <https://www.nature.com/nature/for-authors>).

- provide a point-by-point rebuttal to the full referee reports verbatim, as provided at the end of this letter.

- provide the completed Reporting Summary (found here <https://www.nature.com/documents/nr-reporting-summary.pdf>). This is essential for reconsideration of the manuscript will be available to editors and referees in the event of peer review. For more information see <http://www.nature.com/authors/policies/availability.html> or contact me.

Nature Cell Biology is committed to improving transparency in authorship. As part of our efforts in this direction, we are now requesting that all authors identified as 'corresponding author' on published papers create and link their Open Researcher and Contributor Identifier (ORCID) with their account on the Manuscript Tracking System (MTS), prior to acceptance. ORCID helps the scientific community achieve unambiguous attribution of all scholarly contributions. You can create and link your ORCID from the home page of the MTS by clicking on 'Modify my Springer Nature account'. For more information please visit www.springernature.com/orcid.

This journal strongly supports public availability of data. Please place the data used in your paper into a public data repository, or alternatively, present the data as Supplementary Information. If data can only be shared on request, please explain why in your Data Availability Statement, and also in the correspondence with your editor. Please note that for some data types, deposition in a public repository is mandatory - more information on our data deposition policies and available repositories appears below.

Link Redacted

We would like to receive a revised submission within six months.

We hope that you will find our referees' comments, and editorial guidance helpful. Please do not hesitate to contact me if there is anything you would like to discuss.

Best wishes,

Angela Parrish

Angela R Parrish, PhD
Locum Senior Editor

Reviewers' Comments:

Reviewer #1 (Remarks to the Author):

The authors successfully addressed all my concerns

Reviewer #2 (Remarks to the Author):

The authors have performed new experiments and included fantastic new data and analyses to answer my comments. I'm happy to recommend for publication.

Reviewer #4 (Remarks to the Author):

This article shows very interesting data on the connection between the glycolytic intermediate fructose 1,6-bisphosphate (FBP) and cell adhesion mediated by the interaction between Rac1 and its negative regulator RCC2. Based on the data from a whole genome siRNA screen, the authors show that silencing of ALDOA leads to increased cell area and a higher number of focal adhesions per cell. The authors then show that the catalytic activity of ALDOA rather than substrate binding is responsible for this effect. Furthermore, the authors show that silencing of PFK (all three isoforms) reduces cell area and focal adhesion number while depletion of GAPDH has no effect. Metabolomics analysis revealed that ALDOA silencing resulted in increased levels of its substrate, fructose 1,6-bisphosphate (FBP) and inhibition of FBP accumulation by inhibition of hexokinase prevented increased cell area. Using time-lapse analysis, the authors next show that PFK silencing reduces focal adhesion assembly while ALDOA depletion increase focal adhesion assembly rates, consistent with a role of FBP in this process. The authors next employ LiP-MS to show that FBP induces conformational changes in Rac1 and that Rac1 is required for FBP-mediated F-actin reorganisation. Finally, the authors also implicate RCC2, a negative regulator of Rac1, in this process, resulting in a model in which FBP leads to the disassembly of RCC2 from Rac1 to promote cell spreading.

Overall, the manuscript shows highly interesting data. A particular strength lies in the identification of Rac1 and RCC2 as FBP binding proteins by LiP-MS. This finding offers interesting insight into metabolite/protein interactions and provides an important link between metabolic activity and cell regulation. However, there are some issues that need to be addressed before the manuscript can be considered for publication.

Specific comments:

- 1) The authors discuss that FBP levels tightly correlate with glycolytic flux in many systems, making FBP a signalling metabolite that informs about the metabolic state of a cell. However, experiments shown in the manuscript mostly use siRNA-mediated depletion of FBP-producing and FBP-consuming enzymes, i.e. PFK and ALDOA, rather than studying the effect of modulating glycolytic flux of Rac1 activation. It is not clear whether changes in FBP levels observed during modulation of glycolytic flux, for example after acute exposure to glucose, are sufficient to trigger Rac1 activation. Under these conditions, changes in FBP levels are likely to be smaller and more transient compared to those observed after ALDOA depletion. The authors include results to show that FBP levels are comparable to those found in migrating or spreading cells. However, these situations reflect the reverse regulation, i.e. cytoskeletal changes modulating glycolytic flux rather than the opposite. Given the broad and interesting implications of the findings, a more detailed analysis of the link between glycolytic flux and FBP-mediated Rac1 activation would be warranted.
- 2) It is quite surprising that siRNA mediated depletion of glycolytic enzymes (PFK, ALDOA, GAPDH) using a double transfection protocol does not have major effects on cell number after 96 hours of incubation. The authors use CYTOX to confirm cell viability but they should also perform experiments to demonstrate any effects of proliferation. Changing cell cycle distribution in the population is likely to also have effects on cell size and focal adhesions
- 3) As ALDOA is also a target for cytoskeletal regulation via Rac1, it is surprising that the authors did not include experiments using the actin binding mutant of ALDOA (R43A) in their experiments shown in Figure 2a-c.
- 4) The authors mention that accumulation of FBP in response to ALDOA downregulation would promote cancer cell spreading and potentially drive liver carcinogenesis. However, a recent study demonstrated that ALDOA is essential for liver tumour formation in a mouse model by preventing imbalanced glycolysis, leading to very high levels of FBP accumulation (DOI: 10.1038/s42255-024-01201-w). The authors should include this publication in their discussion.
- 5) There is no method provided for the LC-MS/MS analysis of FBP levels. It is essential to provide a full protocol, including the use of standards for FBP quantification.

Reviewer #4 (Remarks on Protocol(s)):

Inclusion of a detailed protocol for LC-MS/MS analysis of FBP levels is mandatory.

Methods should be written concisely, but should contain all elements necessary to allow interpretation and replication of the results. As a guideline, Methods sections typically do not exceed 3,000 words. The Methods should be divided into subsections listing reagents and techniques. When citing previous methods, accurate references should be provided and any alterations should be noted. Information must be provided about: antibody dilutions, company names, catalogue numbers and clone numbers for monoclonal antibodies; sequences of RNAi and cDNA probes/primers or company names and catalogue numbers if reagents are commercial; cell line names, sources and information on cell line identity and authentication. Animal studies and experiments involving human subjects must be reported in detail, identifying the committees approving the protocols. For studies involving human subjects/samples, a statement must be included confirming that informed consent was obtained. Statistical analyses and information on the reproducibility of experimental results should be provided in a section titled "Statistics and Reproducibility".

All Nature Cell Biology manuscripts submitted on or after March 21 2016 must include a Data availability statement as a

separate section after Methods but before references, under the heading "Data Availability". For Springer Nature policies on data availability see <http://www.nature.com/authors/policies/availability.html>; for more information on this particular policy see <http://www.nature.com/authors/policies/data/data-availability-statements-data-citations.pdf>. The Data availability statement should include:

- Accession codes for primary datasets (generated during the study under consideration and designated as "primary accessions") and secondary datasets (published datasets reanalysed during the study under consideration, designated as "referenced accessions"). For primary accessions data should be made public to coincide with publication of the manuscript. A list of data types for which submission to community-endorsed public repositories is mandated (including sequence, structure, microarray, deep sequencing data) can be found here <http://www.nature.com/authors/policies/availability.html#data>.
- Unique identifiers (accession codes, DOIs or other unique persistent identifier) and hyperlinks for datasets deposited in an approved repository, but for which data deposition is not mandated (see here for details <http://www.nature.com/sdata/data-policies/repositories>).
- At a minimum, please include a statement confirming that all relevant data are available from the authors, and/or are included with the manuscript (e.g. as source data or supplementary information), listing which data are included (e.g. by figure panels and data types) and mentioning any restrictions on availability.
- If a dataset has a Digital Object Identifier (DOI) as its unique identifier, we strongly encourage including this in the Reference list and citing the dataset in the Methods.

We recommend that you upload the step-by-step protocols used in this manuscript to protocols.io. More details can be found at <https://www.protocols.io/help/publish-articles>.

All imaging data should be accompanied by scale bars, which should be defined in the legend.

Cropped images of gels/blots are acceptable, but need to be accompanied by size markers, and to retain visible background signal within the linear range (i.e. should not be saturated). The boundaries of panels with low background have to be demarked with black lines. Splicing of panels should only be considered if unavoidable, and must be clearly marked on the figure, and noted in the legend with a statement on whether the samples were obtained and processed simultaneously. Quantitative comparisons between samples on different gels/blots are discouraged; if this is unavoidable, it should only be performed for samples derived from the same experiment with gels/blots were processed in parallel, which needs to be stated in the legend.

EXTENDED DATA FIGURES - When re-submitting your manuscript, please ensure that any supplementary figures and tables that are crucial to the manuscript's conclusions are converted into Extended Data figures and tables to increase visibility of these data. Extended Data figures and tables are online-only (present in the online PDF and full-text HTML versions of the paper), peer-reviewed display items that provide essential background to the article but are not included in the main article due to space constraints. A maximum of ten Extended Data display items (figures and tables) is permitted.

The total number of Supplementary Figures (not including the “unprocessed scans” Supplementary Figure) should not exceed the number of main display items (figures and/or tables (see our Guide to Authors and March 2012 editorial <http://www.nature.com/ncb/authors/submit/index.html#suppinfo>; <http://www.nature.com/ncb/journal/v14/n3/index.html#ed>). No restrictions apply to Supplementary Tables or Videos, but we advise authors to be selective in including supplemental data.

GUIDELINES FOR EXPERIMENTAL AND STATISTICAL REPORTING

REPORTING REQUIREMENTS – We are trying to improve the quality of methods and statistics reporting in our papers. To that end, we are now asking authors to complete a reporting summary that collects information on experimental design and reagents. The Reporting Summary can be found here <https://www.nature.com/documents/nr-reporting-summary.pdf>. If you would like to reference the guidance text as you complete the template, please access these flattened versions at <http://www.nature.com/authors/policies/availability.html>.

STATISTICS – Wherever statistics have been derived the legend needs to provide the n number (i.e. the sample size used to derive statistics) as a precise value (not a range), and define what this value represents. Error bars need to be defined in the legends (e.g. SD, SEM) together with a measure of centre (e.g. mean, median). Box plots need to be defined in terms of minima, maxima, centre, and percentiles. Ranges are more appropriate than standard errors for small data sets. Wherever statistical significance has been derived, precise p values need to be provided and the statistical test used needs to be stated in the legend. Statistics such as error bars must not be derived from $n < 3$. For sample sizes of $n < 5$ please plot the

individual data points rather than providing bar graphs. Deriving statistics from technical replicate samples, rather than biological replicates is strongly discouraged. Wherever statistical significance has been derived, precise p values need to be provided and the statistical test stated in the legend.

Version 2:

Decision Letter:

Our ref: NCB-A52036B

5th January 2026

Dear Dr. Maritzen,

Thank you for submitting your revised manuscript "The signaling metabolite fructose-1,6-bisphosphate couples glycolytic activity to cell adhesion and protrusion" (NCB-A52036B). It has now been seen by the original referee and their comments are below. The reviewer finds that the paper has improved in revision, and therefore we'll be happy in principle to publish it in Nature Cell Biology, pending minor revisions to comply with our editorial and formatting guidelines.

We are now performing detailed checks on your paper and will send you a checklist detailing our editorial and formatting requirements in about 1-2 weeks. Please do not upload the final materials and make any revisions until you receive this additional information from us.

Thank you again for your interest in Nature Cell Biology. Please do not hesitate to contact me if you have any questions.

Sincerely,

Melina Casadio, PhD
Senior Editor, Nature Cell Biology
ORCID ID: <https://orcid.org/0000-0003-2389-2243>

Reviewer #4 (Remarks to the Author):

The authors have made excellent arguments to address the points raised. They can be congratulated on their insightful study.

Reviewer #4 (Remarks on Protocol(s)):

The protocols provide sufficient experimental detail.

Version 3:

Decision Letter:

Dear Dr Maritzen,

I am pleased to inform you that your manuscript, "Fructose-1,6-bisphosphate couples glycolytic activity to cell adhesion", has now been accepted for publication in Nature Cell Biology.

Thank you for sending us the final manuscript files to be processed for print and online production, and for returning the manuscript checklists and other forms. Your manuscript will now be passed to our production team who will be in contact

with you if there are any questions with the production quality of supplied figures and text.

Please note that *Nature Cell Biology* is a Transformative Journal (TJ). Authors may publish their research with us through the traditional subscription access route or make their paper immediately open access through payment of an article-processing charge (APC). Authors will not be required to make a final decision about access to their article until it has been accepted. [Find out more about Transformative Journals](https://www.springernature.com/gp/open-research/transformative-journals)

Authors may need to take specific actions to achieve compliance with funder and institutional open access mandates. If your research is supported by a funder that requires immediate open access (e.g. according to [Plan S principles](https://www.springernature.com/gp/open-science/plan-s-compliance) or the [NIH public access policy](https://www.springernature.com/gp/open-science/us-federal-agency-compliance)) then you should select the gold OA route, and we will direct you to the compliant route where possible. Because authors warrant under our subscription licensing terms that they haven't committed to licensing any version of their article under a licence inconsistent with the terms of our agreement – including the applicable embargo period – publication under the subscription model isn't suitable for authors whose funders require no embargo.

If you have not already done so, we strongly recommend that you upload the step-by-step protocols used in this manuscript to protocols.io (<https://protocols.io>), an open online resource that allows researchers to share their detailed experimental know-how. All uploaded protocols are made freely available and are assigned DOIs for ease of citation. Protocols and Nature Portfolio journal papers in which they are used can be linked to one another, and this link is clearly and prominently visible in the online versions of both. Authors who performed the specific experiments can act as primary authors for the Protocol as they will be best placed to share the methodology details, but the Corresponding Author of the present research paper should be included as one of the authors. By uploading your Protocols onto protocols.io, you are enabling researchers to more readily reproduce or adapt the methodology you use, as well as increasing the visibility of your protocols and papers. You can also establish a dedicated workspace to collect your lab Protocols. Further information can be found at <https://www.protocols.io/help/publish-articles>.

Nature Cell Biology encourages authors presenting evidence for cell, biological, molecular, and genetic interactions to consider communicating these findings using Biofactoid (<https://biofactoid.org/>). This tool helps users share a searchable representation of interactions (e.g. binding, gene expression, post-translational modification) between genes, gene products, or chemicals. Information added to Biofactoid, with author attribution, is shared on social media and public databases, such as Pathway Commons, where it can be discovered and analyzed in the context of a large and growing corpus of knowledge.

With kind regards,

Melina Casadio, PhD
Senior Editor, Nature Cell Biology
ORCID ID: <https://orcid.org/0000-0003-2389-2243>

** Visit the Springer Nature Editorial and Publishing website at http://editorial-jobs.springernature.com?utm_source=ejp_NCB_email&utm_medium=ejp_NCB_email&utm_campaign=ejp_NCB for more information about our career opportunities. If you have any questions please click [here](mailto:editorial.publishing.jobs@springernature.com).

Point-by-point Responses to Reviewer Comments

Reviewer #1:

The manuscript by Hoffmann et al identified that Fructose-1,6-bisphosphate acts as a signalling molecule to couple cell glycolytic activity to cell adhesion and protrusion. Overall the study is well designed and executed. The insights from this study is likely to have significant impact on our understanding of the relationship between cell morphology to the metabolic state.

We thank the Reviewer for his/her positive appraisal of our study.

There are some minor concerns and suggestions to further improve this study. First of all, FBP Aldolase exists as three major isoforms including ALDOA, B and C. These three isoforms share large homological sequences thus difficult to distinguish experimentally. However, the distinct tissue and cell type-specific expressions are important for their enzymatic and non-enzymatic functions. In the genome-wide siRNA screen, how did the authors exclude ALDOB or C isoforms in this study?

The genome-wide screen contained siRNA pools against all three aldolase isoforms. However, only the siRNA pool targeting aldolase A caused a focal adhesion phenotype. This is due to the fact that the three aldolase isoforms show indeed very different tissue- and cell-type specific expression patterns with aldolase A being by far the most strongly expressed in U-2 OS cells. The Human Protein Atlas (www.proteinatlas.org), an open access resource on human proteins and their expression, provides detailed information on the expression pattern of the aldolase enzymes specifically in U-2 OS cells. Its cell line dataset indicates that the RNA transcript level of aldolase A in U-2 OS cells is 744.3 nTPM (normalized transcripts per million), while the transcript level of aldolase B is 0.4 nTPM and that of aldolase C 5.4 nTPM. Thus, aldolase A is expressed 138-fold higher than aldolase C and 1861-fold higher than aldolase B making the contribution of aldolase B and C to the total aldolase pool in U-2 OS cells negligible. The same holds true for the hTERT-RPE1 cells which are also included in the cell line dataset and in which we reproduced the focal adhesion phenotype with siRNAs targeting aldolase A. Here, the transcript level for aldolase A is at 2129.3 nTPM, while aldolase B is at 0.1 nTPM and aldolase C at 1.7 nTPM. We now include this information in the Results section on page 4.

(Source:

<https://www.proteinatlas.org/ENSG00000149925-ALDOA/summary/rna>

<https://www.proteinatlas.org/ENSG00000136872-ALDOB/summary/rna>

<https://www.proteinatlas.org/ENSG00000109107-ALDOC/summary/rna>)

Secondly, Fructose-1,6-bisphosphate (F-1,6-BP) and Fructose-2,6-bisphosphate (F-2,6-BP) are regional-isomers with distinct biological functions. A fluorometric assay kit was used to quantify the amount of F-1,6-BP. It is recommended to use mass spectrometry coupled with liquid chromatography to confirm the identity and quantity of these metabolites. It is such a critically important evidence underpinning the proposed mechanism. Chemically characterised standards of these two isomers must be used for confirmation.

We agree that mass spectrometry (MS) coupled with liquid chromatography is the method of choice to quantify F1,6BP in a highly reliable manner. However, it was originally very difficult for us to find a collaboration partner who had the necessary expertise to resolve F1,6BP from other phosphorylated sugars such as glucose-1,6-bisphosphate. However, a colleague introduced us to Dr. John Lunn from the MPI of Molecular Plant Physiology, who has a well established LC-MS/MS platform, using a triple quadrupole mass spectrometer, for measuring phosphorylated intermediates. The anion-exchange HPLC phase provides baseline separation

of F1,6BP from F2,6BP and Glucose1,6BP. The mass spectrometry involves selection of the parent ion in the first quadrupole, fragmentation in the second quadrupole, and then quantification of three product ions in the third quadrupole. The fragmentation patterns are characteristic for each compound, so this tandem mass spectrometry provides an additional level of specificity for the measurements. Furthermore, each individual sample is spiked with a known amount of a stable-isotope labelled version of F1,6BP – [¹³C₆]F1,6BP – as an internal standard to correct for an ion suppression or other matrix effects. The LC-MS/MS measurements of F1,6BP are thus highly specific and confirmed the strong increase in F1,6BP upon aldolase A depletion that we had previously seen with our fluorometric measurements (see **new Figure 5f**). Even more excitingly, when we compared confluent cells to spreading cells which we had postulated to rely on elevated F1,6BP levels for their protrusive activity, we indeed found a similarly striking increase in F1,6BP, and also migrating cells displayed significantly elevated F1,6BP levels (see **new Figure 5i**).

We can confidently exclude the possibility that F2,6BP confounded our F1,6BP measurements, as there is baseline separation of F1,6BP and F2,6BP standards on the HPLC column used for the LC-MS/MS analysis. Furthermore, F2,6BP and F1,6BP can be differentiated in the tandem mass spectrometry due to their characteristic fragmentation patterns. No peaks with the retention time and mass spectral characteristics of F2,6BP were detected in the samples. This was not unexpected as the levels of this metabolite are generally very low in cells (several orders of magnitude less than F1,6BP – see below). In addition, F2,6BP is very acid labile (Hers & Van Schaftingen, 1982; <https://doi.org/10.1042/bj2060001>) and therefore special precautions (for example, extraction with alkaline reagents) are usually needed to prevent its hydrolysis to fructose 6-phosphate during sample preparation. It is known that the chloroform-methanol extraction method used to prepare samples for F1,6BP measurements does not always preserve F2,6BP, with recoveries being strongly dependent on the type of tissue or cells being extracted. F2,6BP is most commonly measured by an enzyme activation assay using the pyrophosphate-dependent phosphofructokinase from potato tubers (Stitt (1990) Fructose 2,6-bisphosphate. In *Methods in Plant Biochemistry* (Lea, P.J., ed.). London: Academic Press, pp. 87–92). John Lunn's group has extensive experience with this assay, but it was not feasible for them to measure F2,6BP in our samples using the enzymatic assay because the required F2,6BP standards are no longer commercially available.

F2,6BP is produced by phosphofructokinase 2 (PFK2) and serves as a regulator of PFK1 and fructose 1,6-bisphosphatase (FBPase) rather than being a glycolytic intermediate, and levels of F2,6BP are typically about one to two orders of magnitude lower than those of F1,6BP so that their contribution to the total FBP pool is very low. For example, a study analyzing HeLa and rat hepatoma cells reported 90- to 4200-fold lower levels of F2,6BP (measured by the mentioned elaborate enzyme assay based on purified components) compared to F1,6BP (Marín-Hernández, 2011; Table 3 and Table 4; <https://doi.org/10.1016/j.bbabi.2010.11.006>). Similarly, studies of mouse tissues such as rat muscles demonstrated 375-fold up to 1000-fold lower levels of F2,6BP compared to F1,6BP depending on the used stimulation protocols (Minatogawa, 1984; Table 1 and Table 2; <https://doi.org/10.1042/bj2230073>). We now include this information in the Results section on page 8. Thus, although we were unable to measure F2,6BP in the samples, we can exclude the possibility that F2,6BP confounded the LC-MS/MS-based measurements of F1,6BP, as this method fully resolves the two compounds. Based on published data, the levels of the F2,6BP in the cells would most likely be too low to make a significant contribution to the total FBP pool anyway. Finally, the fact that the PFK1 knockdown completely rescues the aldolase knockdown also argues for a F1,6BP-dependent mechanism.

Thirdly, what's the upstream signal responsible for the F-,1,6-BP and aldolase-mediated cell adhesion and protrusion?

This is a very interesting question for which a study by Kondo et al. (<https://doi.org/10.1016/j.celrep.2021.108750>) might provide a partial answer. In line with our observation of elevated F1,6BP levels in migrating and spreading cells as compared to confluent cell cultures, Kondo et al. observed in a scratch wound assay that the cells close to the scratch display a higher rate of glucose uptake and consumption. The higher glucose uptake was traced back to an upregulation of the glucose transporters GLUT1 and GLUT4. The authors found that glucose levels were in fact linked to cell density. In lower confluency states, which are conducive to cell spreading, GLUT1 and GLUT4 were significantly elevated compared to high confluency cell cultures. Thus, the cellular sensing of available space for protrusion apparently triggers an increased glucose uptake. This translates into an increased glucose consumption resulting in the elevated F1,6BP levels which we detected in migratory and spreading cells and which then promote their protrusive activity (see **new Figure 5i**). Consistent with our hypothesis of glycolytic flux as requirement for cellular protrusion and migration, Kondo et al observed that replacing glucose with 2-deoxyglucose significantly impaired wound closure. We now include this additional information in the Discussion on page 9-10.

Lastly, ALDOA and ALDOB play a distinct role in different cancer types. For example, in liver tissue and hepatocellular carcinoma (HCC), ALDOB is the major isoform and is significantly down regulated during the progression of HCC. Thus how does accumulation of FBP due to the ALDOB down regulation may affect cancer cell adhesion with the mechanism identified in this study? Please comment in the discussion.

Indeed, there are several studies documenting that aldolase B downregulation in hepatocellular carcinoma is linked to a poor prognosis for affected patients (Li et al, 2020; and references therein; <https://doi.org/10.1038/s43018-020-0086-7>). There are already diverse reasons discussed for the tumor promoting effect of aldolase B downregulation such as disinhibition of the pentose phosphate pathway supplying building blocks for tumor growth (Li et al, 2020), activation of insulin receptor signaling (Liu et al, 2021; <https://doi.org/10.1002/hep.32064>) and activation of Akt (He et al, 2020; <https://doi.org/10.1371/journal.pbio.3000803>). Based on our study we would like to complement these findings with the hypothesis that the following events will likely contribute to tumorigenesis: (a) Decreased levels of aldolase B will cause an increase in F1,6BP levels in hepatic tumor cells as shown for aldolase B-deficient mouse liver tumor cells (Li et al, 2020). (b) Increased F1,6BP levels with their effect on Rac1 activity might promote protrusion and migration of hepatic tumor cells. In line with this, Li et al show a 2-fold higher migration of aldolase B silenced liver cancer cells in transwell assays. (c) The higher migration rate might promote the frequently observed metastatic spread of hepatocellular carcinomas (Arora et al, 2021; <https://doi.org/10.1007/s00261-021-03151-3>). We now comment on this in the Discussion on page 10.

Reviewer #2:

The manuscript by Hoffman et al. describes a striking new link between flux in the glycolytic pathway, the activation of the small GTPase Rac and the ability of cells to generate focal adhesions and actin-based protrusions. Whilst others have shown that glycolytic enzymes can interact with F-actin, this study differs in that it describes a potential interaction between a specific metabolite, fructose-1,6 bisphosphate (FBP) and RCC2, a regulator of Rac activity. The study manipulates flux through the glycolytic pathway by knocking down key enzymes, beginning with aldolase which they identify in an siRNA screen for regulators of adhesion. Knockdown of aldolase results in an accumulation of FBP and an increase in FA number/cell spreading, whilst knockdown of PFK, the enzyme upstream of FBP, or inhibition of glycolytic flux with 2-deoxyglucose both result in a decrease in FAs and cell spreading. The authors then use another ‘global’ approach, limited proteolysis mass spectrometry, to reveal that Rac and

the Rac binding protein RCC2 change conformation in the presence of FBP. They further demonstrate that RCC2 is a potential target of FBP binding, and that RCC2 can suppress the ability of Rac and aldolase knockdown to promote adhesion formation and cell protrusion. The study is highly innovative in ideas and approaches and is generally very well controlled revealing an exciting new connection between metabolism and Rac signalling to the cytoskeleton/adhesion formation.

We thank the Reviewer for her/his very positive evaluation of our study.

My main concern is that the influence FBP on Rac-RCC2 complex formation, and Rac activity, is not addressed and hence whilst the mechanism inferred is very novel, it seems less than completely proven at this stage.

We agree with the Reviewer that this was indeed a shortcoming of our submitted manuscript. However, we now provide data for the ability of F1,6BP to influence Rac1-RCC2 complex formation as we discuss below.

Specific comments:

1) The authors very elegantly show that Rac and RCC2 conformations are altered upon increasing concentrations of FBP. They used combined knockdown of Rac/RCC2, LiP-MS on lysates and purified proteins along with modelling of potential interaction sites to suggest that binding of FBP to RCC2 prevents RCC2-Rac binding, relieving suppression of Rac activity which in turn influences adhesion/spreading. However they do not address the fundamental interaction at the heart of this mechanism: RCC2-Rac complex formation and the influence of FBP on it. With the tools they have in hand it should be relatively straightforward to show this, and also the direct impact on Rac activity with purified proteins.

Following the Reviewer's suggestion we have established a biochemical interaction assay to probe the F1,6BP influence on Rac1 - RCC2 complex formation. For this, we have expressed GFP-RCC2 in HEK293 cells and bound it to magnetic GFP-Trap beads in the presence or absence of F1,6BP. In parallel, we expressed GST-Rac1 in bacteria and purified it via glutathione-coupled beads. Eluted GST-Rac1 was added to the GFP-RCC2 on the GFP-trap beads. Following an incubation at 4°C and extensive washing, GFP-RCC2 and potentially interacting Rac1 were eluted from the beads, and samples were probed by western blotting. We were able to confirm the previously published interaction between RCC2 and Rac1 (Williamson et al, 2014; <https://doi.org/10.1242/jcs.154864>). In line with our hypothesis, our biochemical results demonstrate that the addition of F1,6BP significantly lowers the amount of Rac1 bound to RCC2 as depicted in the **new Figure 8m, n**. Since binding of RCC2 was previously shown to inhibit Rac1 (Williamson et al, 2014), which was confirmed by our own functional data (Figure 8), the decreased Rac1 - RCC2 complex formation will result in elevated Rac1 activity.

2) In figure 8K/M the authors express eGFP-RCC2 in aldolase knockdown cells and claim 'Strikingly, ectopic RCC2 expression sufficed to suppress the FBP-mediated FA increase and elevated cell size'. However there is no indication that aldolase knockdown cells respond differently to the expression of eGFP or eGFP-RCC2. I feel that this is a very important comparison to analyse statistically.

Our previous statistical analysis showed that eGFP expression did not abrogate the statistically highly significant increase of focal adhesion numbers and cell area in aldolase A knockdown cells, while the overexpression of GFP-RCC2 decreased this increase to a non-significant small

alteration, i.e. largely rescued the phenotype. However, we agree that this might not be the most straightforward way for comparing the effect of eGFP and GFP-RCC2 on the aldolase A knockdown cells. Therefore, we have now conducted a Tukey post-test to statistically evaluate all possible comparisons which clearly indicates a significant difference between the expression of eGFP as a control and of GFP-RCC2 in the aldolase A knockdown cells. We have added a corresponding indication of significance to the graph in the **new Figure 8k, l**.

3) The authors manipulate glycolytic flux to modulate FBP levels. However, there is no indication of the physiological relevance of the levels reached, and/or the levels required to suppress RCC2-Rac regulation. Whilst some of this could be local regulation, the authors do suggest that in certain contexts glycolytic flux could be altered (metastatic cancer cells, endothelial cells, immune cells etc). Do the levels of FBP found in these cells correspond to levels that might impact RCC2-mediated Rac regulation?

This is a very relevant question that we have tried to approach from different angles. So far, we had shown based on a fluorometric assay that the knockdown of aldolase A, which is indeed a rather artificial situation, leads to a ~6.5-fold increase in F1,6BP levels. First of all, we tested whether such a large increase is indeed necessary to observe effects on cell size. For this, we performed a single round of knockdown instead of our usual two rounds of siRNA transfection. This still led to a decrease in aldolase A, but (now based on LC-MS/MS measurements) the increase in F1,6BP was less than half that detected upon two rounds of knockdown (2.6 nmol/mg protein instead of 5.7 nmol/mg protein). Nevertheless, we still observed a significant increase in cell area, albeit smaller.

The next question was whether similar F1,6BP concentrations occur under physiological conditions. To stay within our experimental system, we decided to measure F1,6BP levels in U-2 OS cells under those cellular conditions where we had predicted that greater F1,6BP levels should be present to drive cellular protrusion, i.e. during cell spreading upon replating and during cell migration. As a control condition, we measured F1,6BP in stationary cells within confluent cell layers where protrusive activity is necessarily limited. Confluent cells contained 0.6 nmol FBP/mg protein. Excitingly, migrating cells, which need localized protrusions for their motility, had about 3-fold higher F1,6BP levels which is comparable to the level after a single round of aldolase A knockdown. Most strikingly, we detected 5.4 nmol F1,6BP/mg protein in spreading cells, which need massive Rac1-driven protrusions in all directions to enlarge their adhesive cell area. This is very similar to what we observe upon two rounds of aldolase A knockdown. We include these important new data as **new Figure 5e-i**. Thus, F1,6BP levels change indeed quite dramatically across different cell states. Therefore, the cellular changes in F1,6BP should be well within the range needed to allow RCC2 binding and thus influence RCC2 - Rac1 complex formation as demonstrated by our functional characterization of the aldolase A knockdown cells.

To also gain insights into changes in F1,6BP levels upon oncogenic cell transformation we consulted the literature. A study by Mazurek et al. showed that normal rat kidney cells very similar to our confluent cells contain less than 1 nmol F1,6BP/mg protein (i.e. 0.4 nmol/mg protein; Mazurek et al, 2001; DOI: 10.1038/sj.onc.1204792). However, the expression of Ras, one of the most common oncoproteins and known for its ability to rewire glycolysis (Chesney et al, 2013; DOI: 10.2174/1389201011314030002), elevated the F1,6BP concentration to 13.4 nmol/mg protein (Mazurek et al. 2001). The additional expression of the oncoprotein E7 of the human papillomavirus type 16 increased the F1,6BP concentration even further to 28.3 nmol/mg protein (Mazurek et al. 2001). Therefore, oncogenic transformations can also elevate cellular F1,6BP levels in the range of an aldolase A knockdown so that it is well imaginable that the glycolytic changes in cancer cells promote cell protrusion via our postulated F1,6BP-mediated mechanism.

4) The authors control for off target effects of aldolase knockdown very well. However, for other knockdowns this doesn't seem to have been a consideration. It is important that for some key experiments PFK and Rac are depleted using an alternative siRNA.

We agree that it is best to always use alternative siRNAs to control for off-target effects. Therefore, we have now tested additional siRNAs for PFK and Rac1 depletion in key experiments. For PFK, we have performed a knockdown with a PFKP-specific siRNA in U-2 OS cells. This confirmed the significant decrease in focal adhesions per cell and in cell area which we had previously observed in this cell line with a mix of siRNAs targeting all three isoforms PFKL, PFKM and PFKP. In addition, this showed that PFKP expression is relevant for PFK activity in U-2 OS cells. This is now depicted in **Extended Data Figure 2a-d**.

For Rac1, we have tested two new siRNAs in hTERT RPE-1 cells to follow also the advice of Reviewer 3 to recapitulate key experiments in other cell lines. Therefore, we have silenced Rac1 with two independent siRNAs alone and in combination with the knockdown of aldolase A in RPE-1 cells to confirm that Rac1 is essential for the ability of aldolase A loss to cause an increase in FA numbers and cell size. This is now depicted in **Extended Data Figure 7b-e**.

Minor comments:

Figure 8B modelling: use of colour could help the reader discern specific binding interfaces

We make use of colour in Figure 8b to make it easier for the readers to discern the two responding peptide sites.

Extended data 7: b and c switched in legend?

We thank the Reviewer for his/her careful reading of our manuscript. The legends are indeed switched. We have now corrected this error.

Reviewer #3:

Hoffman et al. utilized a genome-wide loss-of-function RNAi screen to identify aldolase A as an important regulator of focal adhesion assembly and cell morphogenesis. Aldolase A generates DHAP and G3P from fructose 1,6 bisphosphate (FBP). The authors argue that changes in glycolytic flux (specifically aldolase A activity) change the concentration of FBP, and that FBP subsequently binds to RCC2. Binding of FBP to RCC2 prevents RCC2 binding to Rac1, thereby promoting actin reorganization, increased focal adhesions, and protrusive activity.

Although aldolase A is known to be released from the actin cytoskeleton upon Rac1 mediated actin remodeling to increase glycolytic flux, this manuscript proposes that increased glycolytic flux can also act in a reciprocal way to influence cytoskeletal dynamics. This would be a novel function for FBP and an exciting way for metabolism to drive a cell adhesion phenotype. In this referee's opinion, it is an interesting mechanism that would be of broad interest to a general cell biological audience.

We thank the Reviewer for pointing out that the mechanism we postulate is novel and exciting and for highlighting the broad general interest of our study that renders it suitable for publication in a high profile journal such as Nat Cell Biol.

That said, several major key experiments are still needed to substantiate the claim and to verify its physiological relevance.

We discuss below the crucial experiments we have now added to our manuscript to substantiate our claims.

1. One important question relates to the physiological relevance of FBP activation. In the cell line studied here by the authors, upon knock down of aldolase A, FBP increases by a staggering 10-15 fold. It is questionable whether FBP would ever increase anywhere close to that magnitude in a naturally occurring setting without knockdown. When comparing proliferating and non-proliferating cancer cells, for example, some studies report only a modest change in FBP levels of a few percent.

Indeed, this is a very relevant question. Therefore, we followed the Reviewer's advice to probe F1,6BP levels in a more naturally occurring setting without knockdown. Strikingly, we found an elevation of F1,6BP levels in spreading cells that is similar to that observed upon aldolase A knockdown as we describe in detail in our response to point B.

A. The authors should measure the actual concentrations of FBP (currently they only show relative changes after aldolase A knockdown).

We had already measured actual F1,6BP concentrations for our original manuscript version, however, only based on a fluorometric assay. Following the suggestion of Reviewer 1 we now employed liquid chromatography coupled with tandem mass spectrometry to determine the F1,6BP concentration, as described above in our response to a similar comment from Reviewer 1. These results, which are depicted in the **new Figure 5f**, nicely recapitulate the increase in F1,6BP upon aldolase A knockdown which we had previously seen based on the fluorometric assay.

B. The authors claim that FBP levels changes as glycolytic flux changes, however, they never show this. They should manipulate glycolytic flux and quantify changes in FBP in the same cell line in which this study was conducted. This could be accomplished by administering 2DG at different levels. Currently, evidence that FBP links metabolic flux to focal adhesions is missing.

To manipulate glycolytic flux, we have indeed administered 2-DG in our experiments. To show that the expected reduction in glycolytic flux is mirrored by a decrease in F1,6BP, we have also measured F1,6BP levels upon 2-DG application. This demonstrated a dramatic decrease of FBP upon 2-DG treatment in control cells and also in aldolase A depleted cells (**Extended Data Figure 5b**) which we showed to be rescued under these conditions in regards to focal adhesion number and cell size (Figure 3g, h). Apart from that, F1,6BP does not only act as glycolytic flux sensor in U-2 OS cells, but has thoroughly been established as a flux-signaling metabolite across a spectrum of species (Ortega, 2023; <https://doi.org/10.3390/biom13050765>) including in addition to mammalian cells also bacteria (Kochanowski, 2012; <https://doi.org/10.1073/pnas.1202582110>) and yeast (Peeters et al, 2017; <https://doi.org/10.1038/s41467-017-01019-z>). Regarding the evidence for F1,6BP as general glycolytic flux sensor in mammalian cells, we would like to point out two impressive papers. Tanner et al. systematically analyzed glycolytic flux in two mammalian cell lines by individually overexpressing all glycolytic enzymes. Their meticulous analysis led them to the conclusion that the "intracellular F1,6BP concentration mirrors glycolytic flux" and that the F1,6BP concentration of all analyzed metabolites in fact most strongly correlates with

glycolytic flux (Tanner et al, 2018; <https://doi.org/10.1016/j.cels.2018.06.003>). Miyazawa et al. investigated how glycolytic flux impacts embryonic development and likewise "identified F1,6BP as an in vivo sentinel metabolite that mirrors glycolytic flux" (Miyazawa et al, 2022; <https://doi.org/10.7554/eLife.83299>). Therefore, F1,6BP can be considered as a well-established and wide-spread glycolytic flux sensor consistent with the fact that we were able to replicate our findings in four independent cell lines (see **new Extended Data Figure 3**).

C. How closely do the changes in FBP levels under physiological conditions (eg, during proliferation, hypoxia, nutrient availability, etc) mimic quantitative changes in FBP under which this study was conducted? Aldolase A is unlikely to be unexpressed in wildtype cells. Its activity is regulated by mechanisms that provide less dynamic range.

Since we have postulated that F1,6BP-driven protrusion is especially relevant during cell spreading and cell migration, we decided to measure F1,6BP concentrations in confluent cell cultures in comparison to freshly seeded cells that were in the process of spreading and to sparsely plated cells that were in the process of migration. Like the Reviewer we were not expecting to observe differences in the range of an aldolase A knockdown. But to our excitement, the LC-MS/MS measurements quantified 5.4 nmol F1,6BP/mg protein in spreading cells which is very similar to the level of 5.7 nmol F1,6BP/mg protein found by the same method in aldolase A depleted cells and corresponds to a 9x increase over the 0.6 nmol F1,6BP/mg protein detected in confluent cell cultures. Migrating cells which need more localized protrusions for their motility displayed with 1.8 nmol F1,6BP/mg protein still a 3x higher FBP levels than confluent cells. Therefore, the degree of cellular protrusive activity clearly correlates with its F1,6BP level, and the increase in F1,6BP levels achieved upon aldolase depletion is in fact similar to naturally occurring changes. Thus, F1,6BP levels change indeed quite dramatically across different cell states (see **new Figure 5f, i**).

To also gain insights into changes in F1,6BP levels upon oncogenic cell transformation we consulted the literature. A study by Mazurek et al. showed that normal rat kidney cells very similar to our confluent cells contain less than 1 nmol F1,6BP/mg protein (i.e. 0.4 nmol/mg protein; Mazurek et al, 2001; DOI: 10.1038/sj.onc.1204792). However, the expression of Ras, one of the most common oncoproteins and known for its ability to rewire glycolysis (Chesney et al, 2013; DOI: 10.2174/1389201011314030002), elevated the F1,6BP concentration to 13.4 nmol/mg protein (Mazurek et al. 2001). The additional expression of the oncoprotein E7 of the human papillomavirus type 16 increased the F1,6BP concentration even further to 28.3 nmol/mg protein (Mazurek et al. 2001). Therefore, oncogenic transformations can also elevate cellular F1,6BP levels in the range of an aldolase A knockdown so that it is well imaginable that the glycolytic changes in cancer cells promote cell protrusion via our postulated F1,6BP-mediated mechanism.

D. Do changes in FBP levels less than 15-fold (which I suspect will reflect most of physiology) still impact focal adhesions?

While our result that migrating cells show a ~3-fold increase in their F1,6BP level suggests that such smaller changes in the F1,6BP concentration still impact focal adhesions, we also addressed this question by performing a single round of knockdown of aldolase A instead of our usual two rounds of siRNA transfection. This still led to a decrease in aldolase A levels, but based on LC-MS/MS measurements the concomitant increase in F1,6BP was less than half that observed upon two rounds of knockdown (2.6 nmol/mg proteins instead of 5.7 nmol/mg protein). Nevertheless, we still observed a significant increase in cell area, albeit smaller.

2. As a control, the authors knockdown PFK and GAPDH. These are relatively crude biochemical assays in that they are not especially specific. Cancer cells generally do not proliferate without glycolysis. I imagine knocking down PFK would cause the cells to become less viable, or even die, which would result in less focal adhesions. Less focal adhesions due to cell death or cells being unhealthy would mean the data cannot be used to support the authors' mechanism.

This is true, but we can show that the PFK knockdown did not have such adverse effects on cell viability in our assays as we detail below.

A. Did the authors take changes in cell viability and proliferation rate into account when quantifying the number of focal adhesions with each condition? Do cells show any changes in proliferation, invasion, attachment, etc?

To exclude that changes in cell viability interfere with the interpretation of our results, we now performed an assay with the dye SYTOX, a high-affinity nucleic acid stain that can only penetrate unhealthy/dying cells whose plasma membrane is compromised. When evaluating the number of SYTOX-labeled cells upon knockdown of PFK, aldolase A and GAPDH, we did not find any significant increase in compromised cells relative to the sample which had been treated with scrambled siRNA. This control is depicted in the **new Extended Data Figure 2e, f**.

B. Why doesn't GAPDH lead to similar phenotype as aldolase A knockout? GAPDH is immediately downstream of aldolase A. It should impede glycolytic flux. Further, it should lead to the accumulation of FBP just as aldolase A knockdown does. Quantitation of FBP would be helpful here to interpret these findings.

This is indeed a good question. First of all, we now quantified the F1,6BP level of the GAPDH knockdown also by LC-MS/MS and confirmed our earlier fluorometric measurement (Figure 3a) showing that F1,6BP levels upon GAPDH depletion are indeed as low as in the control condition (**new Figure 5f**). A possible explanation for this finding is provided by Liberti et al. who showed that GAPDH inhibition with koniginic acid resulted in a buildup of intracellular F1,6BP after 16 h of treatment. However, this increase was transient, and F1,6BP levels returned to baseline after 24 h (Liberti et al., 2017; <https://doi.org/10.1016/j.cmet.2017.08.017>) This suggests that the accumulating GAPDH substrate glyceraldehyde 3-phosphate may have been diverted into other pathways such as, for example, glycerol synthesis (Orozco et al., 2020; <https://doi.org/10.1038/s42255-020-0250-5>).

3. Specificity of mechanism to fructose 1,6 bisphosphate.

A. It is well established that F2,6BP increases glycolytic flux through PFK2. Knocking down aldolase A could increase both F1,6BP and F2,6BP production. The authors only performed assays with F1,6BP. Is binding of FBP to RCC2 specific to F1,6BP, or could they have actually elucidated a mechanism for F2,6BP? Do limited proteolysis coupled to MS provide the same results when performed with F2,6BP as F1,6BP?

No peak with the retention time and mass spectral properties of F2,6BP was detected during the LC-MS/MS analysis of the samples. This suggests that the levels of F2,6BP were below the limits of detection in the cells or that F2,6BP, which is very sensitive to acid hydrolysis, was not preserved during sample preparation. Indeed, there is a large amount of literature stating that the concentration of this regulatory molecule is typically about one to two orders of magnitude lower than F1,6BP so that its contribution to the total FBP pool is very low. For

example, a study analyzing HeLa and rat hepatoma cells reported 90-fold to 4200-fold lower levels of F2,6BP (measured by an elaborate enzyme activation assay based on purified components) compared to F1,6BP (Marín-Hernández, 2011; Table 3 and Table 4; <https://doi.org/10.1016/j.bbabc.2010.11.006>). Similarly, studies of mouse tissues such as rat muscles demonstrated 375-fold up to 1000-fold lower levels of F2,6BP compared to F1,6BP depending on the used stimulation protocols (Minatogawa, 1984; Table 1 and Table 2; <https://doi.org/10.1042/bj2230073>).

We also followed the Reviewer's suggestion to repeat the limited proteolysis MS with F2,6BP instead of F1,6BP. On this basis, it is indeed not possible to exclude that F2,6BP also interacts with RCC2 (**new Extended Data Figure 9c**). However, due to the low general F2,6BP concentration, it is hard to envision F2,6BP to be physiologically more relevant as an RCC2 binder than F1,6BP. We present these new findings within our results section to alert the readers to this theoretical possibility. We would also like to point out that the fact that the PFK1 knockdown completely rescues the aldolase knockdown also supports the notion of a F1,6BP-dependent mechanism.

4. Can the authors introduce a mutation into RCC2 that prevents FBP binding but enables its WT activity? Given that FBP and Rac1 could bind in the same region, this may not be possible. If feasible, however, it would provide compelling evidence for the proposed mechanism. Can the authors incubate RCC2 with Rac1 and increasing concentrations of FBP, followed by a co-immunoprecipitation analysis?

The first part is beyond our possibilities since we could not identify a mutation with the required characteristics so far. The chances seem also low since F1,6BP and Rac1 likely compete for the same binding site on RCC2. However, we did succeed with setting up an interaction assay for RCC2 and Rac1 which allowed us to show that FBP can outcompete Rac1 binding (**new Figure 8m, n**).

5. While I do not think the authors need to do any *in vivo* experiments to prove their mechanism, I do agree that it would be useful to obtain data from more than a single cell line to assess broad applicability and reproducibility.

We completely agree that such a new mechanism should be shown in more than one cell line. Therefore, we had confirmed our findings already in MDA-MB-231 and hTERT RPE-1 cells (**Extended Data Figure 3a-h**). Now we show in addition that they also hold true for human foreskin fibroblasts (**new Extended Data Figure 3i-l**). Moreover, using an independent set of siRNAs against Rac1 we replicated the finding that the increase in focal adhesion numbers and cell size upon aldolase A depletion is dependent on Rac1 (**new Extended Data Figure 7b-e**). In conclusion, the effect of increased F1,6BP levels on cell protrusion and focal adhesions was recapitulated in two cancer cell lines and two non-malignant cell lines.

Minor:

1. Statistics: From statistical lines in bar graphs, it is sometimes unclear which groups are being compared.

We have now made sure that it is always clear from the visual depictions and/or figure legends which groups are being compared.

Some bars look significant, but do not include any statistical test or p-value.

This is true, for instance in Figure 2 we only indicate the results of the statistical comparisons of the different conditions with the control condition since this is sufficient to support our respective hypothesis. Where possible without loss of important information, we have refrained from testing all possible comparisons since this is largely redundant and quickly overloads the visual data presentation. However all the statistical source data are contained in the accompanying source data file so that all possible comparisons are accessible for interested readers.

Some plots appear to only have $n=2$, which is underpowered.

To our knowledge, there was only a single experiment which was performed with $N=2$ which was the measurement of absolute F1,6BP concentrations by the fluorometric assay. Since the results were very black and white and clearly confirmed our earlier measurements of the relative increase in F1,6BP concentration upon aldolase A knockdown, we had foregone to repeat the absolute quantification one more time. However, this shortcoming is now resolved due to the fact that we newly measured the F1,6BP levels for the revision by LC-MS/MS with $N=3$.

All ANOVAs were conducted using Dunnett's post-test. Although this test is appropriate, a Tukey's post hoc test might be more beneficial.

As Reviewer 2 also pointed out there is for instance the quantification in Figure 8k, l which would indeed benefit from a Tukey's post hoc test. Therefore, we have now redone the statistical analysis with this test.

2. The legends for Supplemental Figure 8B and C appear to have been accidentally flipped.

Thanks for pointing this out. The legends of Extended Data Figure 7b and c had indeed been accidentally flipped. We have now corrected this mistake (**new Extended Data Figure 9b, d**).

3. Some prior work has indicated that FBP is membrane permeable (doi: 10.1007/s11010-012-1279-x, doi: 10.1023/b:mcbi.0000021356.89867.0d, doi: 10.1152/ajpheart.1994.267.6.h2325). Although surprising, if true, this might be a useful approach to the authors to test mechanism.

Yes, we had also come across some papers where F1,6BP was applied from the outside. This would indeed be a great way of testing our proposed mechanism. However, we were not able to replicate these findings with our cellular model systems.

4. Figure captions use the notation $N=3$. The authors should clarify what this means (eg, how many cells/plates/etc.).

The N is always the number of independently performed experiments. We have double checked our figure legends to make sure that this is always made clear.

5. Although it was once thought that cancer cells switch their metabolism from oxphos to glycolysis as the authors state in the Discussion, increasing data suggests that oxphos increases to its maximum rate of activity and glycolysis is an overflow channel (doi: 10.1016/j.tcb.2023.03.013).

True, the situation is certainly more complex than a clear switch from oxphos to glycolysis, and the review the Reviewer pointed out on this topic was very interesting to read. We have now

modified our statement in the discussion in order to reflect the current ideas about the Warburg effect in a more accurate manner.

Point-by-point Responses to Reviewer Comments

Reviewer #1:

The authors successfully addressed all my concerns.

We are happy that we could dispel all concerns of Reviewer 1.

Reviewer #2:

The authors have performed new experiments and included fantastic new data and analyses to answer my comments. I'm happy to recommend for publication.

This praise of Reviewer 2 regarding the data we added during our last revision made us very happy, and we thank the reviewer for his/her recommendation to publish our manuscript.

Reviewer #4:

This article shows very interesting data on the connection between the glycolytic intermediate fructose 1,6-bisphosphate (FBP) and cell adhesion mediated by the interaction between Rac1 and its negative regulator RCC2. Based on the data from a whole genome siRNA screen, the authors show that silencing of ALDOA leads to increased cell area and a higher number of focal adhesions per cell. The authors then show that the catalytic activity of ALDOA rather than substrate binding is responsible for this effect. Furthermore, the authors show that silencing of PFK (all three isoforms) reduces cell area and focal adhesion number while depletion of GAPDH has no effect. Metabolomics analysis revealed that ALDOA silencing resulted in increased levels of its substrate, fructose 1,6-bisphosphate (FBP) and inhibition of FBP accumulation by inhibition of hexokinase prevented increased cell area. Using time-lapse analysis, the authors next show that PFK silencing reduces focal adhesion assembly while ALDOA depletion increase focal adhesion assembly rates, consistent with a role of FBP in this process. The authors next employ LiP-MS to show that FBP induces conformational changes in Rac1 and that Rac1 is required for FBP-mediated F-actin reorganisation. Finally, the authors also implicate RCC2, a negative regulator of Rac1, in this process, resulting in a model in which FBP leads to the disassembly of RCC2 from Rac1 to promote cell spreading. Overall, the manuscript shows highly interesting data. A particular strength lies in the identification of Rac1 and RCC2 as FBP binding proteins by LiP-MS. This finding offers interesting insight into metabolite/protein interactions and provides an important link between metabolic activity and cell regulation. However, there are some issues that need to be addressed before the manuscript can be considered for publication.

We thank Reviewer 4 for regarding our data as highly interesting. We will address his/her specific comments below.

Specific comments:

1) The authors discuss that FBP levels tightly correlate with glycolytic flux in many systems, making FBP a signalling metabolite that informs about the metabolic state of a cell. However, experiments shown in the manuscript mostly use siRNA-mediated depletion of FBP-producing and FBP-consuming enzymes, i.e. PFK and ALDOA, rather than studying the effect of modulating glycolytic flux of Rac1 activation. It is not clear whether changes in FBP levels observed during modulation of glycolytic flux, for example after acute exposure to glucose, are sufficient to trigger Rac1 activation. Under these conditions, changes in FBP levels are likely to be smaller and more transient compared to those observed after ALDOA depletion. The authors include results to show that FBP levels are comparable to those found in migrating or spreading cells. However, these situations reflect the reverse regulation, i.e. cytoskeletal

changes modulating glycolytic flux rather than the opposite. Given the broad and interesting implications of the findings, a more detailed analysis of the link between glycolytic flux and FBP-mediated Rac1 activation would be warranted.

The fact that acute glucose treatment can trigger Rac1 activation is well-established, while the mechanism behind this observation has remained elusive for more than twenty years. Li et al. reported already in 2004 for the β -cell line INS-1 that an increase in glucose leads to an increase in GTP-Rac1 which is crucial to trigger the necessary actin remodeling for efficient glucose-dependent insulin release (Li et al., 2004; Fig. 4; <https://doi.org/10.1152/ajpendo.00307.2003>).

This finding has meanwhile been reproduced by at least three other groups (Kolic et al., 2014; Fig. 6; <https://doi.org/10.1074/jbc.M114.577510>; Elumalai et al., 2017; Fig. 1, Fig. 5; <https://doi.org/10.1016/j.redox.2016.11.009>; Thamilselvan & Kowluru, 2019; Fig. 4; <https://doi.org/10.1080/21541248.2019.1635403>). Furthermore, the ability of glucose to trigger Rac1 activation was confirmed also for the Chinese Hamster Ovary cell line K1 (CHO K1; Lamers et al., 2012; Fig. 4; <https://doi.org/10.1371/journal.pone.0022865>). Interestingly, Lamers et al. observed also that the glucose treatment resulted in an increased protrusive activity which supports the link between glycolytic flux and cell protrusion which we propose.

In addition, glucose was shown to induce elevated Rac1 activity in human umbilical vein endothelial cells (HUVECs), primary human platelets, mouse mesenteric arteries (Schiattarella et al., 2018; Figs 3, 4, S2A; <https://doi.org/10.1161/JAHA.117.007322>), human aortic endothelial cells (Vecchione et al., 2005; Fig. 4; <https://doi.org/10.1161/01.RES.0000200440.18768.30>) and cultured adult rat ventricle cardiomyocytes (Shen et al., 2009; Fig. 1; <https://doi.org/10.2337/db08-0617>). Interestingly, an increase in Rac1 activity was also observed in vivo in murine aorta downstream of an elevation in blood glucose triggered by the induction of diabetes in a mouse model (Vecchione et al., 2005; Fig. 1; <https://doi.org/10.1161/01.RES.0000200440.18768.30>)

However, we agree with Reviewer 4 that it would further strengthen our manuscript to also show the acute effect of glucose on Rac1 activation and consequently on the ability of cells to protrude and spread in one of our cellular systems. We chose the non-cancerous hTERT RPE-1 cell model for this and include the new data as **Fig. 7 e-h**. Our new data clearly shows that cell spreading strikingly depends on glucose. In fact, cells can even spread in the absence of serum and without an extracellular matrix coating as long as glucose is present, with 1 mM glucose being already a sufficient trigger. To rule out that this is solely an effect of glucose-dependent energy production, cells were provided throughout the experiment with pyruvate and the L-glutamine substitute GlutaMAX as alternative energy sources. As expected, the glucose-induced cell spreading is accompanied by a significant increase in Rac1-GTP of ~2-fold. Therefore, these new experiments clearly support our hypothesis of FBP-mediated Rac1 activation.

In addition, we also view the high FBP levels which we detected in spreading cells as strong point supporting the notion of a link between glycolytic flux and FBP-mediated Rac1 activation. Reviewer 4 stated in this regard that "these situations reflect the reverse regulation, i.e. cytoskeletal changes modulating glycolytic flux rather than the opposite", thus saying that the high FBP levels during cell spreading are triggered by the associated cytoskeletal changes rather than causing them. Reviewer 4 did not provide a reference for this, but we assume that this idea is based on the findings by Hu et al. (Hu et al., 2016; <http://dx.doi.org/10.1016/j.cell.2015.12.042>) where the authors showed that insulin signaling

via PI3 kinase in breast epithelial cells leads to a cytoskeletal reorganization that triggers the dissociation of aldolase A from actin and its translocation to the cytosol, thereby activating it.

However, there was no addition of insulin in our experimental settings, and we also never observed a translocation of aldolase A in our cells, rendering it unlikely that Rac1-dependent actin remodeling triggers the high FBP levels we measured rather than the high FBP levels triggering Rac1 activity.

In line with this, Kondo et al. (Kondo et al, 2021, <https://doi.org/10.1016/j.celrep.2021.108750>) provide a possible explanation for how elevated FBP levels can come about in a similar situation: They showed that cells with space around them increase their glucose uptake by upregulating the glucose transporters GLUT1 and GLUT4. In this manner, the cellular sensing of available space for protrusion translates into an increased glucose uptake, presumably leading to enhanced glycolysis and elevated FBP levels triggering Rac1-mediated protrusive activity.

The idea that Rac1 activity is downstream rather than upstream of glycolytic events is further supported by the mentioned findings from β -cells. In these cells FBP levels were shown to increase in line with increasing glucose concentrations (Koberstein et al., 2022; <https://doi.org/10.1073/pnas.2204407119>) which at the same time trigger elevated Rac1 activity (Li et al., 2004; <https://doi.org/10.1152/ajpendo.00307.2003>). This elevated Rac1 activity is in β -cells a prerequisite for the mobilization of insulin granules from behind an actin barrier to allow insulin secretion. Thus, the order of events here appears to be that glucose triggers increased FBP levels and activates Rac1 activity prior to cytoskeletal remodeling.

2) It is quite surprising that siRNA mediated depletion of glycolytic enzymes (PFK, ALDOA, GAPDH) using a double transfection protocol does not have major effects on cell number after 96 hours of incubation. The authors use CYTOX to confirm cell viability but they should also perform experiments to demonstrate any effects of proliferation. Changing cell cycle distribution in the population is likely to also have effects on cell size and focal adhesions.

Reviewer 4 is right that cellular problems with cell cycle progression can have effects on cell morphology and FAs, especially since different cell cycle phases have been associated with different levels of FAs. This was prominently shown by Jones et al. (Jones et al., 2018; Fig. 1B; <https://doi.org/10.1083/jcb.201802088>) who revealed that there is a large increase in adhesion complex area as cells transition from G1 into S. In fact, cells in S phase can have more than twice the number of FAs than cells in G1 or G2 phase. Therefore, we now analyzed the cell cycle profile of the different knockdown cells after an incubation with EdU to label cells in S phase. This new data is presented in **Extended Data Fig. 2g**. As our results show, cells silenced for aldolase A were to a lower proportion in S phase. These data align well with the cell cycle profiling results of aldolase A silenced liver cancer cells from the paper of Snaebjornsson et al. (Snaebjornsson et al., 2025; <https://doi.org/10.1038/s42255-024-01201-w>) which Reviewer 4 mentions under point 4. Therefore, we can exclude that the higher number of adhesions in aldolase A knockdown cells is due to an enrichment of these cells in S phase, as now also stated on page 5 of our manuscript.

3) As ALDOA is also a target for cytoskeletal regulation via Rac1, it is surprising that the authors did not include experiments using the actin binding mutant of ALDOA (R43A) in their experiments shown in Figure 2a-c.

We agree with Reviewer 4 that it would be an interesting experiment to investigate the ability of an actin-binding deficient mutant of aldolase A to rescue the FA phenotype of aldolase A knockdown cells. However, when the properties of the known actin-binding deficient mutant of aldolase A, R42A, were analyzed in detail in the past (Wang et al., <https://www.ncbi.nlm.nih.gov/pubmed/8636111>), it became apparent that this is not a "clean" actin-binding deficient mutant in the sense that also its catalytic abilities are impacted with V_{\max} being only half that of the wt aldolase A and K_m being 3x as high (see table below with data from Wang et al.). Thus, it is likely that the result of such an experiment would not be easy to interpret. Therefore, and knowing also that the editor regarded this point as beyond the scope of the present study, we refrained from performing this experiment.

	WT	Catalytic mutant D33S	Actin-binding mutant R42A
V_{\max} (Units/mg)	20.8 ± 0.5	$0.0056 \pm 1 \times 10^{-6}$	10.2 ± 0.4
K_m (μ M)	14.3 ± 0.7	36.5 ± 0.002	45 ± 5

4) *The authors mention that accumulation of FBP in response to ALDOB downregulation would promote cancer cell spreading and potentially drive liver carcinogenesis. However, a recent study demonstrated that ALDOA is essential for liver tumour formation in a mouse model by preventing imbalanced glycolysis, leading to very high levels of FBP accumulation (DOI: 10.1038/s42255-024-01201-w). The authors should include this publication in their discussion.*

This is an interesting publication which we now include in our discussion on page 10 with the following statement:

"Of note, the situation might differ depending on tumor type and aldolase expression pattern. For example, for liver cancers high aldolase A levels have been associated with a poor patient survival⁴¹; and the knockdown of aldolase A in liver cancer cells with their specific metabolic wiring does not only lead to increased FBP levels, in line with our data, but also to energy exhaustion and strongly reduced cell viability due to imbalanced glycolysis⁴¹."

The situation across different cancer cell types is certainly complex due to their high metabolic plasticity. For curiosity, we tested with the help of GEPIA2, the same tool Snaebjornsson et al. used, whether the relationship between high aldolase A and poor patient survival holds true for other tumors. However, for sarcoma cells, the category U-2 OS cells belong to, there is no significant correlation.

5) *There is no method provided for the LC-MS/MS analysis of FBP levels. It is essential to provide a full protocol, including the use of standards for FBP quantification.*

To keep the methods section short, we had so far described only the parts of the FBP LC-MS/MS analysis protocol which were specific to our experiment while referencing earlier publications for the experimental details that had been described previously. However, of course, it is no problem to provide the entire protocol within the methods section of our manuscript which we have done now.